# Wind dataset assessment and energy estimation for potential future offshore wind farm development areas on the Scotian Shelf

Yongxing Ma[1], Jinshan Xu[1], Yongsheng Wu[1], Michael Z. Li[2], Ryan Stanley[1], Brent Law[1], and Marc Skinner[1]

[1]Fisheries and Oceans Canada, Bedford Institute of Oceanography, 1 Challenger Drive, Dartmouth, B2Y 4A2, Nova Scotia, Canada
[2]Geological Survey of Canada (Atlantic), Bedford Institute of Oceanography, 1 Challenger Drive, Dartmouth, B2Y 4A2, Nova Scotia, Canada

**Correspondence:** Yongxing Ma (Yongxing.Ma@dfo-mpo.gc.ca) and Yongsheng Wu (Yongsheng.Wu@dfo-mpo.gc.ca)

**Abstract.** The Scotian Shelf is one of the top wind regions in the world. To assess the wind energy potential on the shelf, in this study we first assessed the uncertainties of 4 commonly used wind datasets: ERA5, CFSv2, NARR, and HRDPS, by comparing them against observational wind data from both nearshore and offshore sites. The assessment showed that the Root Mean Square Error (RMSE) of the datasets ranged from 1.6 m/s to 2.4 m/s in wind speed and from 24.6° to 36.4° in wind direction. HRDPS performed best at the nearshore sites, while ERA5 was more accurate at the offshore sites. We then estimated the wind energy potential of 6 potential future development areas (PFDAs) on the shelf using ERA5. The estimates showed that wind energy varied seasonally, with summer wind energy production being 34–40% lower than in winter. The uncertainties in wind datasets amplified the variation in wind energy production, up to 28% in winter and 50% in summer. The energy output was sensitive to turbine spacing due to wind wakes, which reduced energy production by 19–30% in winter and 37–46% in summer, depending on the configuration of wind speeds, wind directions, and the specific layout of the wind farms. This strong variation in wind energy output suggests that a more feasible operational method should be used to balance energy production and usage.

## 1 Introduction

### 1.1 Background

Offshore wind farms have rapidly developed globally over the past decade (World Forum Offshore Wind, 2024), driven in part by greater consistency and abundance of wind resources compared to onshore. By the end of 2023, the capacity in operation of global offshore wind farms had reached 67.4 GW, and is projected to reach 414 GW by 2032, which is a significant increase from 7.9 GW in 2014 (World Forum Offshore Wind, 2024). Although no wind turbines have been installed in Canadian offshore waters to date, offshore wind is expected to play a key role in Canada's electricity portfolio in support of the country's net-zero emissions goal by 2050 (Canada Energy Regulator, 2023). Nova Scotia's offshore waters rank among the world's best wind resources, with average wind speeds of 9–11 m/s at 100 m above the ocean surface (Aegir Insights ApS, 2023; Nicholson,

2023). The federal and provincial governments plan to offer leases for 5 GW of offshore wind development on the Scotian Shelf by 2030 (Government of Nova Scotia, 2023).

As planning for offshore wind development on the Scotian Shelf progresses, there remains a lack of comprehensive assessments of wind datasets and, particularly, estimates of wind energy potential that accounts for wake effects associated with varying wind turbine spacing. To address this gap, this study evaluated available wind datasets, comparing their accuracy against regional wind observations to better inform the region's offshore wind potential. These datasets were then used to simulate wind farm performance across potential future development areas (PFDAs), incorporating turbine spacing and wake effects to assess their influence on energy production. While the PFDAs analyzed in this study generally align with proposed offshore wind energy areas for the Scotian Shelf, their exact locations, shapes, and sizes may differ from final areas that are approved for development.

This research aimed to provide a more accurate estimate of wind energy potential on the Scotian Shelf, as well as offer insight into future wind farm planning, design, and development in the region. The manuscript is structured as follows: Section 2 introduces the wind datasets, regional wind observations, and metrics used for evaluation, along with the PyWake model configuration; Section 3 presents the wind dataset assessment results for wind speed and wind direction; Section 4 presents PFDAs power production simulation results; and finally, Sections 5 and 6 present discussions and conclusions, respectively.

## 1.2 Wind Datasets

Offshore wind development on the Scotian Shelf requires reliable wind resource assessments to guide investment and planning, particularly as this industry is new to the region. Previous studies evaluating potential power generation for the PFDAs on the Scotian Shelf have relied on climatological wind speeds and idealized conditions (i.e., Aegir Insights ApS (2023); Kilpatrick et al. (2023)). However, these studies did not account for turbine wake effects, which can significantly influence overall energy potential and lead to inaccurate energy estimates. A more robust approach involves simulating offshore wind farms using numerical models that incorporate time-varying wind speed and wind direction data from wind datasets, providing a more accurate foundation for decision-making.

There are several reanalysis and forecast wind datasets that cover the Scotian Shelf region, including: 1) the fifth-generation European Centre for Medium-Range Weather Forecasts (ECMWF) atmospheric reanalysis (ERA5); 2) Climate Forecast System Version 2 (CFSv2); 3) North American Regional Reanalysis (NARR); and 4) High-Resolution Deterministic Prediction System (HRDPS). Assessments of wind speed from these datasets have been carried out for other regions (e.g., Fan et al., 2021; Gualtieri, 2021; Kardakaris et al., 2021; Wang et al., 2019). Although these assessments have had varied objectives, such as dataset evaluation (Milbrandt et al., 2016), inter-dataset comparison (Wang et al., 2019; Fan et al., 2021), and wind energy estimations (Li et al., 2010; Murcia et al., 2022), they all strengthened our understanding of different wind datasets and provided guidance in selecting the most suitable dataset for specific applications.

The ability of wind datasets to represent real world conditions is commonly assessed by comparing them with observational wind measurements and evaluating statistical metrics such as Root Mean Square Error (RMSE), bias, Mean Absolute Error (MAE), and coefficient of determination ($R^2$) (Gualtieri, 2021; Fan et al., 2021; Kardakaris et al., 2021; Wang et al., 2019;

Milbrandt et al., 2016; Murcia et al., 2022). To align gridded wind datasets with the more limited observational data, horizontal interpolation using 2-D linear or cubic methods, and vertical extrapolations using a power law relationship, assuming atmospheric neutral stability (Wang et al., 2019; Kardakaris et al., 2021; Murcia et al., 2022), can be used.

Among the 4 wind datasets, ERA5 has been the most widely assessed and often deemed to be one of the most accurate. Fan et al. (2021) evaluated 5 wind datasets (i.e., ERA5, ERA-Interim, JRA-55, MERRA-2, and CFSv2) by comparing 10 m wind speed data to wind observations from over 1000 meteorological stations worldwide. The authors found that ERA5 demonstrated the best overall performance among the 5 reanalysis wind dataset products, with ERA5 exhibiting a mean percent bias for all stations of -4.54%, while the mean percent bias was -54.22% for JRA-55, -49.63% for CFSv2 and 42.03% for MERRA-2. Similarly, in a recent dataset validation study, Murcia et al. (2022) compared multiple wind datasets with wind observations from various sites across Europe and found that after calibration, ERA5 outperformed all other datasets, including the European-level atmospheric reanalysis (EIWR). In general, the ERA5 dataset exhibited the lowest MAE, smallest RMSE, and highest correlation coefficient.

Gualtieri (2021) compared ERA5 wind speeds against wind measurements taken from 6 tall towers spread across a diverse range of global locations. This comparison noted that the normalized bias of wind speed ranged from -0.18 to 0.53, while the correlation coefficient between ERA5 and wind observations varied from 0.38 to 0.96, depending on location. Similar findings were reported by Fan et al. (2021), whose results indicated notable regional differences, with the percent bias for ERA5 ranging from -11.55% in Australia to 16.13% in Central Asia.

Even within a relatively small region, wind dataset reanalysis products can exhibit considerable spatial variability. For example, Kardakaris et al. (2021) assessed ERA5 wind speed using measurements from 6 buoys in the Greek seas and found that the relative difference between ERA5 and observed wind speeds ranged from 6.5% to 34.7%. Similarly, Fernandes et al. (2021) compared ERA5 wind speed data at the height of 100 m above sea surface with wind observations from both coastal and offshore sites in Brazil. The findings showed that in the coastal region the bias was less than 0.5 m/s (with a mean wind speed of approximately 6 m/s), whereas in the offshore region the bias was nearly 0 (with a mean wind speed of 7.19 m/s).

Li et al. (2010) compared wind speeds at 80 m height from rawinsonde observations at 5 stations in the Great Lakes region of the United States with those from the NARR wind dataset over periods ranging from 14 to 30 years. The all-time mean wind speeds at the 5 stations ranged from 5.35 m/s to 6.18 m/s, with biases between -0.64 m/s to 0.59 m/s, and correlation coefficients close to 0.8. These results suggest that NARR provided an accurate simulation of wind speed in the study region. Further, Wang et al. (2019) assessed the 10 m wind speed and wind direction from various datasets, including NARR, against wind observations from three ocean buoys along the Central California Coast. The authors found that the NARR dataset generally underestimated wind speed compared to observations from all three buoys, where the bias ranged from -2.78 m/s to -0.15 m/s and RMSE from 1.90 m/s to 4.00 m/s for mean wind speeds of 4–11 m/s.

There are limited studies that have evaluated the HRDPS dataset against observed wind speeds (e.g., Milbrandt et al., 2016; Moore-Maley and Allen, 2022). Notably, in a nearshore area, Moore-Maley and Allen (2022) examined 5-year hourly surface wind speed records against wind observations from 4 stations (meteorological stations and ocean buoys) in the Salish Sea. The

authors observed an overall qualitative consistency between HRDPS and the observations in terms of wind speed and wind direction.

In general, most wind dataset assessment studies have focused on the evaluation of wind speed, with fewer studies assessing wind direction (Moore-Maley and Allen, 2022). Assessing wind direction, however, is important for the purpose of conducting wind farm simulations, as wind direction and turbine layout can influence wind farm efficiency due to wake effects (Gaumond
et al., 2014; Stieren et al., 2021).

## 1.3   Wake Effects

In general, studies that have estimated offshore wind farm energy potential from wind datasets (e.g., Wang et al., 2022; Gualtieri, 2021; Kardakaris et al., 2021; Fernandes et al., 2021) typically estimate wind power using simple formulas or interpolate using wind turbine power curves. However, these approaches can overlook a key aspect of real-world conditions;
primarily, wind turbines can generate wake effects that reduce wind speeds available to downstream turbines, leading to lower overall energy production from a wind farm. Wake effects have been estimated to result in energy losses on the order of 10% to 25% in medium-sized offshore wind farms, such as the Horns Rev, Lillgrund, and Nysted wind farms (Barthelmie et al., 2009, 2010; Niayifar and Porté-Agel, 2015; Simisiroglou et al., 2019; Wu and Porté-Agel, 2015). For large-sized offshore wind farms, Pryor et al. (2021) estimated, through simulation, an overall 35.3% energy loss associated with wake effects.

Given the impact wake effects can have on wind farm efficiency, substantial research has been dedicated to predicting turbine wakes using analytical models (e.g., Bastankhah and Porté-Agel, 2014; Jensen, 1983; Niayifar and Porté-Agel, 2015), numerical simulations (e.g., Calaf et al., 2010; Pryor et al., 2021; Stevens, 2016; Troldborg et al., 2010), and laboratory experiments (e.g., Chamorro and Porté-Agel, 2010). In recent studies (e.g., Fischereit et al., 2022; Murcia et al., 2022), wake effects were incorporated into wind farm energy production estimates using PyWake (Pedersen et al., 2023), which is a Python package
designed to efficiently calculate wake interactions in wind farms.

In addition to wake effect models, wind farm simulations also require detailed turbine models and turbine layouts. Older offshore wind farms deployed smaller turbines; for example, the Horns Rev wind farm utilized Vestas V80 2 MW turbines (Hansen et al., 2012). In contrast, Siemens 2.3 MW turbines were installed at Nysted and Lillgrund (Barthelmie et al., 2010; Simisiroglou et al., 2019). More recently, there has been a move towards installation of larger turbines. The average rated
capacity of installed offshore wind turbines globally has been increasing, with an average of 4.0 MW in 2013, 9.7 MW in 2023, and a projected increase to 14.8 MW by 2028 (McCoy et al., 2024). In the U.S., several offshore wind farms currently under construction are now incorporating 15 MW turbines (Tetra Tech Inc., 2022).

Turbine spacing is a critical factor that influences wake effect energy losses in a wind farm. Larger turbine spacing allows downstream wind more space to regain velocity through turbulent mixing, which draws kinetic energy downward from higher
atmospheric layers (Frandsen, 1992). This larger spacing thus can improve the efficiency of downstream wind turbines, compared to those spaced closer together. However, increased spacing can also lead to overall reduced energy generation given fewer turbines being emplaced within a development area. These factors emphasize the importance of understanding the trade-offs between turbine spacing and wake effects, in an effort to inform overall economics of wind farm planning, design, and

development (Mulas Hernando et al., 2023; Stevens et al., 2017). Typical turbine spacing ranges from 4 to 11 D, where D is the turbine rotor diameter (Bosch et al., 2019; Pryor et al., 2021; Stevens et al., 2017). At the Lillgrund offshore wind farm in Sweden, turbine spacing ranges 3.3 to 4.3 D (Simisiroglou et al., 2019), while at the Horns Rev offshore wind farm in Denmark the turbines are spaced at 7 D (Barthelmie et al., 2010).

## 2 Datasets and Methods

### 2.1 Regional Wind Observations

Hourly wind data from meteorological stations within the Scotian Shelf area were obtained from the Government of Canada's Historical Climate Data website (https://climate.weather.gc.ca). Two island-based meteorological stations located at a nearshore site (Beaver Island) and an offshore site (Sable Island) on the Scotian Shelf were selected for analysis. At the meteorological stations, wind speed and direction were measured using type U2A cup anemometers mounted on masts at a height of 10 m above ground level (Environment and Climate Change Canada, 2023; Wan et al., 2010). For the oceanic domain, wind data were obtained from moored marine buoy sites. Four buoys were selected based on data coverage for the analysis period and minimal gaps in observed wind data (Figure 1). These data were obtained from the Fisheries and Oceans Canada (DFO) Marine Environmental Data Section Archive (https://meds-sdmm.dfo-mpo.gc.ca). On the marine buoys, two types of anemometers, the R.M. Young helicoid propeller-vane anemometer and the Vaisala WS425 Ultrasonic, were mounted at different positions but at approximately the same height of 5 m above the sea surface (Thomas and Swail, 2011). Upon comparison, the wind data records from the Vaisala WS425 ultrasonic anemometers were found to be more persistent, with fewer invalid data points over time. Therefore, only data from this instrument were used in the analysis. The wind direction from observations at all sites, as well as throughout this manuscript, refers to the direction from which the wind blows, expressed in units of degrees true, meaning degrees clockwise from true north. All sites were summarized in Table 1. The sites were numbered in a sequence based on distance away from the coastline of Nova Scotia and in a northeast to southwest direction. Due to different regimes of wind dynamics (Cañadillas et al., 2023; Djath et al., 2022), the sites have been categorized as nearshore (Sites 1 and 2) and offshore (Sites 3–6).

Wind speed and direction data obtained from meteorological stations and marine buoys did not include quality flags (quality flags were only available for ocean wave variables in the buoy data). To ensure data reliability, a basic quality control procedure was applied: wind speed values equal to 0 m/s or greater than 50 m/s were considered erroneous and excluded from the analysis. The number of valid hourly records per site and per month is summarized in Figure A1. For the purpose of wind dataset assessment, any month with fewer than 120 valid hourly records was considered invalid and omitted from further analysis.

### 2.2 Wind Datasets

The ERA5 dataset, developed by ECMWF, is a reanalysis climate product that assimilates historical observational data globally (Hersbach et al., 2020). It has global coverage with spatial resolution of 0.25° and spans from January 1940 to the present

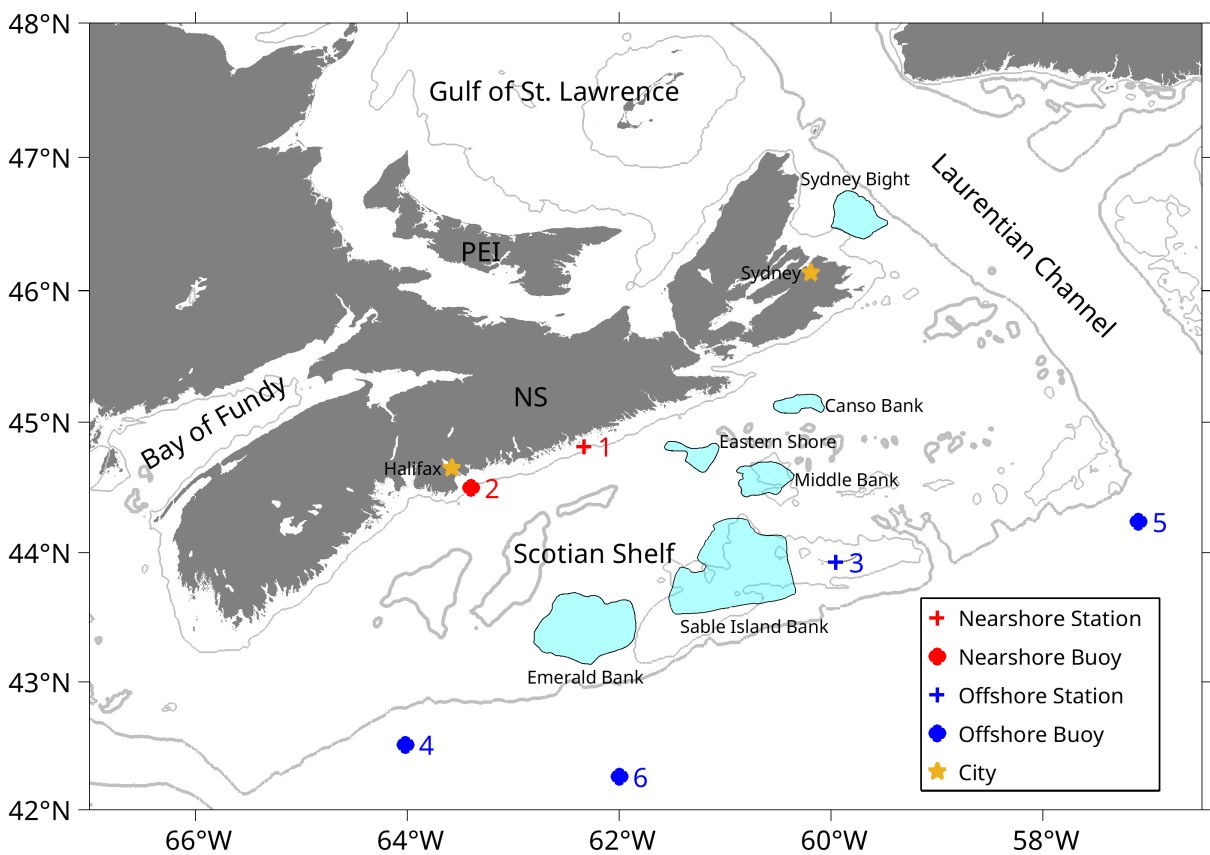

**Figure 1.** Map of the Scotian Shelf study area located in the offshore of Nova Scotia, Atlantic Canada. The map illustrates locations of regional wind observation sites, including meteorological stations (+) and marine buoys (●) at both nearshore (red) and offshore (blue) locations. The potential future development areas (PFDAs) for offshore wind farms used in this study are shown as blue polygons with names labeled alongside. These PFDAs are adapted from general areas described by Committee for the Regional Assessment of Offshore Wind Development in Nova Scotia (2024). Although the PFDAs used in this study generally align with offshore wind energy areas being discussed for the Scotian Shelf, the exact areas used in this study may differ in location, shape, and size from those areas finalized by regulators for offshore wind development consideration. Cities are illustrated with yellow stars. Contour lines at 100 m and 200 m isobaths are depicted with thin and thick grey curves, respectively. NS = Province of Nova Scotia; PEI = Province of Prince Edward Island.

with hourly frequency. The 10 m wind velocity components in east-west and north-south directions can be accessed at the Copernicus Climate Data Store (Hersbach et al., 2023).

     The CFSv2 is a coupled model that contains ocean, land, and atmosphere components (Saha et al., 2014). The National Centers for Environmental Prediction (NCEP) provides selected hourly time-series products of CFSv2 dataset that span from April 1, 2011, to the present. Hourly time series of 10 m wind velocity components in two directions, with a 0.2° horizontal

resolution, can be accessed from the Research Data Archive at the National Center for Atmospheric Research (Saha et al., 2011).

**Table 1.** Information on the meteorological stations and marine buoy sites used in this study. For station height represents the elevation above sea level, while the instrument height represents the mounted height of the wind measurement device from the station.

| Site | Longitude (°W) | Latitude (°N) | Station Height (m) | Instrument Height (m) | Group | Type |
|------|------|------|------|------|------|------|
| 1 | 62.33 | 44.82 | 16.0 | 10.0 | nearshore | meteorological station |
| 2 | 63.40 | 44.50 | 0.0 | 5.0 | nearshore | marine buoy |
| 3 | 59.96 | 43.93 | 1.2 | 10.0 | offshore | meteorological station |
| 4 | 64.02 | 42.51 | 0.0 | 5.0 | offshore | marine buoy |
| 5 | 57.10 | 44.24 | 0.0 | 5.0 | offshore | marine buoy |
| 6 | 62.00 | 42.26 | 0.0 | 5.0 | offshore | marine buoy |

The NARR dataset produced by NCEP provides a high-resolution reanalysis of atmospheric variables, including wind velocities (Mesinger et al., 2006). The 3-hourly wind velocity with a 32 km spatial resolution at the 10 m height can be acquired from the Research Data Archive at the National Center for Atmospheric Research (National Centers for Environmental Prediction, National Weather Service, NOAA, U.S. Department of Commerce, 2005).

The HRDPS developed by Environment and Climate Change Canada (ECCC) is a high-resolution numerical weather prediction model with assimilation (Milbrandt et al., 2016). It has a spatial resolution of 2.5 km and an hourly temporal frequency. The dataset spans from April 23, 2015, to the present, with coverage extending across Canada and its surrounding marine regions. Information on accessing the HRDPS dataset can be found at the Meteorological Service of Canada Open Data portal (https://eccc-msc.github.io/open-data/msc-data/readme_en/).

## 2.3 Spatial and Temporal Interpolation

Since the wind datasets and regional wind observations do not align in space or time, the respective coordinates were standardized. To do this, a 2-D linear interpolation was applied to the gridded wind datasets to match the wind observation site locations. Since the wind datasets provided velocity components along the east–west and north–south directions, separate interpolations were performed for each, and the results were subsequently converted to wind speed and direction. The ERA5, CFSv2, and HRDPS wind datasets have identical time intervals, corresponding to exact hours. In contrast, the NARR dataset is provided at 3-hour intervals (i.e., 00:00, 03:00, 06:00, ..., 21:00). Although the wind observation times were approximately one hour apart, they did not align exactly with the hour marks, e.g., 00:00, 01:00. To calculate the metrics described in Section 2.5, the wind speeds and directions from wind datasets were temporally interpolated to match the observation times at each site. For wind speed, simple linear interpolation was applied. In contrast, handling wind direction required special attention due to the circular nature of angular data, e.g., 0° and 360° represent the same direction. Direct linear interpolation or averaging, as used in the calculation of metrics described in Section 2.5, can produce incorrect results. For instance, the arithmetic average of 10° and 350° is 180°, corresponding to a wind direction from south to north, even though both original directions are close to the direction from north to south. To address this issue, the method described by Berens (2009) was adopted. Angular values were

185 first transformed into unit vectors using their sine and cosine components. Linear interpolation or averaging was then applied separately to each component, and the resulting vectors were converted back into angles using the four-quadrant inverse tangent function.

## 2.4 Extrapolating Wind Speed

Wind measurements from marine buoys were taken at a height of 5 m above the sea surface, while data from the 4 wind datasets
were taken at 10 m. To compare data at the same height, the empirical power-law relationship was adopted to extrapolate wind speed from 5 m to 10 m, as below.

$$\frac{U_2}{U_1} = \left(\frac{z_2}{z_1}\right)^{\alpha}, \tag{1}$$

where $U_1$ and $z_1$ are the wind speed and the height of the observation data, respectively; $z_2$ is the standard height used in the wind datasets; $U_2$ is the extrapolated wind speed at height $z_2$. The shear exponent $\alpha$ in (1) depends on both surface roughness
and atmospheric stability, and varies at different heights and over time (Emeis, 2018; Gualtieri, 2016; Jung and Schindler, 2021; Shu et al., 2016). Despite these known variations, a constant $\alpha$ is often used in practice due to limited availability of *in situ* stability measurements. In this study, since the buoy data are available at only one level, when extrapolating wind speed from 5 m to 10 m height, the $\alpha$ value was chosen to be 0.14 following the IEC 61400-3-1:2019 standard (IEC, 2019).

For wind datasets, the situation is different because wind speeds are available at multiple heights. For example, in the case
of ERA5, which provides wind speeds at both 10 m and 100 m, the shear exponent $\alpha$ can be estimated directly as:

$$\alpha = \frac{\ln(U_{100}/U_{10})}{\ln(100/10)}, \tag{2}$$

where $U_{100}$ and $U_{10}$ are the wind speeds at 100 m and 10 m above surface, respectively. In the wind farm simulations presented in Section 4, wind speeds at the turbine hub height were extrapolated from ERA5 wind speeds at 10 m or 100 m using the spatially and temporally varying shear exponent derived from (2). This dynamic extrapolation can reduce uncertainties associated
with assuming a constant shear exponent, thus improving the accuracy of wind speed estimates at hub height and subsequent power production calculations (Wang et al., 2019).

## 2.5 Assessment Metrics

Four metrics to compare wind speed and wind direction observations (denoted as 'O' in the following equations) with the wind datasets (denoted as 'M' in the following equations) were selected: 1) Root Mean Square Error (RMSE); 2) bias; 3) Mean
Absolute Error (MAE); and 4) the coefficient of determination ($R^2$). The metrics were defined as follows.

RMSE is a measure of the magnitude of error between a wind dataset and the observed wind values (3). It provides an indication of how well wind dataset values align with observed wind data, with lower RMSE values indicating better dataset performance. RMSE is calculated as:

$$\text{RMSE} = \sqrt{\frac{1}{N}\sum_{i=1}^{N}(M_i - O_i)^2}, \tag{3}$$

where $N$ is the total number of data points.

Bias is a measure of the overall deviation between a wind dataset and the observed wind values (4). A positive or negative bias indicates that the dataset overestimates or underestimates the wind observations, respectively. Bias is calculated as:

$$\text{Bias} = \frac{1}{N} \sum_{i=1}^{N} (M_i - O_i). \tag{4}$$

MAE is a measure of the average absolute error between a wind dataset and the observed wind values (5). Given each error
influences MAE linearly, this metric is straightforward to interpret. MAE is calculated as:

$$\text{MAE} = \frac{1}{N} \sum_{i=1}^{N} |M_i - O_i|. \tag{5}$$

The coefficient of determination, $R^2$, is a measure of the degree to which the wind dataset matches the observed wind values (6). Its value ranges from 0, representing the worst prediction, to 1, representing a perfect match. $R^2$ is calculated as:

$$R^2 = 1 - \frac{\sum_{i=1}^{N} (M_i - O_i)^2}{\sum_{i=1}^{N} (O_i - \bar{O})^2}, \tag{6}$$

where $\bar{O} = \frac{1}{N} \sum_{i=1}^{N} O_i$ denotes the average value of the observations.

Metrics were calculated using wind speeds and wind directions at 10 m above the island surface or sea surface depending on the observation site. Because the study focused on evaluating a wind dataset for wind speed within a turbine's operating range, which is 3 m/s to 25 m/s at the hub height of a 150 m high turbine (Figure 2), the corresponding wind speed range at 10 m height is approximately 2 m/s to 17 m/s based on (1), assuming $\alpha = 0.14$. Therefore, all metrics were only calculated using
wind data during periods of wind speed that fell within a range of 2 m/s to 17 m/s. The percentage of time with 10 m wind speeds exceeding 17 m/s was estimated using the ERA5 dataset. These strong wind events occurred approximately 0.3% of the time at both nearshore sites, and between 1.8% and 2.5% at offshore sites.

## 2.6   Configuration of Power Production Model For Wind Farm Development Areas

PyWake is a Python package used to simulate wind farm flow fields. It integrates multiple wake deficit models and wake
interaction models (Pedersen et al., 2023). Validations of PyWake have demonstrated that its results agree well with those from Computational Fluid Dynamic models and observational data (PyWake development team, n.d.; Quick et al., 2024). The turbine model used for simulation in this study was the IEA 15 MW wind turbine (Gaertner et al., 2020). The IEA 15 MW wind turbine features a hub height of 150 m and the rotor diameter is 240 m.

The thrust coefficient ($C_t$) represents the portion of wind energy extracted by the rotor. At lower wind speeds, $C_t$ is high,
meaning a larger portion of the wind's energy is extracted for producing electricity, resulting in more pronounced wake effects (Figure 2). As wind speed increases beyond 10.6 m/s (equivalent to 7.2 m/s at 10 m height above the surface according to (1) with a constant shear exponent $\alpha$ of 0.14), $C_t$ decreases, reducing the portion of energy extracted, while the turbine reaches its rated power output of 15 MW. Consequently, wake effects become less pronounced.

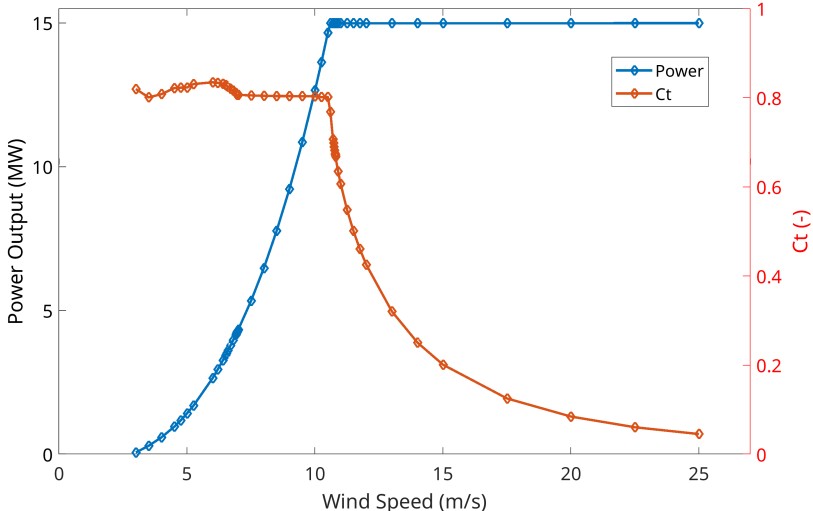

**Figure 2.** Power curve and thrust coefficient ($C_t$) versus wind speed used in this study. The turbine model adopted was an IEA 15 MW turbine (Gaertner et al., 2020).

In wind farms, turbines located in the interior experience lower wind speeds due to wake effects from upstream turbines. This
reduction in wind speed leads to a decrease in power production compared to an ideal scenario with no wake interference. To assess the impact of wakes on turbine performance, the wake efficiency metric was employed in this study, which quantifies how effectively a turbine generates power under wake-influenced conditions. Wake efficiency, $\eta$, defined as the ratio of a turbine's actual power output in the presence of wakes, denoted as $P_{\text{wake}}$, to its theoretical power output in an idealized scenario without wake effects, $P_{\text{ideal}}$, is expressed as:

$$\eta = \frac{P_{\text{wake}}}{P_{\text{ideal}}}. \tag{7}$$

The wake effect was simulated using the Gaussian-profile wake deficit model developed by Bastankhah and Porté-Agel (2014). This model is known for its accuracy in representing wake expansion and velocity deficits, particularly for modern, large-scale turbines. The Gaussian-profile wake model assumes self-similarity in the wind velocity deficit profile, which is valid in the far-wake region (Medici and Alfredsson, 2006). The extent of the near-wake region, which marks the onset of the
far-wake region, depends on turbine characteristics (e.g., $C_t$, tip speed ratio, and number of blades) as well as on the ambient turbulence intensity (Sørensen et al., 2014). Bastankhah and Porté-Agel (2014) validated the model using both wind tunnel measurements and Large Eddy Simulations, under incoming turbulence intensities ranging from about 0.05 to 0.13 at hub height (Table 1 in their study), and demonstrated good agreement at downstream distances greater than approximately 2–3 D from the turbine. In Section 4.1, where simulations were conducted with constant wind speed and direction, the minimum
turbine spacing was set to 2 D, corresponding to the lower bound of the model's validated range. Results obtained for spacings within 2–3 D should therefore be interpreted cautiously. However, in Section 4.3, where simulations were driven by time-varying wind speed and direction, the minimum turbine spacing was 3.4 D, which lies well within the far-wake regime.

In PyWake, ambient turbulence intensity is a parameter in such wake models, as it influences the rate of wake expansion and recovery. Turbulence intensity itself depends on factors such as atmospheric stability, wind speed, and measurement height. Due to the lack of direct measurements in the study region, a constant value of 0.1 was adopted, which is a typical value observed in offshore environments (Shu et al., 2016; Viselli et al., 2022; Türk and Emeis, 2010; Argyle et al., 2018; Gualtieri, 2015). At a given location, wind speed deficits were often influenced by wake effects from multiple upstream turbines. To account for combined wake effects in this study, the linear superposition sub-model in PyWake was used.

The simulations used hourly wind speed and wind direction sourced from the wind dataset of ERA5. Since winds on the Scotian Shelf are relatively consistent in space, a spatially averaged wind speed and wind direction were used in the simulation of each PFDA. This approach simplified simulation setup while also maintaining focus on temporal variability.

## 3 Wind Dataset Assessments

### 3.1 Assessment of Wind Speed

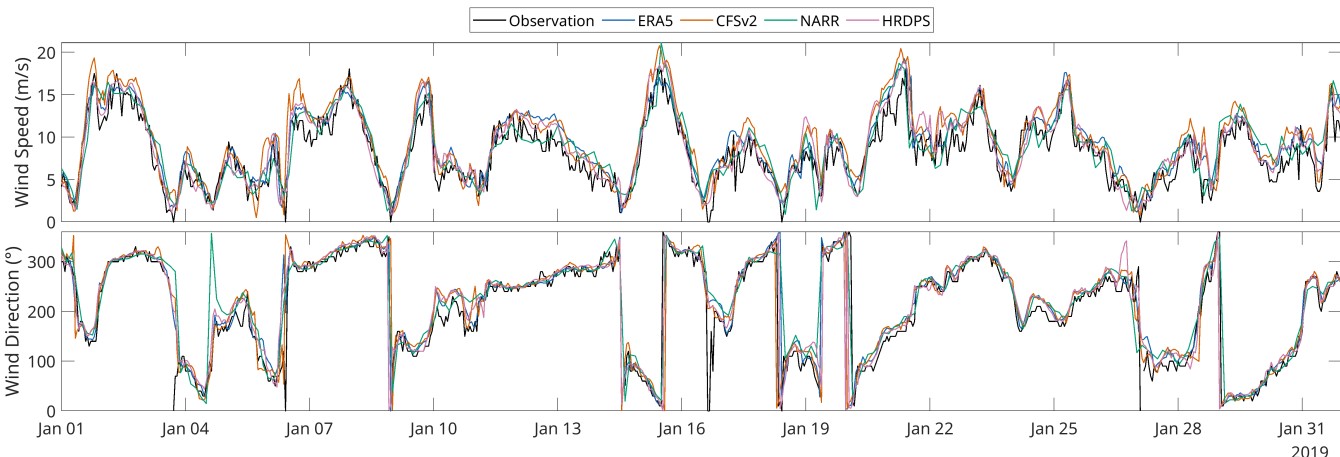

**Figure 3.** Time series of wind speed (upper panel) and wind direction (lower panel) at Site 3 at a 10 m height above surface for observation and the 4 wind datasets ERA5, CFSv2, NARR, and HRDPS. The comparison was for the month of January 2019.).

Time series of wind speed and wind direction at a height of 10 m above surface at Site 3, demonstrated general agreement between wind datasets and regional wind observations at the site (Figure 3). All datasets generally captured the variability and magnitude of the observed wind speed at Site 3. However, there were notable discrepancies between the datasets and observations during periods of higher observed wind speeds (e.g., January 6–8, 2019, and January 20–21, 2019), which illustrate that performance of the datasets does vary in time.

In terms of wind direction, the datasets exhibited good agreement with wind observations at Site 3 during most periods of moderate to high wind speeds. In contrast there were larger discrepancies in wind direction during periods of low wind speeds

(e.g., January 16–17, 2019). In general, the datasets performed well in capturing variations over longer timescales (days to weeks), although they did not consistently capture short-term fluctuations (on a daily scale).

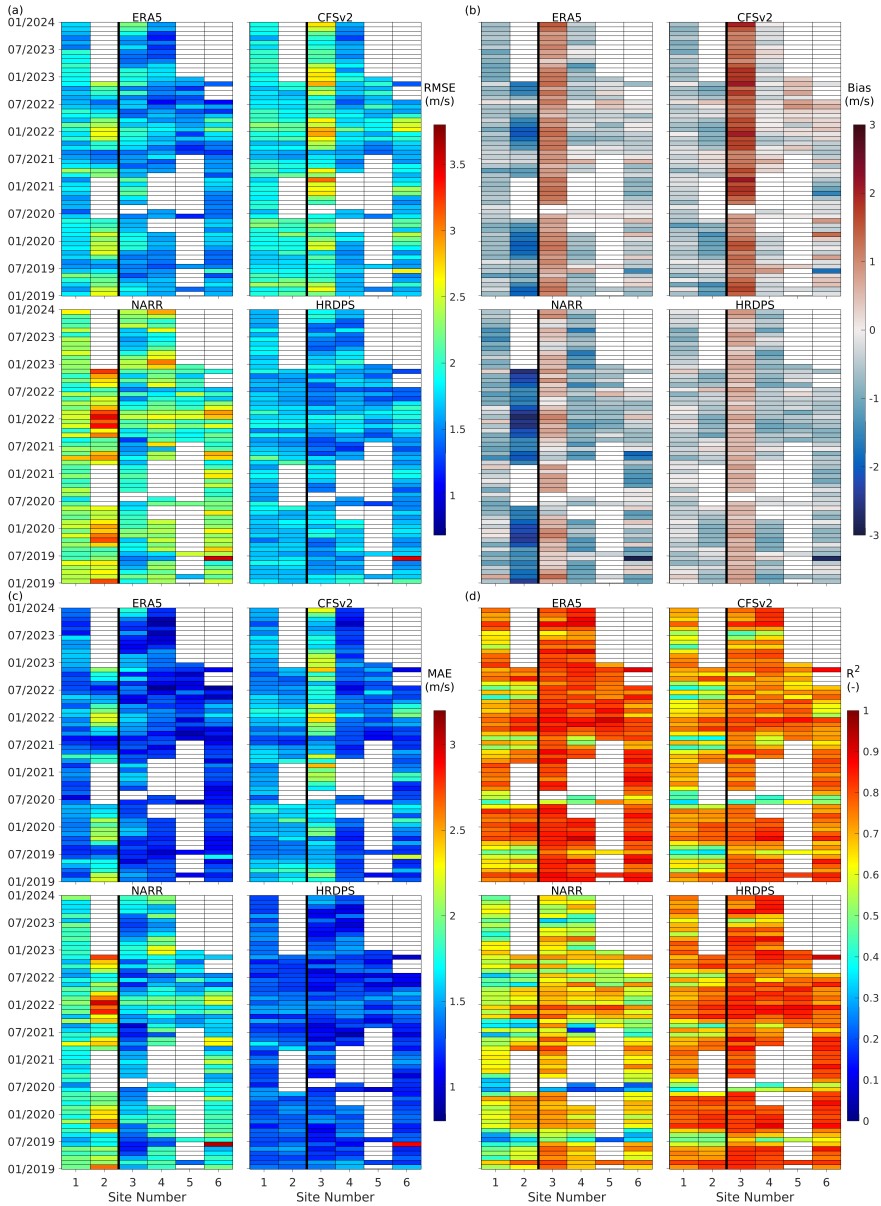

**Figure 4.** Pseudocolor plots displaying monthly (a) RMSE, (b) bias, (c) MAE, and (d) $R^2$ for wind speed for each dataset per wind observation site from January 1, 2019, to December 31, 2023. Sites 1 and 2 are representative of the nearshore (left of bold black line in each subplot) and Sites 3 - 6 are representative of the offshore (right of bold black line in each subplot) on the Scotian Shelf. The wind dataset is indicated at the top of each subplot. Blank areas (white pixels) indicate months and sites that had insufficient, valid observation records (considered to be less than 120 observation records in a month). These were considered to be non-valid for purposes of this study.

To quantitatively evaluate the wind datasets against the wind observations, the 4 metrics were used, as defined in equations (3)–(6) described above. The 4 evaluation metrics (RMSE, bias, MAE, and $R^2$) were calculated on a monthly basis over a 5-year period from January 1, 2019 to December 31, 2023, using data pairs of each wind dataset and the wind observations (Figure 4).

**ERA5:** ERA5 demonstrated strong performance, particularly at offshore sites. Comparing to other datasets, it exhibited the lowest 5-year averaged RMSE values (Figure 4 a) at Sites 4, 5, and 6, with values of 1.54 ± 0.18 m/s (mean ± standard deviation calculated from RMSE for all months per observation site), 1.45 ± 0.18 m/s, and 1.58 ± 0.23 m/s, respectively. At Site 3 (Sable Island), ERA5 showed the second-best RMSE (1.72 ± 0.24 m/s), slightly higher than HRDPS. At nearshore Sites 1 and 2, ERA5's RMSE was higher than HRDPS, at 1.76 ± 0.20 m/s and 2.08 ± 0.37 m/s, respectively. For bias (Figure 4 b), ERA5 generally showed a moderate tendency to underestimate wind speeds at the nearshore sites. The 5th–95th percentile ranges of monthly bias were -1.05 m/s to -0.17 m/s at Site 1 and -2.00 m/s to -0.30 m/s at Site 2, with only 30% and 11% of monthly bias values falling within ±0.5 m/s, respectively. At Site 3, ERA5 and all other datasets exhibited predominantly positive bias values across most months. This was likely attributable to the small physical size of the island (approximately 33.5 km east–west and less than 1.5 km north–south), which was not resolved by the numerical models underlying the wind datasets. As a result, surface roughness was likely underestimated in the models, leading to overestimated wind speeds at this site. At the marine buoy sites (Sites 4–6), ERA5 exhibited 5th–95th percentile bias ranges of -0.89 m/s to 0.12 m/s, -0.78 m/s to 0.44 m/s, and -0.85 m/s to 0.48 m/s, respectively. The corresponding percentages of monthly bias values within ±0.5 m/s were 49%, 70%, and 64%, ranking ERA5 second-best among the datasets at these offshore locations. In terms of MAE (Figure 4 c), ERA5 performed well offshore, showing the lowest 5-year averaged values of 1.20 ± 0.14 m/s, 1.13 ± 0.14 m/s, and 1.22 ± 0.17 m/s at Sites 4–6, compared to other datasets. At Site 3, ERA5 had the second-best MAE of 1.36 ± 0.19 m/s among the 4 datasets, following HRDPS. At nearshore sites, its MAE was slightly higher than HRDPS but comparable to CFSv2. ERA5 also had the highest $R^2$ values (Figure 4 d) at offshore sites, with values close to 0.80 at all 4 offshore locations. At nearshore sites, ERA5, with $R^2$ values of 0.69 and 0.72, ranked second behind HRDPS.

**HRDPS:** HRDPS consistently outperformed the other datasets at nearshore sites. It exhibited the lowest 5-year averaged RMSE values at Sites 1 and 2: 1.72 ± 0.15 m/s and 1.70 ± 0.13 m/s, respectively. Offshore, HRDPS performed best at Site 3 (1.61 ± 0.19 m/s RMSE) and ranked second at Sites 4–6. HRDPS also showed the smallest bias range at nearshore sites, with values mostly close to 0. The 5th–95th percentile ranges of monthly bias were -0.85 m/s to 0.15 m/s at Site 1 and -0.92 m/s to 0.17 m/s at Site 2, with 75% and 45% of monthly bias values falling within ±0.5 m/s, respectively. At the marine buoy Sites 4, 5, and 6, HRDPS was the third-best performing dataset. The 5th–95th percentile bias ranges at these sites were -1.10 m/s to -0.01 m/s, -0.95 m/s to 0.09 m/s, and -0.95 m/s to 0.00 m/s, respectively. The corresponding percentages of monthly bias values within ±0.5 m/s were 39%, 60%, and 43%. MAE values for HRDPS were the lowest among the 4 datasets at Sites 1 and 2 (1.33 ± 0.11 m/s and 1.34 ± 0.11 m/s). At Site 3, it exhibited the lowest MAE (1.24 ± 0.14 m/s) compared to all other datasets, and showed comparably strong performance at the remaining offshore sites. For $R^2$, HRDPS performed best at nearshore sites (0.70 ± 0.10 and 0.72 ± 0.11), and ranked second behind ERA5 offshore, with values around 0.77.

**CFSv2:** CFSv2 showed mixed performance. Its 5-year averaged RMSE was slightly higher than ERA5 at Site 1 (1.91 ± 0.17 m/s), but lower at Site 2 (1.92 ± 0.19 m/s). At offshore sites, its RMSE was higher than ERA5 and HRDPS. In terms of bias, CFSv2 ranked second-best at the nearshore sites. The 5th–95th percentile ranges of monthly bias were -0.96 m/s to 0.04 m/s at Site 1 and -1.29 m/s to 0.21 m/s at Site 2, with 58% and 37% of monthly bias values falling within ±0.5 m/s, respectively. The better performance of CFSv2 at Site 2 than ERA5 in terms of smaller RMSE and bias closer to 0 was likely attributed to its slightly finer horizontal resolution ($0.2°$ versus $0.25°$ for ERA5), which can better capture local wind gradients in the nearshore environment. Offshore, CFSv2 exhibited the best performance in terms of bias among all datasets. At the marine buoy Sites 4, 5, and 6, the 5th–95th percentile bias ranges were -0.50 m/s to 0.46 m/s, -0.44 m/s to 0.83 m/s, and -1.21 m/s to 0.80 m/s, respectively. The corresponding percentages of monthly bias values within ±0.5 m/s were 92%, 80%, and 75%, respectively—outperforming the other datasets at these offshore sites. Its MAE at nearshore sites was close to ERA5, and slightly higher at offshore sites. For $R^2$, CFSv2 generally ranked third across both nearshore and offshore sites, with values lower than ERA5 and HRDPS, but higher than NARR.

**NARR:** Among the 4 datasets, NARR had the weakest overall performance. It exhibited the highest 5-year averaged RMSE at all sites: 2.27 ± 0.21 m/s and 2.69 ± 0.38 m/s at Sites 1 and 2, and differences exceeding 0.6 m/s compared to ERA5 at Sites 4–6. Bias values for NARR showed the widest ranges. At nearshore sites, bias was consistently negative, especially at Site 2. Offshore, NARR exhibited the widest bias ranges among the 4 datasets. The 5th–95th percentile ranges of monthly bias were -1.56 m/s to -0.36 m/s at Site 4, -1.04 m/s to 0.29 m/s at Site 5, and -1.73 m/s to 0.32 m/s at Site 6. The corresponding percentages of monthly bias values falling within ±0.5 m/s were 14%, 35%, and 50%, respectively. MAE values from NARR were the highest among the 4 datasets across all sites. Last, NARR exhibited the lowest $R^2$ values among the 4 datasets at nearshore sites. At offshore locations, $R^2$ values for NARR were below 0.60.

Seasonal variations were evident across all 4 evaluation metrics. For RMSE and MAE, values generally increased during the winter months, when wind speeds were higher, and decreased in summer, when wind speeds were lower, at both nearshore and offshore sites. However, this seasonal pattern was primarily due to the higher wind speed magnitudes in winter compared to summer. When using normalized RMSE (i.e., RMSE divided by the mean observed wind speed), errors were actually smaller in winter and larger in summer. Bias also varied seasonally: nearshore sites tended to show more positive bias in fall and more negative bias in spring, whereas offshore sites exhibited the opposite trend. Seasonal variation in $R^2$ was most pronounced at nearshore sites, with values typically decreasing in spring and summer and increasing in fall and winter.

While the performance metrics exhibited seasonal and site specific variability, some consistent patterns exist when grouping the sites by location. At the nearshore sites, HRDPS consistently outperformed the other datasets across all 4 metrics. In contrast, offshore site performance varied slightly depending on the metric. ERA5 generally performed best for RMSE, MAE, and $R^2$, showing the greatest number of months with the best values. For bias, although CFSv2 yielded the best overall performance at most offshore sites, ERA5 still showed strong results, ranking as the second-best performer.

To further assess nearshore versus offshore Site groups, all observed wind speed data over the 5-year period was aggregated per group (i.e., nearshore and offshore). Each metric was subsequently calculated using the aggregated data to yield a 5-year averaged value per dataset.

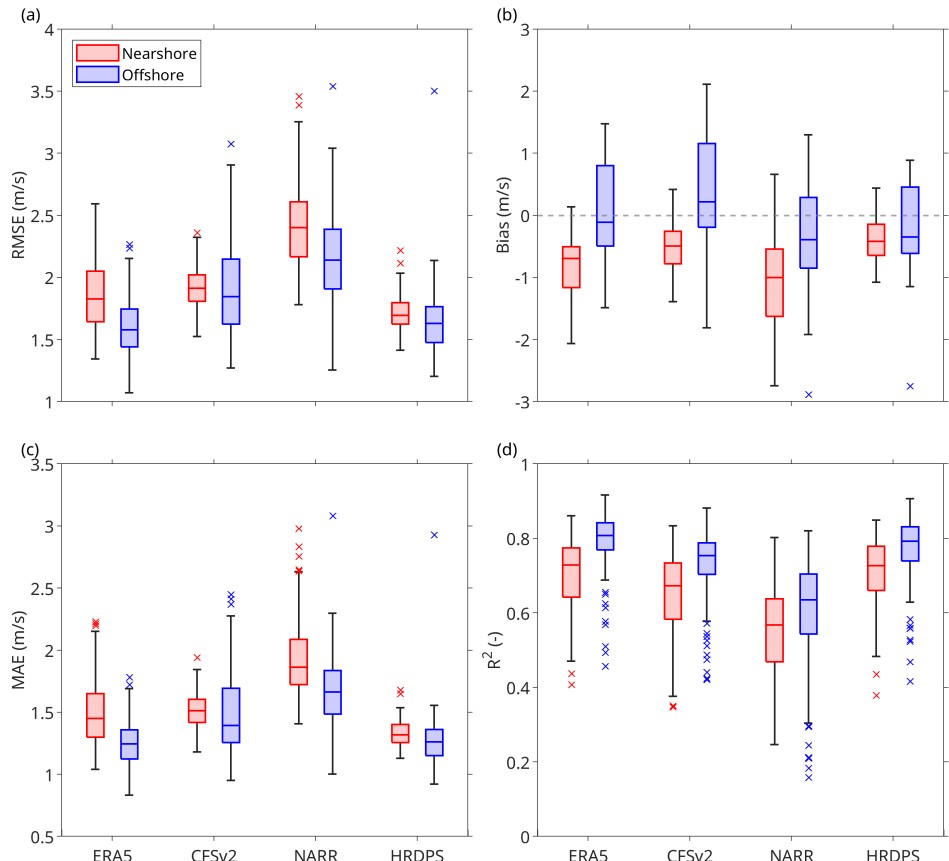

**Figure 5.** Box charts summarizing the monthly values of 4 wind speed evaluation metrics, as shown in Figure 4, of (a) RMSE, (b) bias, (c) MAE, and (d) $R^2$ for the 4 wind datasets of ERA5, CFSv2, NARR, and HRDPS. Sites are categorized into (red) nearshore and (blue) offshore groups. Each box spans the first and the third quartiles of the data, with the horizontal line inside each box indicating the median value. The whiskers extending from the box represent the minimum and maximum values that are within the 1.5 times the interquartile range (IQR). The individual markers represent the outliers, defined as values exceeding 1.5 times the IQR.

Performance was found to be higher for wind speed in the offshore site group, as indicated by lower median absolute values of RMSE, bias, and MAE, and higher $R^2$, compared to the nearshore site group (Figure 5). The lower performance of the datasets at nearshore sites was likely due to a more complex dynamic environment, where land-sea interactions introduce additional challenges for modeling. However, the wider spread of RMSE, bias, and MAE for offshore sites, with the exception of MAE for ERA5, suggested that dataset performance exhibited greater variability offshore (Figure 5 a–c). In contrast, $R^2$ showed a narrower interquartile range (IQR) offshore than nearshore, indicating more consistent correlations between observed and modeled wind speeds in offshore environments (Figure 5 d).

Among the 4 datasets, HRDPS and ERA5 consistently ranked as the top two performers, each achieving either the best or second-best values across most metrics for both nearshore and offshore site groups. For the nearshore site group, HRDPS

emerged as the top-performing dataset, achieving the best mean values for all metrics (Table 2) and the best median values for RMSE, bias, and MAE (Figure 5). While ERA5 held the highest median $R^2$, HRDPS closely followed with the second-best median value (Figure 5 d). Additionally, HRDPS exhibited the narrowest IQRs for all 4 metrics, which suggested greater consistency in performance compared to the other datasets (Figure 5). ERA5 ranked second, with the second-best median and mean values for RMSE and MAE (Figure 5 a and c; Table 2), as well as the highest median and second-highest mean value for $R^2$ (Figure 5 d; Table 2).

**Table 2.** Mean values of the monthly metrics for wind speed over the 5-year period from January, 2019 to December, 2023. Sites were grouped into nearshore and offshore groups. Only wind speed data within the range of 2–17 m/s were considered. The best-performing dataset metric is highlighted in bold. (-) = no units, as a dimensionless metric.

| Metric | ERA5 | CFSv2 | NARR | HRDPS |
|---|---|---|---|---|
| **Nearshore** | | | | |
| RMSE (m/s) | 1.89 | 1.92 | 2.43 | **1.72** |
| Bias (m/s) | -0.81 | -0.49 | -1.06 | **-0.38** |
| MAE (m/s) | 1.49 | 1.51 | 1.92 | **1.33** |
| $R^2$ (-) | 0.73 | 0.69 | 0.58 | **0.75** |
| **Offshore** | | | | |
| RMSE (m/s) | **1.62** | 1.98 | 2.15 | 1.64 |
| Bias (m/s) | 0.15 | 0.50 | -0.27 | **-0.13** |
| MAE (m/s) | **1.26** | 1.52 | 1.65 | 1.26 |
| $R^2$ (-) | **0.80** | 0.74 | 0.66 | 0.79 |

For the offshore site group, ERA5 and HRDPS exhibited similarly strong performance, with closely matched median and mean values across all metrics. ERA5 achieved the best median values for all 4 metrics, while HRDPS ranked second for RMSE, MAE, and $R^2$ (Figure 5). In terms of mean values, ERA5 achieved the best values for RMSE, MAE, and $R^2$, and second-best value for bias, while HRDPS achieved the best mean bias and second-best values for the other three metrics (Table 2). Additionally, both ERA5 and HRDPS showed narrower IQRs for RMSE, MAE, and $R^2$ compared to CFSv2 and NARR, which suggested greater consistency in their offshore performance (Figure 5 a, c and d).

Domestic electricity consumption often fluctuates throughout the day and varies by season. Therefore, wind dataset evaluations should align with these timescales to accurately capture variability and better inform wind energy development. To achieve this, this study aggregated local hourly wind speed data over the 5-year period (January 1, 2019 to December 31, 2023) (Figure 6). Data recorded at the same local hour on different days within the same calendar month, across all 5 years and all 6 sites, were grouped together for analysis.

Based on RMSE, it was found that wind speed estimation error varied by hour of the day and by month (Figure 6 a). The RMSE exhibited clear seasonal variations for all 4 datasets, with lower values observed in the spring and summer months (i.e.,

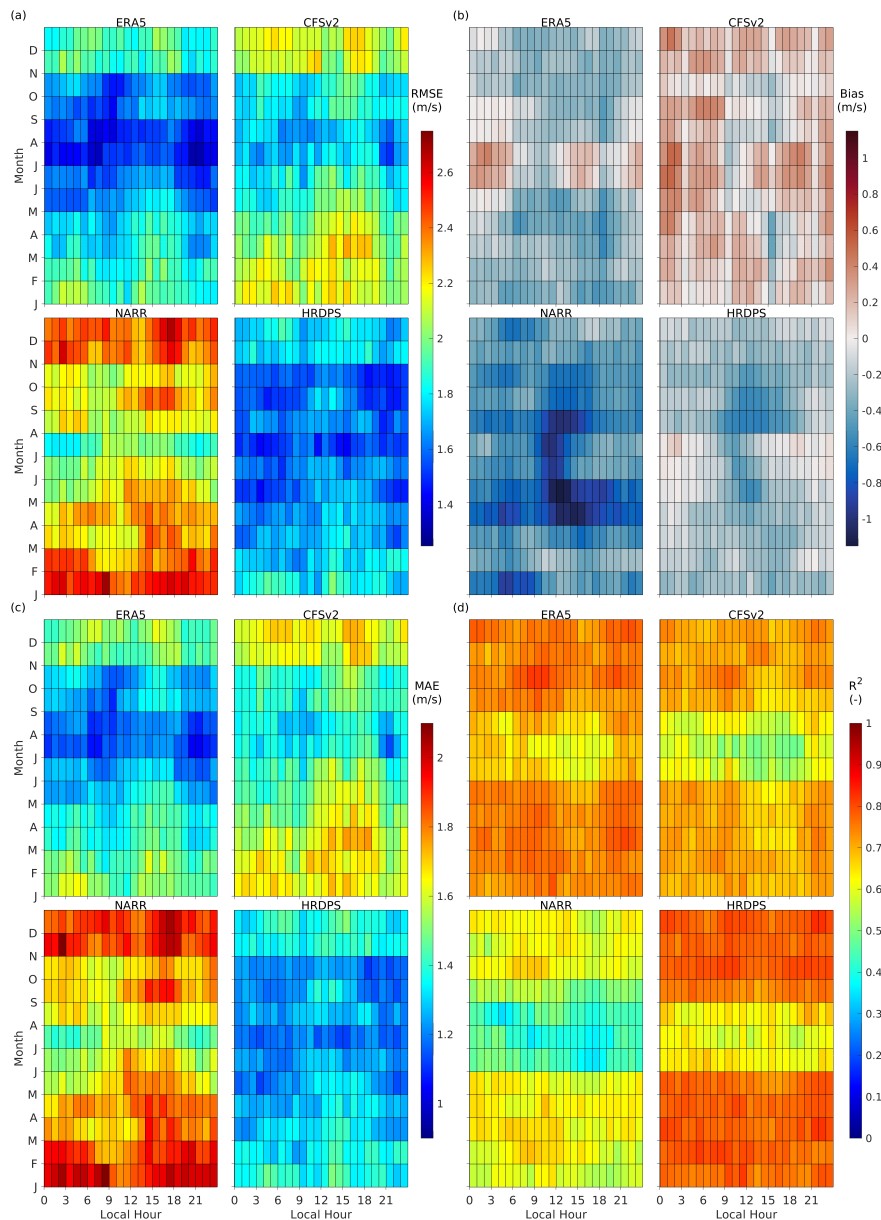

**Figure 6.** Pseudocolour plots displaying diurnal (local hour) and seasonal (monthly) wind speed variations in (a) RMSE, (b) bias, (c) MAE, and (d) $R^2$ for each dataset and grouped observation sites from January 1, 2019, to December 31, 2023. The x-axis represents local hours and the y-axis represents months aggregated over the 5 years. Each wind dataset is indicated at the top of each subplot. The wind dataset is indicated at the top of each subplot.

April to September) and higher values observed in the fall and winter months (i.e., October to March). In contrast, diurnal variation in RMSE appeared to differ between datasets. For ERA5, CFSv2, and NARR, RMSE values tended to peak between 15:00 and 18:00 in all months (except January for CFSv2 and February for NARR), relative to the dataset's mean RMSE for the corresponding month. Additionally, RMSE values for ERA5 and CFSv2 were generally lower between 20:00 and 22:00, except in September. In contrast, HRDPS displayed low RMSE values between 12:00 and 15:00 for all months except July. These results highlight the dataset-dependent nature of wind speed estimation errors and emphasize the influence of seasonal and diurnal cycles on dataset performance.

The HRDPS generally had the lowest RMSE values across most months, indicating better wind speed estimation for this metric. ERA5 showed lower RMSE than HRDPS in July and August, but had slightly higher RMSE from November to April. In other months, the RMSE values for ERA5 and HRDPS were comparable. CFSv2 performed within a mid-range, while NARR consistently exhibited the highest RMSE values, exhibiting poorer performance compared to HRDPS and ERA5 for this metric. Overall, winter months displayed the most significant errors in wind speed estimation; particularly, during certain local hours (e.g., between 15:00 and 18:00).

Seasonality in the NARR and HRDPS datasets was evident in the bias metric, which exhibited a higher negative bias during the spring and summer months for NARR and during fall months for HRDPS (Figure 6 b). These negative biases were indicative of significant underestimations of observed wind speeds during these seasons. The bias for ERA5 was generally low, but did exhibit positive values in June and July and negative values during other months. The CFSv2 exhibited an overall positive bias, but lacked a clear seasonal trend. Diurnal variations were notable in some months across all 4 datasets. For ERA5, the bias during summer months was negative in the morning and positive throughout the remainder of the day. For CFSv2, the bias shifted toward negative values at different times across the months: from 14:00 to 18:00 in March and April, 9:00 to 12:00 in May to July, and 9:00 to 17:00 in August to October. Similarly, NARR and HRDPS exhibited a more pronounced negative bias during midday hours in spring and summer months.

In general, NARR bias was consistently negative across all months and hours of the day, suggesting a systematic underestimation of wind speeds. This underestimation was particularly significant during the spring and summer months and midday (10:00 to 14:00). In contrast, ERA5 exhibited a modest bias overall, with slight overestimations observed in June and July and underestimations observed during the fall and winter months. Diurnal variations were also evident, with higher negative values observed during mid-day and lower negative (or even slightly positive values) observed in the early morning and late evening. HRDPS exhibited minimal bias across most months and hours, with relatively larger underestimations observed from August to October; particularly during mid-day hours. Last, CFSv2 generally exhibited positive bias, but lacked significant seasonal or diurnal variation, suggesting relatively stable deviations from observations across all time periods.

The MAE exhibited similar seasonal and diurnal patterns as those for RMSE, due to an inherent similarity between these two metrics (Figure 6 c). For $R^2$, ERA5 and HRDPS generally exhibited higher values compared to the other two datasets (Figure 6 d). Additionally, $R^2$ values were observed to be lower in the months from June to August.

## 3.2 Assessment of Wind Direction

While wind speed is the primary factor influencing electricity production in wind farms, wind direction also plays an important role due to its impact on turbine wakes (Gaumond et al., 2014; Stieren et al., 2021). Variation in wind direction for the same turbine layout can lead to differing wake interactions, which can affect downstream turbines and significantly influence total power output. In order to better understand the performance of the 4 datasets, this study compared the ability of the wind model datasets to replicate observed patterns in wind direction. The analysis was similar to that for wind speed described above, with the same performance evaluation metrics being used.

**ERA5:** ERA5 consistently exhibited strong performance across all sites. At nearshore locations, ERA5 achieved 5-year averaged RMSE (Figure 7 a) values of $23.38° \pm 4.41°$ and $25.30° \pm 5.04°$ at Sites 1 and 2, respectively. At offshore sites, ERA5 outperformed all other datasets with RMSE values of $20.35° \pm 4.59°$, $22.52° \pm 4.56°$, $40.26° \pm 20.36°$, and $29.27° \pm 11.85°$ at Sites 3–6, respectively. ERA5 also showed good agreement in MAE (Figure 7 c), ranking first at offshore sites and closely behind HRDPS at nearshore sites. For the $R^2$ metric (Figure 7 d), ERA5 achieved the highest values at all offshore sites (ranging from $0.81 \pm 0.16$ to $0.89 \pm 0.08$), and ranked second behind HRDPS at the nearshore sites, with $R^2$ values approximately 0.01 lower. For bias (Figure 7 b), ERA5 showed relatively low values at both nearshore and offshore locations, although, like the other datasets, it tended to shift wind direction clockwise at Site 3.

**HRDPS:** HRDPS performed similarly to ERA5 at nearshore sites, with 5-year averaged RMSE values just 0.3° higher than ERA5. At offshore sites, HRDPS ranked second, with RMSE values approximately 2°–3° higher than ERA5. The MAE values for HRDPS were nearly identical to those of ERA5 at nearshore sites and remained slightly higher offshore. HRDPS achieved the highest $R^2$ values among all datasets at nearshore locations: $0.81 \pm 0.08$ at Site 1 and $0.77 \pm 0.13$ at Site 2. At offshore sites, it ranked second, with $R^2$ values about 0.01–0.03 lower than ERA5. In terms of bias, HRDPS showed behavior similar to the other datasets, with generally low values except at Site 3, where it also exhibited a clockwise shift in wind direction.

**CFSv2:** CFSv2 exhibited moderate performance across the metrics. Its 5-year averaged RMSE values were approximately 3°–4° higher than ERA5 and HRDPS at nearshore sites, and 2°–6° higher at offshore sites. MAE values were also consistently higher by about 2°–4° compared to ERA5. For the $R^2$ metric, CFSv2 ranked third, with values about 0.04–0.09 lower than ERA5 across the offshore sites, and lower than both ERA5 and HRDPS at nearshore sites. The bias values for CFSv2 were similar to those of other datasets.

**NARR:** NARR exhibited the weakest performance among the 4 datasets. At both nearshore and offshore sites, its RMSE values were the highest, exceeding ERA5 and HRDPS by 7°–11°. Similarly, NARR's MAE values were the highest, approximately 3°–8° greater than those of ERA5. For the $R^2$ metric, NARR consistently ranked lowest, with values that were 0.12–0.18 lower than ERA5 at offshore sites, and lower than all other datasets at nearshore sites. Like the other datasets, NARR showed a clockwise bias at Site 3, and consistent bias behavior at the offshore buoys.

All datasets exhibited similar seasonal patterns in $R^2$ values, with generally lower values during spring and summer and higher values during fall and winter. This trend was evident in ERA5, HRDPS, and NARR.

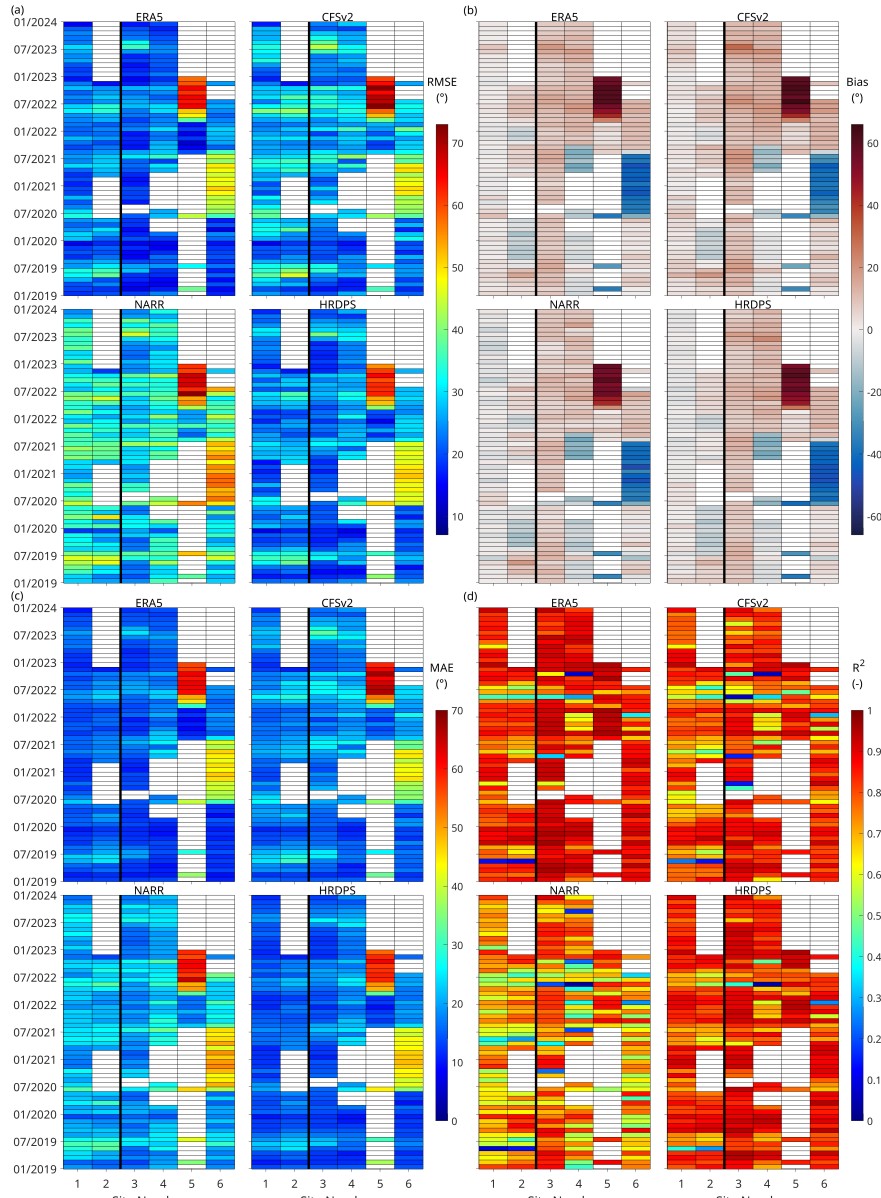

**Figure 7.** Pseudocolour plots displaying monthly (a) RMSE (b) bias (c) MAE and (d) $R^2$ for wind direction for each dataset per wind observation site from January 1, 2019, to December 31, 2023. Sites 1 and 2 are representative of the nearshore (left of bold black line in each subplot) and Sites 3-6 are representative of the offshore (right of bold black line in each subplot) on the Scotian Shelf. The wind dataset is indicated at the top of each subplot. Blank areas (white pixels) indicate months and sites that had insufficient, valid observation records (considered to be less than 120 observation records in a month). These were considered to be non-valid for purposes of this study.

At the buoy based offshore Sites 5 and 6, all datasets exhibited notably large bias values during specific periods, i.e., from April to December 2022 at Site 5 and from June 2020 to July 2022 at Site 6, which was likely due to systematic observational errors in the recorded wind direction. Outside of these periods, bias across offshore sites was generally consistent among the datasets.

Overall, ERA5 and HRDPS demonstrated the best performance in representing wind direction, with ERA5 showing superior accuracy offshore and HRDPS performing best nearshore. CFSv2 followed with moderate performance, while NARR consistently ranked lowest across all metrics and locations.

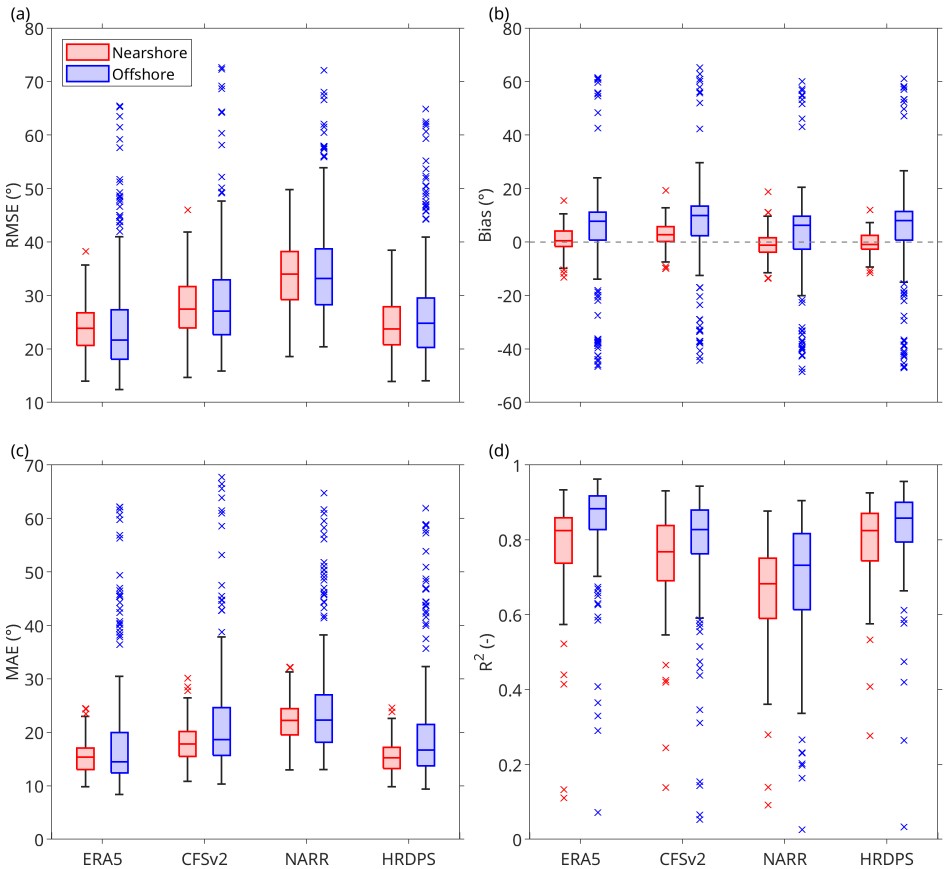

**Figure 8.** Box charts summarizing the monthly values of 4 wind direction evaluation metrics, as shown in Figure 7, of (a) RMSE, (b) bias, (c) MAE, and (d) $R^2$ for the 4 wind datasets of ERA5, CFSv2, NARR, and HRDPS. Sites are categorized into (red) nearshore and (blue) offshore groups. Each box spans the first and the third quartiles of the data, with the horizontal line inside each box indicating the median value. The whiskers extending from the box represent the minimum and maximum values that are within the 1.5 times the interquartile range (IQR). The individual markers represent the outliers, defined as values exceeding 1.5 times the IQR.

Metric results obtained using aggregated data for each site group for wind direction were presented in Figure 8 and Table 3. It is noted that for the offshore group, periods with suspected systematic observational errors at Sites 5 and 6 were not excluded

**Table 3.** Mean values of the monthly metrics for wind direction over the 5-year period from January, 2019, to December, 2023. Sites were grouped into nearshore and offshore groups. Only wind direction data recorded during periods with wind speed in the range of 2–17 m/s were considered. The best-performing dataset metric is highlighted in bold. (-) = no units, as a dimensionless metric.

| Metric | ERA5 | CFSv2 | NARR | HRDPS |
|---|---|---|---|---|
| **Nearshore** | | | | |
| RMSE (°) | **24.58** | 28.59 | 34.30 | 24.98 |
| Bias (°) | 1.07 | 2.70 | -0.81 | **-0.51** |
| MAE (°) | 15.49 | 18.19 | 22.33 | **15.45** |
| $R^2$ (-) | 0.81 | 0.77 | 0.69 | **0.82** |
| **Offshore** | | | | |
| RMSE (°) | **27.47** | 31.40 | 36.36 | 29.41 |
| Bias (°) | 5.54 | 8.07 | **4.00** | 5.18 |
| MAE (°) | **19.09** | 22.11 | 25.02 | 20.43 |
| $R^2$ (-) | **0.79** | 0.75 | 0.66 | 0.77 |

from the analysis. All data from months with at least 120 valid hourly records were retained to maintain a consistent screening criterion across all sites. As a result, the box plots in Figure 8 reflect the influence of these anomalies, as indicated by a greater number of outliers at offshore sites compared to nearshore sites.

    The median RMSE and MAE values were similar between the nearshore and offshore groups across all 4 datasets (Figure 8 a, c), while median bias values were smaller in the nearshore group (Figure 8 b) and median $R^2$ values were higher offshore

(Figure 8 d). The IQRs for bias and MAE were narrower in the nearshore group. In contrast, the IQRs for RMSE and $R^2$ were comparable between the two groups. Across different datasets, the IQRs were generally similar within each site group.

    Similar to the wind speed evaluation, HRDPS and ERA5 ranked as the top two performers for wind direction for both nearshore and offshore site groups. For the nearshore site group, HRDPS and ERA5 exhibited nearly identical best median values across all metrics (Figure 8). In terms of mean values, HRDPS outperformed the other datasets in bias, MAE, and $R^2$,

and achieved the second-best value for RMSE. ERA5 achieved the lowest RMSE and ranked second for MAE and $R^2$. The only exception was bias, where NARR achieved the second-best mean value instead of ERA5 (Table 3).

    For the offshore site group, ERA5 achieved the best median and mean values for RMSE, MAE, and $R^2$, while HRDPS ranked second for these three metrics in both median and mean values (Figure 8 a, c and d; Table 3). For bias, NARR achieved the best median and mean values, while ERA5 and HRDPS shared the second-best median value, and HRDPS achieved the

second-best mean value (Figure 8 b; Table 3).

## 4 Power Production Simulations for Wind Farm Development Areas

The preceding analysis showed that ERA5 and HRDPS performed comparably well and outperformed the other two datasets on the Scotian Shelf. Moreover, because ERA5 provides wind velocity data at 100 m height, allowing the calculation of time-varying values of the exponent $\alpha$ in (1), ERA5 was selected for the power production simulations within the 6 PFDAs presented in this section.

### 4.1 Impact of Turbine Spacing on Wind Farm Performance

Optimizing offshore wind farm layout is a complex process influenced by seabed conditions, environmental impacts, construction feasibility, and wind resource distribution (Hou et al., 2019; Rezaei et al., 2023). The development of offshore wind energy on the Scotian Shelf requires careful consideration of site selection and turbine layout, which currently remain undefined as they depend in part on continued site assessments. To support this process, an idealized scenario was applied in which turbines were uniformly placed within the PFDAs, providing a simplified framework for evaluating potential energy production. Wake effects, caused by turbulence behind turbines, reduce wind speed at downstream turbines and therefore decreases their efficiency. As such, the trade-off between maximizing turbine density and minimizing wake effect wind speed losses is a key consideration in wind farm design.

**Table 4.** Seasonal mean wind speed (WS) and direction (WD) at 10 m height across 6 offshore potential future development areas (PFDAs) during winter (December–February) and summer (June–August) obtained using ERA5 dataset. The parameters $x_m$ and $x_t$ represent the values of $L/D$, obtained from the piecewise function (see Section 4.2), that correspond to the maximum function value and the transition point between the two segments of the piecewise function, respectively. Refer to Figure 1 for the locations of PFDAs on the Scotian Shelf.

| PFDA | Winter | | | | Summer | | | |
|------|--------|--------|-------|-------|--------|--------|-------|-------|
| | WS (m/s) | WD (°) | $x_m$ | $x_t$ | WS (m/s) | WD (°) | $x_m$ | $x_t$ |
| Sydney Bight | 9.6 | 281.4 | 4.1 | 5.7 | 6.4 | 209.7 | 4.1 | 7.5 |
| Canso Bank | 9.4 | 286.3 | 3.6 | 5.6 | 6.4 | 219.7 | 3.9 | 6.8 |
| Eastern Shore | 9.3 | 290.2 | 3.4 | 6.0 | 6.3 | 222.4 | 3.4 | 8.9 |
| Middle Bank | 9.8 | 289.1 | 3.8 | 6.0 | 6.4 | 222.9 | 4.7 | 9.4 |
| Sable Island Bank | 9.7 | 289.4 | 4.5 | 6.8 | 6.2 | 223.1 | 5.9 | 9.8 |
| Emerald Bank | 9.7 | 290.3 | 4.6 | 6.9 | 6.2 | 235.4 | 5.7 | 10.3 |

To explore how turbine spacing affects the potential total electricity production within the PFDAs, simulations were carried out using PyWake for two seasonal scenarios: winter (December to February) and summer (June to August). To focus on the relationship between total power production and turbine spacing, while reducing computational costs, constant wind speeds and wind directions derived from the ERA5 dataset were used for each seasonal scenario. However, it is important to note that because wind turbine power output is proportional to the cube of wind speed, the simulation result using seasonal mean wind

speed does not accurately represent the seasonal mean energy yield. In Section 4.3, simulations for each PFDA were performed using time-varying wind speed and direction data to provide more realistic estimates of energy production.

For each PFDA, the spatial and seasonal mean wind speed and direction were calculated by averaging wind speed and direction across all ERA5 grid points within the PFDA boundaries and over all times during the respective season across the 5-year period from 2019 to 2023. This approach was applied to both 10 m and 100 m wind speeds. Using the resulting spatial and seasonal mean values at these two heights, the shear exponent $\alpha$ was estimated with (2). Wind speed at the turbine hub height of 150 m was then extrapolated with (1) using the estimated $\alpha$.

Winds over the Scotian Shelf exhibited distinct seasonal patterns. In winter, the seasonal mean wind speed at 10 m height above surface ranged from 9.3 m/s to 9.8 m/s, with wind directions ranging from 281.4° to 290.3° across the 6 PFDAs. In summer, wind speeds ranged from 6.2 m/s to 6.4 m/s, with wind directions ranging between 209.7° and 235.4° (Table 4).

The spacing between neighboring turbines was normalized by the rotor diameter as $L/D$, where $L$ was the distance between two adjacent turbines and $D$ was the rotor diameter. In simulations, $L/D$ was varied incrementally from 2 to 12 in steps of 0.2 to comprehensively assess any impact on energy production.

For each spacing configuration, the total power outputs, $P_{\text{total}}$, of each PFDA were obtained from simulations. The power generated per turbine, $P_{\text{unit}}$, was then obtained by dividing $P_{\text{total}}$ by the total number of turbines within the corresponding PFDA (Figure 9).

For all of the PFDAs simulated in this study, total electricity production was greatest in winter compared to summer (Figure 9), attributed to the stronger seasonal mean wind speeds observed in winter. From the curves of $P_{\text{unit}}$, it can be observed that wake efficiency increased with increasing $L/D$ for the 6 PFDAs in both seasons. Because the wind speeds were constant in these simulations, the increase in $P_{\text{unit}}$ with $L/D$ indicated that energy losses due to wakes were reduced. In winter, turbines reached their rated capacity of 15 MW when $L/D$ exceeded approximately 7 in most PFDAs. In summer, $P_{\text{unit}}$ increased more gradually with turbine spacing, following an asymptotic trend. A larger $L/D$ value was required for turbines to achieve a higher wake efficiency. Specifically, to achieve a wake efficiency of 0.8, as defined in (7), the minimum $L/D$ values ranged from 6 to 10 for the 4 smaller PFDAs (Sydney Bight, Canso Bank, Eastern Shore and Middle Bank). For the two larger PFDAs (Sable Island Bank and Emerald Bank), the wake efficiency only reached a maximum value of 0.7 at $L/D = 12$.

Simulated flow maps from the Middle Bank PFDA illustrated wind speed and wake effects during the winter and summer months (Figure 10). In this example, seasonal mean wind speed and wind direction at 10 m height above surface were 9.8 m/s and 289.1°, respectively, in winter, and 6.4 m/s and 222.9°, respectively, in summer. Wind speeds were extrapolated to a hub height of 150 m above surface using (1) and turbine spacing set to 3.8 D. Winds within the PFDA were stronger and hence produced higher energy in the winter (Figure 10 b) than in the summer (Figure 10 a).

The flow maps revealed some key features, such as areas of significant wind speed reduction directly behind turbines (represented by the dark shaded regions) and areas where wakes began to dissipate and recover (represented by the lighter tails extending downstream) (Figure 10 a and b). The interaction of wakes from multiple turbines was notable in the interior of the Middle Bank PFDA, where overlapping wake regions created more complex wind speed deficits. This clustering of wake effects appeared to cause downstream turbines to experience more pronounced reductions in wind speed due to the cumulative

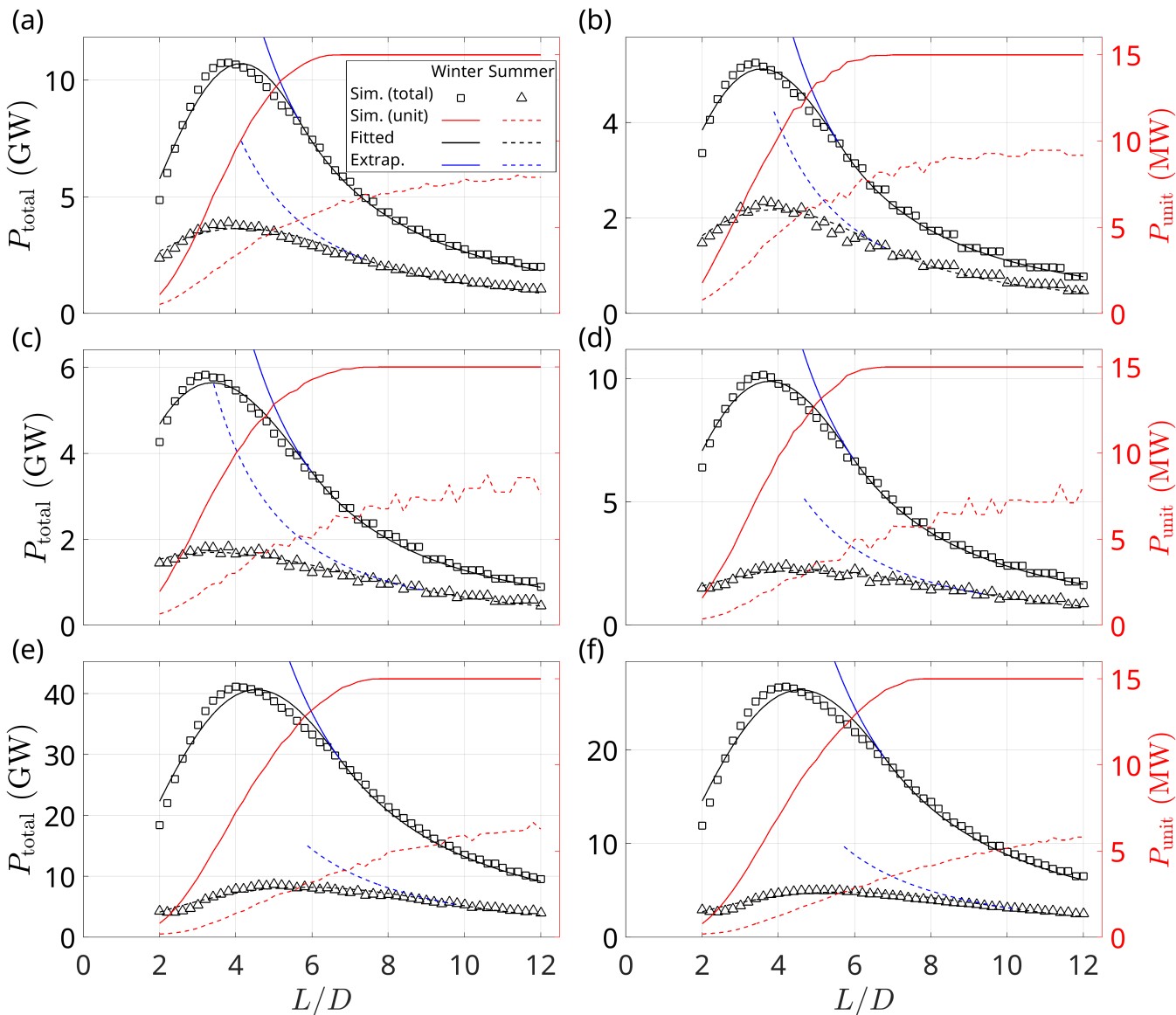

**Figure 9.** Relationships between total power production, $P_{total}$ (Y-axis on the left), and normalized turbine spacings ($L/D$) for the 6 potential future development areas (PFDAs) of (a) Sydney Bight (b) Canso Bank, (c) Eastern Shore, (d) Middle Bank, (e) Sable Island Bank, and (f) Emerald Bank. Rectangular markers ($\square$) and triangular markers ($\triangle$) represent simulation results for winter and summer, respectively. The black solid and dashed curves are fitted piecewise functions for winter and summer, respectively. The Y-axis on the right shows power production per turbine, $P_{unit}$, from simulations for winter (solid red lines) and summer (dashed red lines). Last, the blue curves show extrapolation of the inverse square part of the piecewise function for $L/D < x_t$, which is described further in Section 4.2 below. Refer to Figure 1 for the locations of PFDAs on the Scotian Shelf.

impact of upstream wakes. When turbine spacing was set to 9.6 D, and wind parameters for winter were assumed, impact of wakes caused by upstream turbines on downstream turbines appeared negligible (Figure 10 c and d).

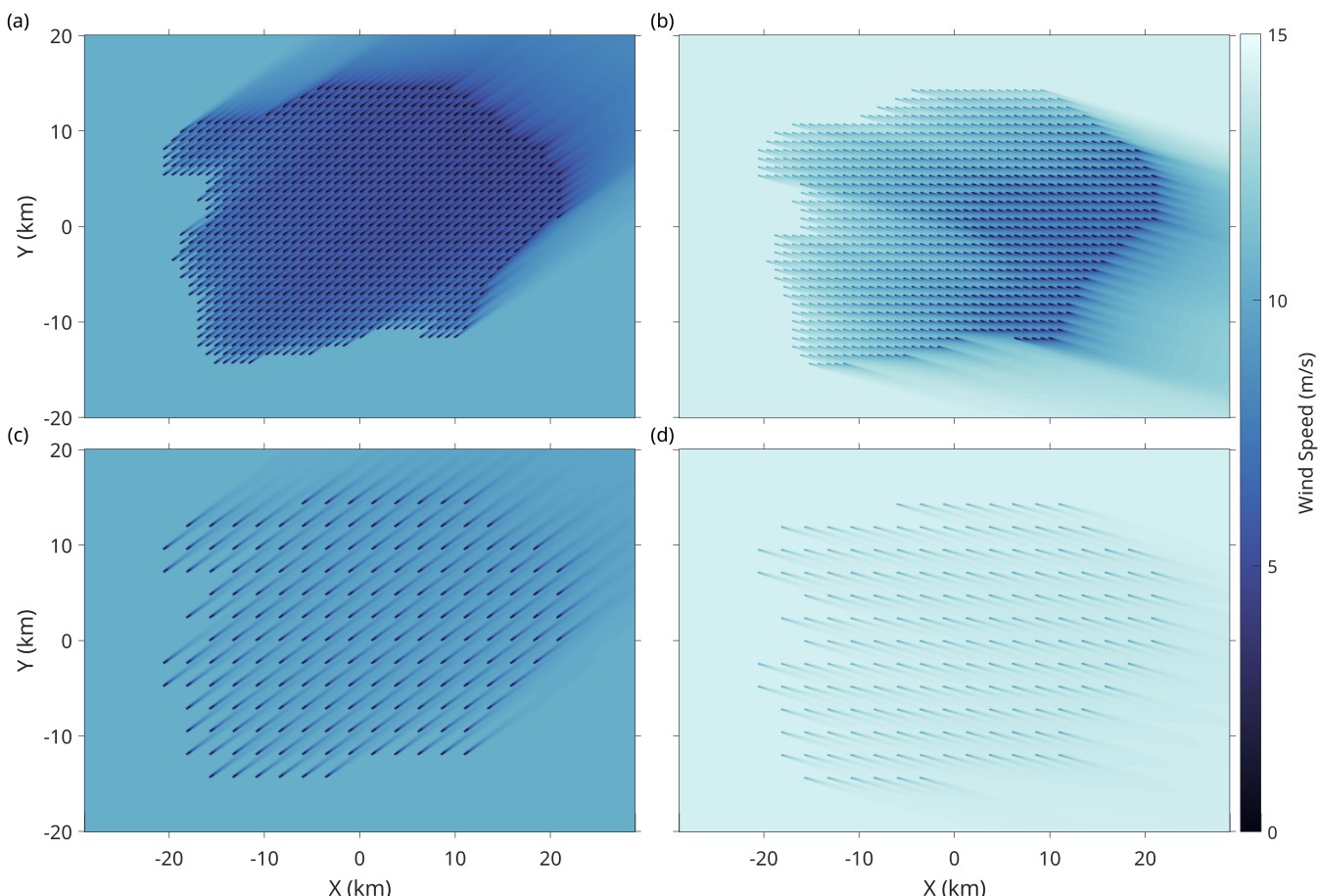

**Figure 10.** Flow maps of wind speed and wake effects simulated using PyWake for the Middle Bank potential future development area (PFDA) during (a, c) summer and (b, d) winter. The assumed turbine spacings were (a, b) 3.8 and (c, d) 9.6 times the rotor diameter, approximately 0.9 km and 2.3 km, respectively. Wind data used in the simulation corresponded to the seasonal mean wind speed and wind direction at 10 m height above surface using the ERA5 dataset. These were 9.8 m/s and 289.1°, respectively, for winter and 6.4 m/s and 222.9°, respectively, for summer. The input wind data was extrapolated to an assumed turbine hub height of 150 m above surface, with results presented in the figure being at hub height. Refer to Figure 1 for the locations of the Middle Bank PFDA on the Scotian Shelf.

Wake-turbine interactions reduced power production for turbines differently depending on turbine locations and wind directions. Figure 11 (a) presented the spatial distribution of seasonally-averaged power production of individual wind turbines for the 6 PFDAs.

Simulations were conducted using hourly wind data from the ERA5 dataset. The hourly power output of each turbine was averaged over the respective 5-year winter and summer periods from 2019 to 2023. Two turbine spacing scenarios were

considered. The first was a dense layout with normalized spacings, $L/D$, ranging from 3.4 to 5.2 across the 6 PFDAs (Table 5). The corresponding spacing in each PFDA represented the average where power production peaked in two seasons under a simplified model with constant wind speed and direction. The second scenario used a uniform spacing of $L/D = 9.6$ for all PFDAs. This spacing represented a scenario where the wake effects were minimal and allowed for approximately 1000 turbines in Sable Island Bank PFDA (Table 6), aligning with estimates from Nicholson (2023). Diagrams illustrating turbine layouts for 6 PFDAs on the Scotian Shelf were provided in Figure A2.

Wind conditions in the two seasons on the Scotian Shelf were illustrated with the wind rose diagrams using the examples of the Sydney Bight and Middle Bank PFDAs (Figure 11 b). In winter, wind speeds were generally higher, predominantly blowing from the northwest, with frequent occurrences of speeds exceeding 12 m/s. In summer, the winds were weaker, primarily blowing from the southwest, with most speeds falling below 12 m/s.

The turbine spacing and seasonal wind variations significantly influenced the power production of individual turbines at different locations. Under the large spacing scenario ($L/D = 9.6$), wake effects were minimal in both seasons, as evidenced by the relatively uniform power production among turbines within each PFDA (bottom panels in Figure 11 a). In contrast, in the smaller spacing scenario, wake effects became more pronounced and reduced the efficiency of individual turbines (top panels in Figure 11 a).

Power production of individual turbines varied within each PFDA, depending on turbine placement and dominant wind direction. Reviewing the Middle Bank PFDA as an example, in winter, when the prevailing winds were from the northwest, turbines located near the northern and western edges exhibited the highest power production (top-left panel in Figure 11 a). These turbines experienced less wake interference as they were positioned upstream relative to the dominant wind direction. In contrast, during summer, when winds predominantly came from the southwest, the highest power production was observed for turbines situated along the western and southern edges of the PFDA (top-right panel in Figure 11 a). For turbines located further downstream in the interior or at the leeward edges of the PFDA, power production was significantly reduced due to wake effects.

## 4.2 Simulation Results and Fitting

The simulated total power production, $P_{\text{total}}$, for the 6 PFDAs during two seasons exhibited a characteristic pattern consisting of two distinct regimes (Figure 9).

In the first regime, where turbine spacing was large and wake losses were negligible, the per-turbine power production approached its theoretical limit depending on the background wind speed (see: Figure 2). Under these conditions, $P_{\text{total}}$ for a given PFDA, using the same turbine model, scaled proportionally with the total number of installed turbines. Since the area of each PFDA was fixed, the total number of turbines followed an inverse square relationship with turbine spacing. Consequently, $P_{\text{total}}$ exhibited an inverse square relationship with $L/D$.

In the second regime, where turbine spacing was smaller and wake effects became significant, the simulation results exhibited a non-monotonic trend in $P_{\text{total}}$. Initially, as $L/D$ decreased, $P_{\text{total}}$ increased due to the increased number of turbines. However,

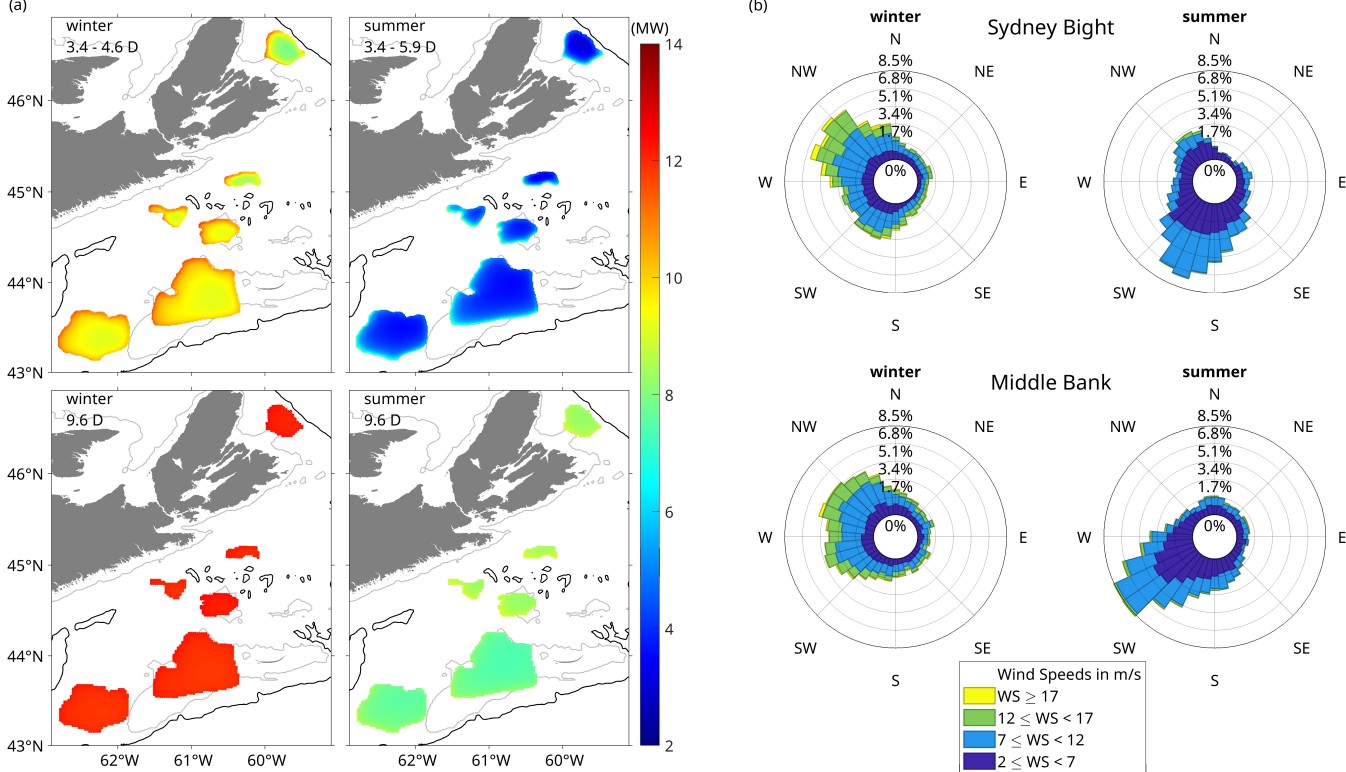

**Figure 11.** (a) Spatial distribution of wind turbine power production for two turbine layouts and two seasons for all potential future development areas (PFDAs) on the Scotian Shelf. Color shading shows the mean power production for each PFDA, averaged across (left panels) winter and (right panel) summers from 2019 to 2023. Normalized turbine spacings, $L/D$, were set to $x_m$, as listed in Table 5, for different PFDAs for top panels, and to 9.6 for bottom panels. (b) Wind rose diagrams for (top panels) the Sydney Bight PFDA and (bottom panels) Middle Bank PFDA, based on spatially averaged ERA5 data from 2019 to 2023. Refer to Figure 1 for the locations of PFDAs on the Scotian Shelf.

beyond a critical threshold, further reduction in $L/D$ led to a sharp decline in $P_{\text{total}}$ due to intensified wake effects. This bell-shaped pattern was similar to the Weibull-like function.

Building on the two-regime behaviours, the simulation results were modeled using an empirical piecewise function for $P_{\text{total}}$ as a function of $L/D$. This function captured the inverse square relationship at large $L/D$ and the Weibull-like behavior at small $L/D$. This piecewise function was formulated as follows:

$$f(x) = \begin{cases} a \cdot \frac{k}{\lambda} \left(\frac{x}{\lambda}\right)^{k-1} e^{-\left(\frac{x}{\lambda}\right)^k}, & \text{for } x < x_{\text{t}}, \\ \frac{c}{x^2}, & \text{for } x \geq x_{\text{t}}. \end{cases} \tag{8}$$

Here, $a$, $k$, and $\lambda$ were parameters of the Weibull-like function; $x_{\text{t}}$ was the critical transition point where the behavior transitions from the Weibull-like regime to the inverse square regime; and $c$ was the coefficient ensuring continuity at $x = x_{\text{t}}$.

To ensure a smooth transition between these two regimes at $x = x_{\text{t}}$, the following continuity conditions were applied:

– Value Continuity:

$$a \cdot e^{-\left(\frac{x_{\mathrm{t}}}{k}\right)^{b}} = \frac{c}{x_{\mathrm{t}}^2}. \tag{9}$$

– Derivative Continuity:

$$-a \cdot \frac{b}{k}\left(\frac{x_{\mathrm{t}}}{k}\right)^{b-1} e^{-\left(\frac{x_{\mathrm{t}}}{k}\right)^{b}} = -\frac{2c}{x_{\mathrm{t}}^3}. \tag{10}$$

From these conditions, the transition point $x_{\mathrm{t}}$ and the coefficient $c$ were determined analytically as:

$$x_{\mathrm{t}} = \lambda \left(\frac{k+1}{k}\right)^{1/k}, \tag{11}$$

and

$$c = a \cdot \frac{k}{\lambda}\left(\frac{x_{\mathrm{t}}}{\lambda}\right)^{k-1} e^{-\left(\frac{x_{\mathrm{t}}}{\lambda}\right)^{k}} \cdot x_{\mathrm{t}}^2. \tag{12}$$

The maximum value of the function was located at:

$$x_{\mathrm{m}} = \lambda \left(\frac{k-1}{k}\right)^{1/k}. \tag{13}$$

Substituting this value of $x_{\mathrm{m}}$ into the first part of the piecewise function yields the maximum value:

$$f_{\mathrm{m}} = a \cdot \frac{k}{\lambda}\left(\frac{k-1}{k}\right)^{(k-1)/k} e^{-\left(\frac{k-1}{k}\right)}. \tag{14}$$

This value represents the maximum total power production predicted by the Weibull-like part of the piecewise function.

Parameters of $a$, $\lambda$, and $k$ were unknown, but were obtained through non-linear fitting. In this fitting process, the independent variable, $x$, represents the normalized turbine spacing, $L/D$.

The simulation results for 6 PFDAs in two seasons (Figure 9) were fitted using the piecewise function. The fitted functions were overlaid on the simulation data for comparison. From the fitted functions, the parameters $a$, $\lambda$, and $k$ were determined, allowing for the calculation of $x_{\mathrm{t}}$ and $x_{\mathrm{m}}$, which were presented in Table 4. The parameter $x_{\mathrm{m}}$ represents the normalized turbine spacing ($L/D$), at which total power production reaches its maximum for a given PFDA. This value ranged from 3.4 to 4.6 in winter and 3.4 to 5.9 in summer for the 6 PFDAs.

The parameter $x_{\mathrm{t}}$ defines the transition point at which wake effects become negligible for $L/D > x_{\mathrm{t}}$, with wake effects becoming significant for $L/D < x_{\mathrm{t}}$. In winter, $x_{\mathrm{t}}$ ranged from 5.6 to 6.9 for the 6 PFDAs. In summer, $x_{\mathrm{t}}$ was notably larger, ranging from 6.8 to 10.3.

The fitted piecewise function was closely aligned with the simulation results. The extrapolated curve for $L/D < x_{\mathrm{t}}$, based on the inverse square relationship, was shown in blue (Figure 9). The difference between the extrapolated curves and the fitted piecewise function illustrated power production losses due to wake effects. These losses were more pronounced in summer than in winter for most PFDAs.

### 4.3 Temporal Variations in Simulated Electricity Production

Total electricity production for the PFDAs on the Scotian Shelf was more accurately estimated by using time-dependent wind speeds and wind directions. After reviewing earlier assessments of wind datasets, the ERA5 dataset were selected for use in this study. Because spatial variation in wind within each PFDA domain was minimal, wind speeds and wind directions were averaged across each area. Before running simulations, wind speed at $10\,\mathrm{m}$ above surface was converted to a wind speed at turbine hub height of $150\,\mathrm{m}$ above surface. Two scenarios for turbine spacings ($L/D$) were then tested: 1) values ranging from 3.4 to 5.2 across the 6 PFDAs (see Table 5), which were obtained as the mean values of $x_\mathrm{m}$ in winter and summer (Table 4); and 2) a fixed turbine spacing of L/D = 9.6.

Uncertainty in power estimation associated with the RMSE between wind datasets and wind observations was also accounted for. Two synthetic wind speed time series were generated by adding or subtracting the RMSE from the $10\,\mathrm{m}$ wind speeds, then separate simulations were run using the data. The resulting power values represented the upper and lower bounds of the uncertainty range. For total power without wake effects, a single-turbine power curve was used (Figure 2), which was then multiplied by the number of turbines. Hourly time series of total power for the 6 PFDAs were obtained from the simulations, which were then time-averaged to create a monthly time series (Figure 12). Seasonal mean results in winter and summer were summarized in Table 5.

For the 6 PFDAs, total electricity production rates ranged from 4.4 to $2.9\,\mathrm{GW}$, for the winter and summer, respectively, at the Canso Bank PFDA to 28.8 to $17.5\,\mathrm{GW}$, for the winter and summer respectively, at the Sable Island Bank PFDA (Table 5). All 6 PFDAs exhibited clear seasonal cycles, with higher energy production observed during winter months (December to February) and lower energy production observed during summer months (June to August). For example, at the Middle Bank PFDA, the total power production observed in winter ($7.7 \pm 1.9\,\mathrm{GW}$) was approximately 60% higher than that observed in summer ($4.8 \pm 2.2\,\mathrm{GW}$).

When compared to the results from a 'No Wake' scenario, where total energy production depended only on wind speed and turbine number, the extent of energy loss due to wake effects was evident. For all 6 PFDAs, simulation results that accounted for wake effects consistently exhibited energy productions that fell below those of the 'No Wake' scenario (Figure 12). Further, wake-induced reductions in electricity production were higher in percentage terms during summer compared to winter. For example, at the Middle Bank PFDA total energy losses associated with wakes were approximately 22% in winter (December to February) compared to 40% in summer (June to August) (Table 5).

In a scenario where turbine spacing was set to $L/D = 9.6$, seasonal cycles (Figure 13) were consistent with those observed in the scenario with small turbine spacings (Figure 12). For example, at the Middle Bank PFDA the total power production observed in winter ($2.0 \pm 0.4\,\mathrm{GW}$, Table 6) was approximately 33% higher compared to summer ($1.5 \pm 0.5\,\mathrm{GW}$, Table 6).

The impact of wake effects across all 6 areas under the $L/D = 9.6$ turbine spacing scenario (Figure 13, Table 6) was significantly diminished when compared to the scenario with smaller turbine spacing (Figure 12, Table 5). In winter, the simulation results were nearly the same as, or slightly lower than, those from the 'No Wake' case. In summer, the results for

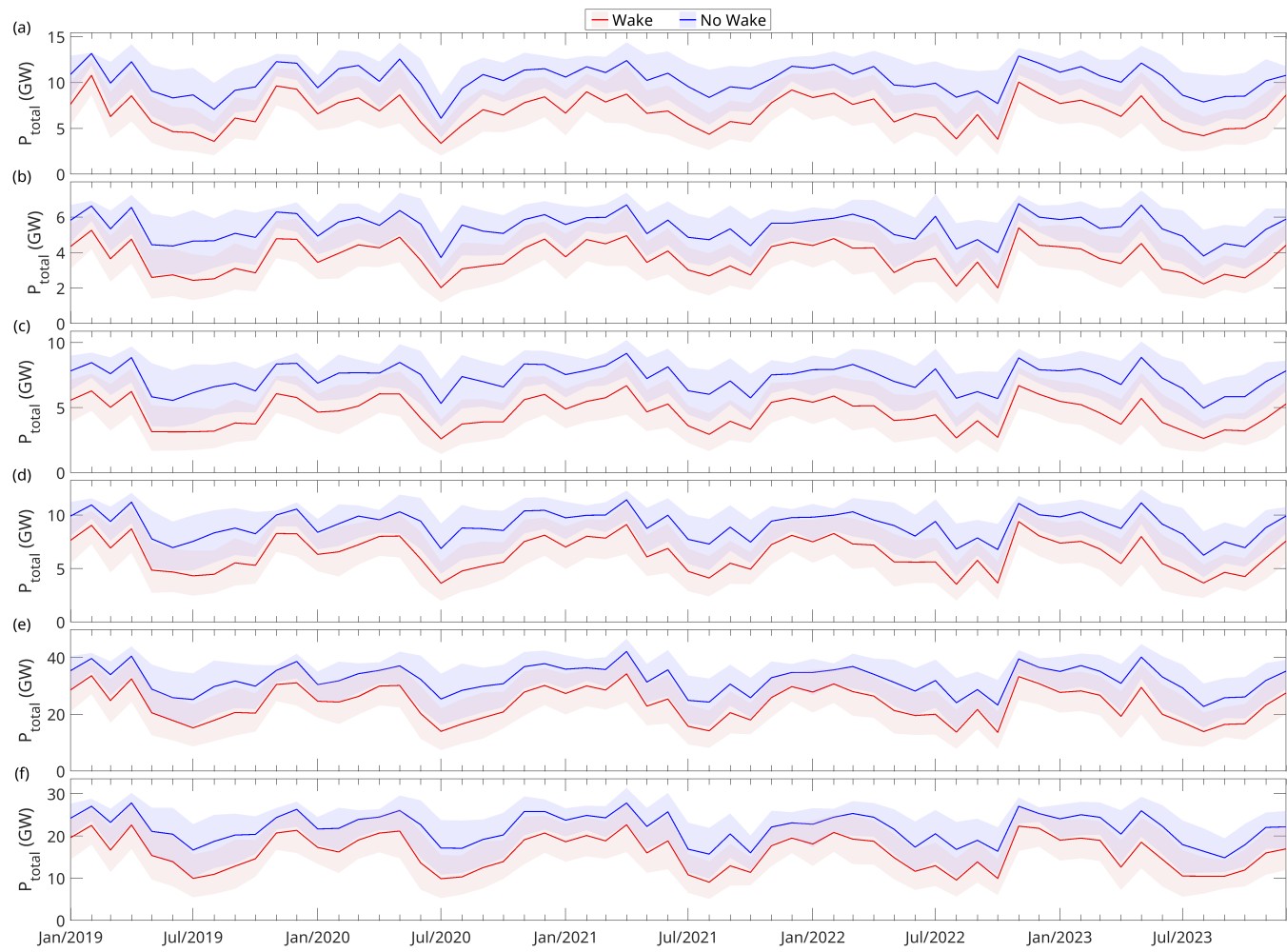

**Figure 12.** Time series of total power estimated from simulations for the 6 potential future development areas (PFDAs) of (a) Sydney Bight, (b) Canso Bank, (c) Eastern Shore, (d) Middle Bank, (e) Sable Island Bank, and (f) Emerald Bank using ERA5 wind dataset. The 'No Wake' curves indicate the theoretical maximum energy production without accounting for wake losses. Turbine spacings (L/D) are listed in Table 4. The shaded areas represent uncertainties due to differences in wind speeds between datasets and offshore wind observation sites. The uncertainties are quantified using the RMSE between the dataset and observed wind speeds. Refer to Figure 1 for the locations of PFDAs on the Scotian Shelf.

**Table 5.** Seasonal mean values of total power production, $P_{total}$ (GW), for the 6 potential future development areas (PFDAs) in winter (December to February) and summer (June to August) derived from the simulation results shown in Figure 12. Turbine spacings ($L/D$) and numbers varied across the 6 PFDAs, and the total number of turbines was determined for each wind farm based on this spacing. Uncertainties are represented by the maximum deviations of the seasonal mean of upper and lower bounds from the mean values.

| PFDA | L/D | Turbine Number | Winter | | Summer | |
|---|---|---|---|---|---|---|
| | | | Wake | No Wake | Wake | No Wake |
| Sydney Bight | 4.1 | 1071 | $8.4 \pm 2.3$ | $11.5 \pm 2.1$ | $5.0 \pm 2.5$ | $8.9 \pm 2.9$ |
| Canso Bank | 3.7 | 554 | $4.4 \pm 1.2$ | $5.9 \pm 1.1$ | $2.9 \pm 1.3$ | $4.9 \pm 1.5$ |
| Eastern Shore | 3.4 | 749 | $5.5 \pm 1.6$ | $7.9 \pm 1.5$ | $3.5 \pm 1.8$ | $6.5 \pm 2.1$ |
| Middle Bank | 4.2 | 926 | $7.7 \pm 1.9$ | $9.9 \pm 1.7$ | $4.8 \pm 2.2$ | $8.1 \pm 2.5$ |
| Sable Island Bank | 5.2 | 3347 | $28.8 \pm 7.0$ | $35.7 \pm 6.0$ | $17.5 \pm 8.4$ | $28.1 \pm 9.3$ |
| Emerald Bank | 5.1 | 2294 | $19.5 \pm 4.7$ | $24.2 \pm 4.1$ | $11.8 \pm 5.7$ | $18.9 \pm 6.3$ |

**Table 6.** Seasonal mean values of total power production, $P_{total}$ (GW), for the 6 potential future development areas (PFDAs) in winter (December to February) and summer (June to August) derived from the simulation results shown in Figure 13. Turbine spacings ($L/D$) were fixed for the 6 PFDAs and the total number of turbines was determined for each PFDA based on this spacing. Uncertainties are represented by the maximum deviations of the seasonal mean of the upper and lower bounds from the mean values.

| PFDA | L/D | Turbine Number | Winter | | Summer | |
|---|---|---|---|---|---|---|
| | | | Wake | No Wake | Wake | No Wake |
| Sydney Bight | 9.6 | 194 | $2.0 \pm 0.4$ | $2.1 \pm 0.4$ | $1.5 \pm 0.5$ | $1.6 \pm 0.5$ |
| Canso Bank | 9.6 | 87 | $0.9 \pm 0.2$ | $0.9 \pm 0.2$ | $0.7 \pm 0.2$ | $0.8 \pm 0.2$ |
| Eastern Shore | 9.6 | 97 | $1.0 \pm 0.2$ | $1.0 \pm 0.2$ | $0.8 \pm 0.3$ | $0.8 \pm 0.3$ |
| Middle Bank | 9.6 | 185 | $2.0 \pm 0.4$ | $2.0 \pm 0.3$ | $1.5 \pm 0.5$ | $1.6 \pm 0.5$ |
| Sable Island Bank | 9.6 | 1004 | $10.4 \pm 2.1$ | $10.7 \pm 1.8$ | $7.5 \pm 2.7$ | $8.4 \pm 2.8$ |
| Emerald Bank | 9.6 | 653 | $6.7 \pm 1.3$ | $6.9 \pm 1.2$ | $4.8 \pm 1.7$ | $5.4 \pm 1.8$ |

the 'Wake' case were generally lower than those for the 'No Wake' case. For instance, at the Middle Bank, simulated power output ($1.5 \pm 0.6$ GW) was only about 6% less than that of the 'No Wake' case ($1.6 \pm 0.5$ GW).

Uncertainties in power estimation arising from wind direction errors between datasets and observations were also accounted for in this analysis. To quantify these uncertainties, separate simulations were conducted using wind directions perturbed based on dataset-specific error characteristics. Unlike wind speed, which has a monotonic relationship with power production, where an increase or decrease in wind speed directly leads to a corresponding change in power output, wind direction does not influence power generation in a strictly linear manner. Variations in wind direction can alter wake interactions in complex ways, making the uncertainty estimation less straightforward.

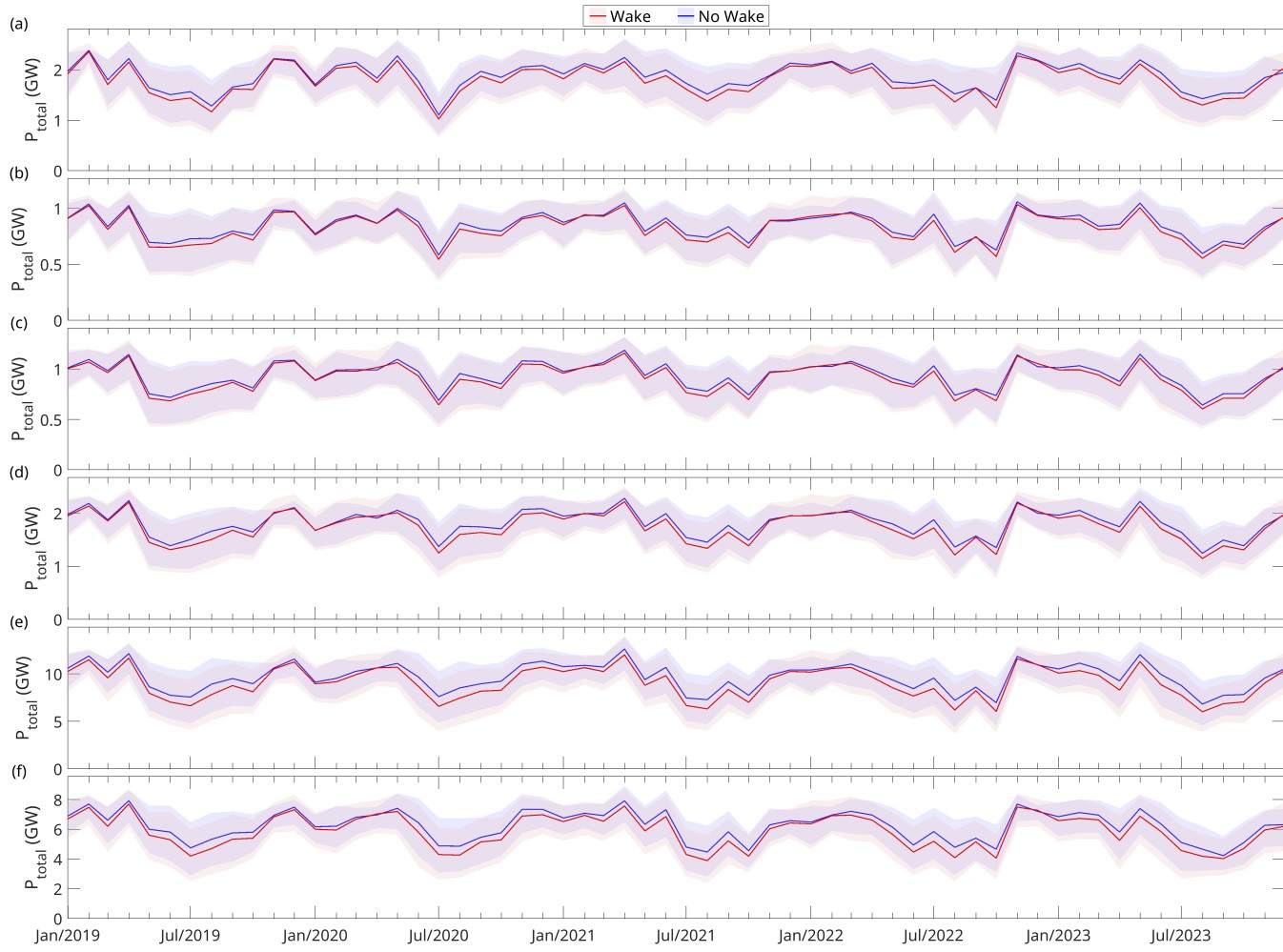

**Figure 13.** Same simulations described in Figure 12 and Table 5, but with a turbine spacing set to 9.6 times the rotor diameter.

To address this, multiple simulations were performed across a range of possible wind directions to capture the potential variation in power output. For each month, the range of possible wind directions was defined by adding and subtracting the monthly RMSE from the original wind direction time series. Within this range, additional wind direction time series were generated, with values evenly distributed between the upper and lower bounds. Based on sensitivity testing, it was found that using 10 perturbations provided sufficient resolution to capture variability in power production without incurring excessive computational cost. These 10 perturbed time series, along with the original, resulted in a total of 11 simulations. At each hour, the uncertainty in power production was then defined by the minimum and maximum values across these 11 simulations.

In the dense layout scenario, uncertainties of power production resulting from wind direction estimation errors (Figure 14 and Table 7) were comparable in magnitude to those caused by wind speed estimation errors (Figure 12 and Table 5). However,

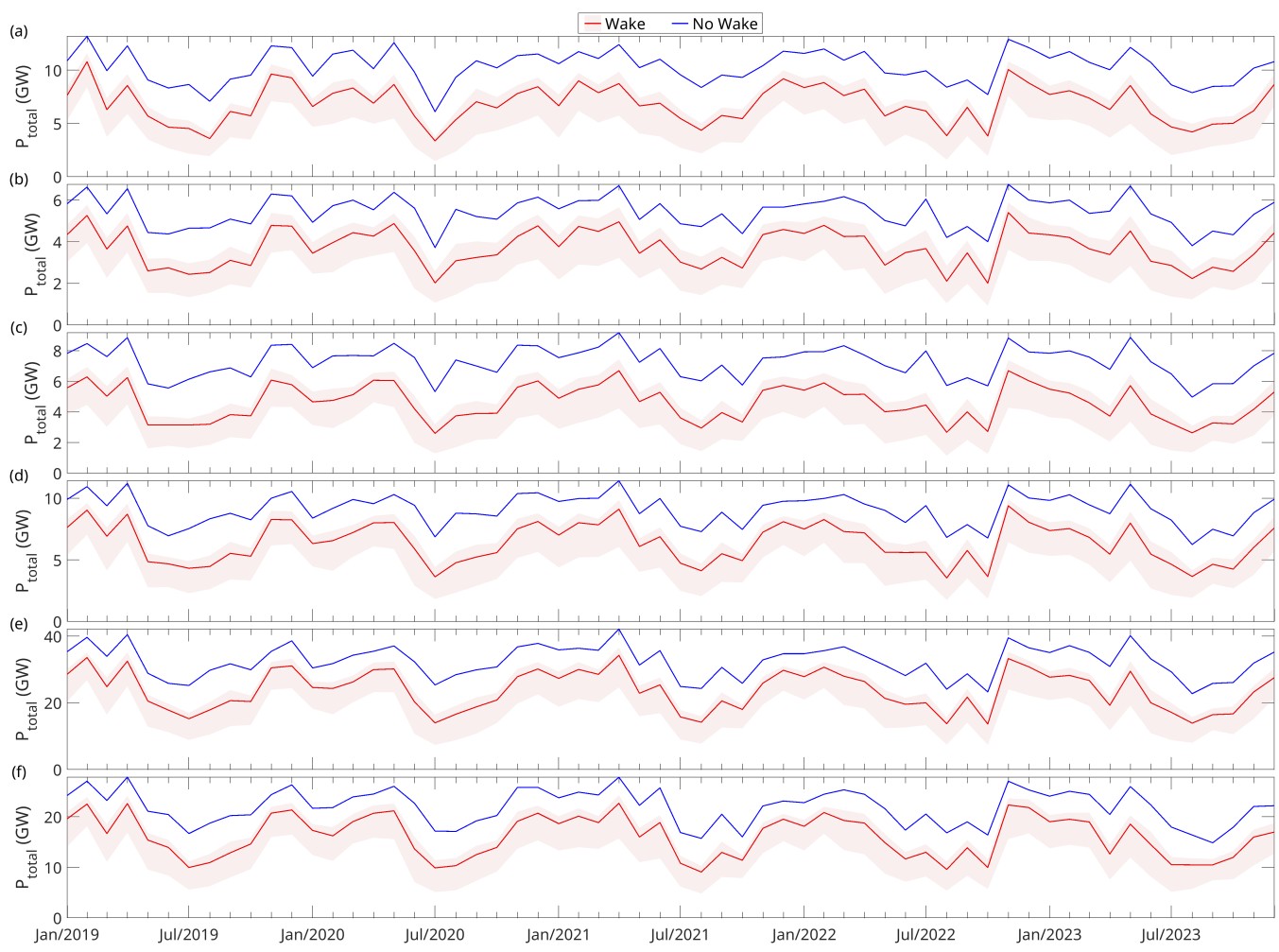

**Figure 14.** Same simulations described in Figure 12 and Table 5, except that the uncertainties are estimated to account for the wind direction estimation errors.

**Table 7.** Seasonal mean values of total power production, $P_{total}$ (GW), for the 6 potential future development areas (PFDAs) in winter (December to February) and summer (June to August) derived from the simulation results shown in Figure 14. Turbine spacings ($L/D$) varied across the 6 PFDAs. Uncertainties are represented by the maximum deviations of the seasonal mean of upper and lower bounds from the mean values.

| PFDA | L/D | Winter | Summer |
|------|-----|--------|--------|
| Sydney Bight | 4.1 | $8.4 \pm 2.5$ | $5.0 \pm 2.4$ |
| Canso Bank | 3.7 | $4.4 \pm 1.3$ | $2.9 \pm 1.3$ |
| Eastern Shore | 3.4 | $5.5 \pm 1.8$ | $3.5 \pm 1.7$ |
| Middle Bank | 4.2 | $7.7 \pm 2.2$ | $4.8 \pm 2.2$ |
| Sable Island Bank | 5.2 | $28.8 \pm 7.2$ | $17.5 \pm 7.3$ |
| Emerald Bank | 5.1 | $19.5 \pm 4.9$ | $11.8 \pm 5.1$ |

in the $L/D = 9.6$ scenario, the impact of wind direction errors (Figure 15 and Table 8) was less significant compared to the uncertainties introduced by wind speed errors (Figure 13 and Table 6).

**Table 8.** Seasonal mean values of total power production, $P_{total}$ (GW), for the 6 potential future development areas (PFDAs) in winter (December to February) and summer (June to August) derived from the simulation results shown in Figure 15. Uncertainties are represented by the maximum deviations of the seasonal mean of the upper and lower bounds from the mean values.

| PFDA | L/D | Winter | Summer |
|------|-----|--------|--------|
| Sydney Bight | 9.6 | $2.0 \pm 0.2$ | $1.5 \pm 0.3$ |
| Canso Bank | 9.6 | $0.9 \pm 0.1$ | $0.7 \pm 0.1$ |
| Eastern Shore | 9.6 | $1.0 \pm 0.1$ | $0.8 \pm 0.1$ |
| Middle Bank | 9.6 | $2.0 \pm 0.2$ | $1.5 \pm 0.3$ |
| Sable Island Bank | 9.6 | $10.4 \pm 1.1$ | $7.5 \pm 1.5$ |
| Emerald Bank | 9.6 | $6.7 \pm 0.7$ | $4.8 \pm 1.0$ |

## 5  Discussion

Offshore wind energy holds significant promise in the global transition from fossil fuels to clean energy. The Scotian Shelf is recognized for its world-class wind resources (Nicholson, 2023; Government of Nova Scotia, 2023), presenting a significant opportunity for offshore wind farm development that has now been embedded in the federal and provincial energy strategies (Government of Nova Scotia, 2023; Canada, n.d.). Motivated by a need to better understand wind energy potential in the Scotian Shelf offshore area, this study focused on two primary aspects: assessment of wind datasets and estimation of potential power production in 6 PFDAs on the Scotian Shelf.

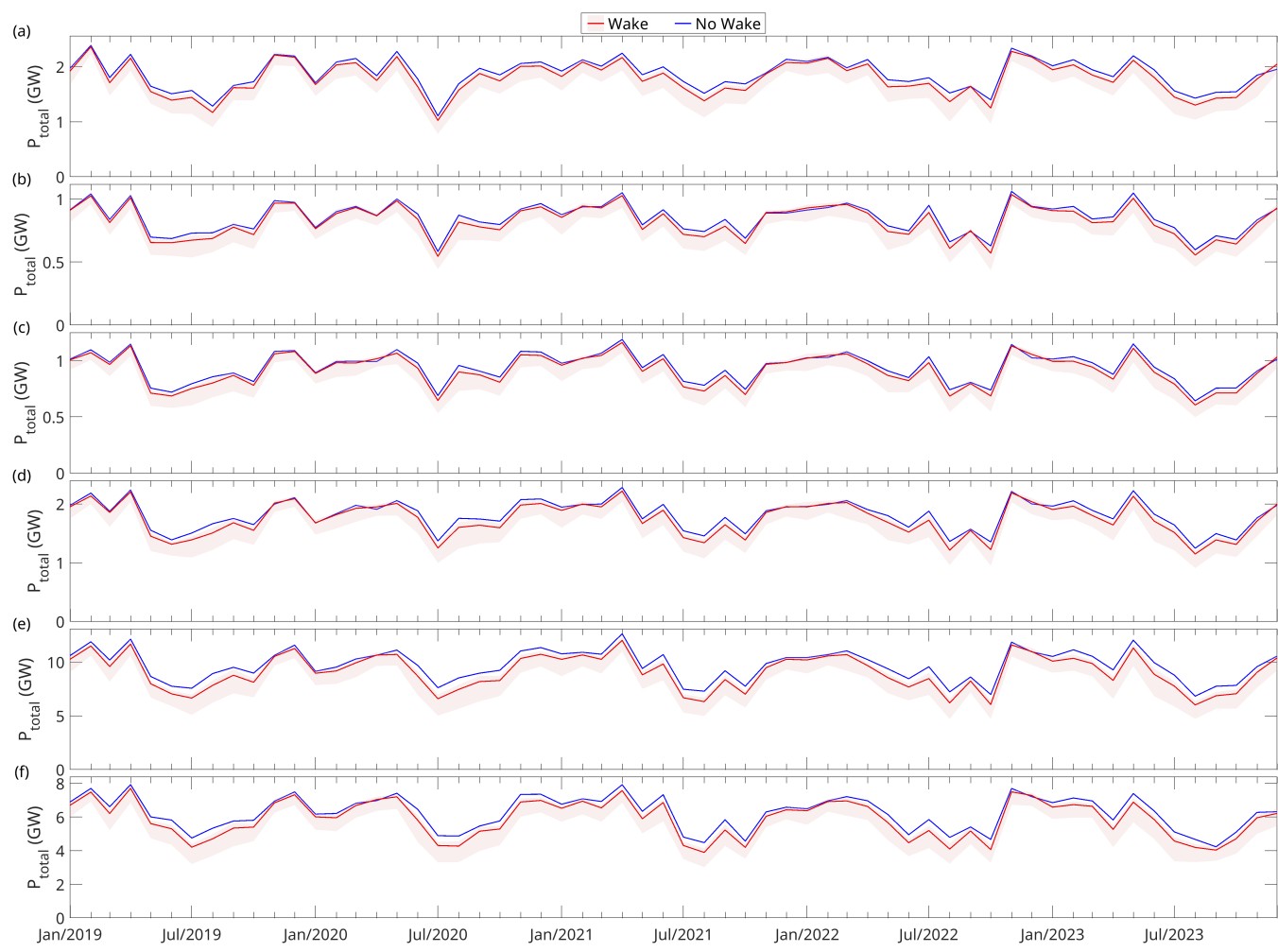

**Figure 15.** Same simulations described in Figure 13 and Table 6, except that the uncertainties are estimated to account for the wind direction estimation errors.

## 5.1 Wind Dataset Assessment

The wind datasets assessed in this study are widely used and have been evaluated in various regions worldwide (Fan et al., 2021; Fernandes et al., 2021; Li et al., 2010; Milbrandt et al., 2016). However, their performance can vary spatially, necessitating region-specific assessments. For wind farm development and design configurations, accuracy of modeled wind data is crucial for reliable energy potential estimates. On the Scotian Shelf, few studies have evaluated the applicability of various wind datasets to the region. This study provides a comprehensive assessment of wind speed and direction for the Scotian Shelf, providing a robust foundation for wind dataset selection in wind energy assessment.

The strong performance of ERA5 in the regional area of the Scotian Shelf aligned with previous studies for other regions, both inland and offshore (Fan et al., 2021; Murcia et al., 2022). These studies consistently found that ERA5 exhibited lower biases and mean absolute errors, along with higher correlations compared to other wind datasets. The overall robust performance of ERA5 was likely attributed to its advanced data assimilation techniques (Hersbach et al., 2020). In offshore regions, in particular, ERA5 has proven to be highly reliable (Gualtieri, 2021). Even when compared to a high-resolution regional Weather Research & Forecasting (WRF) Model, which employed nested grids with resolution ranging from 18 km to 2 km, ERA5 (31 km resolution) demonstrated superior performance. Gualtieri (2021) reported that ERA5 achieved lower RMSE and bias, along with a higher correlation coefficient, when validated against observations from an offshore mast in the North Sea. This advantage can be attributed to the relatively homogeneous wind conditions in offshore regions, where high-resolution models do not provide a notable improvement over ERA5. However, in nearshore areas, the results of this study revealed that HRDPS outperformed ERA5, consistent with the findings of Gualtieri (2021). The reduced performance of ERA5 in coastal transition areas can be attributed to significant differences in surface roughness and temperature, which introduced more complex flow dynamics (Gualtieri, 2021; Dörenkämper et al., 2015; Cañadillas et al., 2023; Djath et al., 2022). These findings highlighted the advantages of high-resolution numerical models, such as HRDPS (2.5 km), in improving the accuracy of wind speed and direction estimates in complex nearshore environments by better resolving small-scale dynamics.

Electricity consumption fluctuates over time, exhibiting diurnal, weekly, and seasonal patterns. In Nova Scotia, energy consumption is typically higher during the colder months of December to February and lower during the warmer months of May to October (see: Figure A5). These variations underscore the importance of assessing wind datasets at different local times and across seasons, in order to better align wind energy production with electricity demand cycles. Results of this study revealed notable seasonal and diurnal variability in performance across the wind datasets. Errors in wind speed estimation were generally higher in the fall and winter months, aligning with periods of stronger and more variable winds. In contrast, the spring and summer months exhibited lower errors.

In this study, the measured wind speeds at buoy sites were extrapolated from 5 m to 10 m height using the power law relationship. This relationship is not based on any fundamental physical principles, but rather is an empirical formulation in which the shear exponent can vary widely depending on terrain characteristics and atmospheric stability conditions (Emeis, 2018). For open water under neutral stability, Hsu et al. (1994) estimated $\alpha = 0.11$, a value that has been widely adopted in offshore wind energy studies. However, this value applies only under neutrally stable conditions, which are not always typical

of the atmospheric boundary layer over the ocean. In reality, atmospheric stability often deviates from neutral. For example, Argyle and Watson (2014) analyzed a three-year dataset from two meteorological masts in the North Sea and found that unstable conditions occurred most frequently. Similarly, Archer et al. (2016) reported that at a meteorological tower off the U.S. Northeast coast, the atmosphere was unstable 61% of the time, neutral 21%, and stable 18%. A more recent study by Dhomé et al. (2025) estimated $\alpha$ using wind data collected by Lidar and ultrasonic anemometers on a research vessel during an Atlantic cruise. Their observations showed that $\alpha$ varied greatly depending on location and atmospheric stability, ranging from -0.32 to 0.93. Lower values were typically found in offshore areas during unstable conditions, while higher values occurred under stable conditions in port areas. In offshore locations, the average values of $\alpha$ were approximately 0.04 for unstable, 0.12 for neutral, and 0.24 for stable conditions.

Because only one measurement height was available at the buoy sites, the shear exponent for wind speed extrapolation in this study was set to a constant value of 0.14, as recommended by the IEC 61400-3-1:2019 standard (IEC, 2019) for normal offshore wind conditions. This simplification introduces uncertainty in the extrapolation. If typical shear exponent values of 0.04 for unstable atmospheric conditions and 0.24 for stable atmospheric conditions were applied instead of the constant value of 0.14, the estimated 10 m wind speeds could differ by approximately $\pm$ 7%. Atmospheric stability exhibits a seasonal pattern, tending to be less stable in winter and more stable in summer, as the ocean is generally warmer than the overlying atmosphere in winter and cooler in summer (Jung and Schindler, 2021; Fragano and Colle, 2025). Consequently, applying a constant shear exponent likely leads to overestimation of wind speeds in winter and underestimation in summer at buoy sites. This may have contributed to the observed seasonal bias patterns, which differed from those at meteorological stations. For example, the bias at Sites 1 and 3 tended to be positive in winter and negative in summer, whereas at Site 2 the bias was more negative in winter and more positive in summer.

The observational wind data used in this study, obtained from meteorological stations and marine buoys, are subject to inherent measurement uncertainties. These can arise from small errors introduced during the calibration process, limitations in instrument accuracy, degradation over time (e.g., corrosion or component aging), environmental influences (such as icing and salt spray), and effects of nearby terrain, structures that distort airflow (World Meteorological Organization (WMO), 2024; Thomas and Swail, 2011; Malačič, 2019; Schlundt et al., 2020). For example, the uncertainties of cup anemometers can be on the order of 10% for wind speed and $\pm 5°$ for wind direction (World Meteorological Organization (WMO), 2024). These measurement uncertainties contribute to the discrepancies observed between datasets and *in situ* records, and therefore directly influence the RMSE, bias, MAE, and $R^2$ values used in dataset evaluation.

The wind datasets in this study were assessed at 10 m above the surface, well below the turbine hub height (150 m). The transferability of performance from low to high altitudes is uncertain. Ji et al. (2025) found that while $R^2$ at 10 m and 100 m were moderately correlated, bias and MAE showed weak correspondence. Similarly, Liu et al. (2023a) reported that using the power-law relationship to estimate higher-level winds led to declining performance, with RMSE increasing from 1.50 m/s at 120 m to 2.42 m/s at 200 m. These findings suggest that strong performance at low heights does not guarantee similar accuracy at hub height. Validating wind datasets with observations at turbine operational heights, obtained using lidar or tall masts, can provide more accurate assessments for wind energy applications.

At some offshore sites that had wind observations from marine buoys, wind directions from all datasets consistently had discrepancies during certain periods. Such consistent discrepancies across all datasets were likely caused by an inaccurate measurement of wind direction measured at the marine buoy at some locations. Unlike wind speed, which is relatively easy to measure, measuring wind direction is more difficult. Multiple factors, such as buoy motion due to waves or a misalignment of buoy orientation relative to true north, can cause errors in wind direction measurements from such platforms (Malačič, 2019; Schlundt et al., 2020).

Beyond wind energy applications, the findings have broader implications for regional ocean modeling, where wind datasets serve as key surface boundary conditions. For large-scale ocean models of the Scotian Shelf, both ERA5 and HRDPS are viable wind datasets to use, while HRDPS is preferable for coastal modeling due to its higher resolution and improved representation of nearshore wind patterns.

## 5.2 Power Production Simulation

Wind energy estimates from wind speed using theoretical formulas often omit energy losses associated with turbine wakes (Nicholson, 2023; Wang et al., 2022). In reality, turbine wakes can significantly reduce total wind farm power output, with downstream effects extending to turbines located further along the flow field. These wake effects thus are key considerations when balancing energy production and overall wind farm efficiency, as the spacing between turbines and the positioning within the wind farm impact the effectiveness of the entire operation.

Simulations in this study emphasized the trade-off between potential energy production and turbine density. In this case, increased density amplified wake losses, which reduced overall efficiency. In contrast, greater turbine spacing decreased wake interactions that enhanced power output per turbine, but reduced the total energy yield due to fewer turbines occupying a given area.

The empirical piecewise function derived in this study provided valuable insights into the trade-off between energy generation efficiency and total power production, offering a quantitative framework for optimizing wind farm layouts. This two-regime function was characterized by a critical transition point, $x_t$, that defined the normalized turbine spacing beyond which wake effects became minimal. For smaller turbine spacings ($L/D < x_t$), the function followed a Weibull-like function form, reaching its maximum at $L/D = x_m$. Below this threshold, total power production declined due to intensified wake interactions. This suggested that normalized turbine spacings smaller than $x_m$ should be avoided, as both total power output and wake efficiency decreased. For larger turbine spacings ($L/D > x_t$), wake effects were minimal, leading to improved wake efficiency.

Winter wind conditions with higher wind speeds generally favored smaller turbine spacings. This was because the thrust coefficient ($C_t$) decreased at higher wind speeds, resulting in less significant wakes and faster wake recovery, which allowed turbines to be placed closer together without substantial efficiency losses. Consequently, the values of $x_t$ and $x_m$ were typically smaller in winter than in summer. For most PFDAs, achieving optimal wake efficiency in summer required the normalized turbine spacings to be greater than 6. Meanwhile, maximizing total power production can be achieved by selecting a normalized turbine spacing corresponding to the mean value of $x_m$ of both seasons, which ranged from 3.4 to 5.2.

Wake losses proved to be significant in the simulations, with values ranging from 19% to 30% in winter and 37% to 46% in summer in the dense turbine layout scenario. Power losses decreased as turbine spacing increased. In the scenario where turbine spacing was set to 9.6 D, the power losses became negligible in winter and were less than 12% in summer. The wake losses in the dense turbine layout scenario were generally higher than the 10% to 25% range reported in other studies on medium-sized offshore wind farms (Barthelmie et al., 2009, 2010; Niayifar and Porté-Agel, 2015; Simisiroglou et al., 2019; Wu and Porté-Agel, 2015). This discrepancy can be attributed to multiple factors, such as turbine models, spacings, and wind farm sizes. In the dense turbine layout scenario presented, turbine spacings that ranged from 3.4 to 5.2 were smaller than those in the aforementioned studies, which contributed to higher wake losses simulated in this study. Last, the findings presented in this study were in line with results of simulations for larger offshore wind farms reported by Pryor et al. (2021), who reported an overall wake loss of 35.3%. This value falls between the results for the dense turbine layout scenario in winter and summer within this study.

Uncertainties in total power output were also assessed based on discrepancies in both wind speed and wind direction between wind datasets and observations. To quantify these uncertainties, wind speed and wind direction time series were perturbed separately, where the dataset-specific RMSE was used to define the error bounds. This approach provided clear illustration of how wind speed and wind direction errors propagated into power estimations. Notably, in the dense layout scenario, errors in wind direction had an impact on power production that was just as significant as errors in wind speed. This result underscores the importance of wind direction accuracy in estimating power production, as deviations can alter wind turbine wake interactions and influence overall energy generation.

Limitations of perturbing wind speed or direction time series should be acknowledged. Perturbing wind speed involved simply adding or subtracting RMSE, while perturbing wind direction employed 10 evenly spaced values within the bounds of positive and negative RMSE. Neither method accounted for the probability distribution of their respective errors; particularly, when the monthly wind speed or direction bias largely deviated from 0. As a result, this method may overestimate or underestimate the upper and lower bounds of the uncertainties. A more sophisticated approach, that could address this issue, would involve probabilistic uncertainty modeling, such as Monte Carlo simulations (Singh and Taylor, 2018; Liu et al., 2023b) that provides a more rigorous representation of how wind speed or direction error distributions impact power estimations. However, compared to the Monte Carlo method, the approach used in this study was computationally efficient, making it a more practical choice for large-scale wind farm assessments.

Many studies that have simulated offshore wind farms have focused on annual or long-term mean energy production or capacity factors (Pryor et al., 2021; Simisiroglou et al., 2019; Wu and Porté-Agel, 2015), where the wind inputs consisted of several combinations of constant wind speeds and directions derived from historical wind statistics. However, temporal variation of power production has received comparatively less research focus (Wang et al., 2022). This study helps address this gap by analyzing the monthly time series of power production across PFDAs on the Scotian Shelf. Seasonal variations were pronounced, with summer production in the dense turbine layout being 34% to 40% lower than in winter. In the less dense 9.6 D layout, reductions in summer power production ranged from 20% to 28% compared to winter. Despite these fluctuations, seasonal wind power generation patterns closely aligned with seasonal variation in Nova Scotia electricity demand, which

peaks in the winter and declines in the summer. This alignment suggests that offshore wind has the potential to complement Nova Scotia's seasonal electricity demand, reinforcing the importance of incorporating temporal variability into wind energy assessments.

## 6  Conclusion

This research study is the first comprehensive assessment of wind speed and direction data from 4 widely used wind datasets, ERA5, CFSv2, NARR and HRDPS, across the Scotian Shelf, which is a region with world-class wind energy potential. The analyses highlighted spatial and temporal variability in dataset performance: HRDPS emerged as the most accurate dataset for nearshore wind conditions, whereas ERA5 proved the most reliable for broader offshore areas. Seasonal and diurnal variations further underscored the need for careful dataset selection when modelling wind energy potential.

The PFDA simulations using PyWake demonstrated the significant impact of wake interactions and turbine spacing on energy production. The results highlighted trade-offs in maximizing total power output and minimizing wake-induced losses. For most PFDAs, achieving effective wake efficiency year-round required turbine spacing to be larger than 6 D, whereas maximizing total power production was best achieved with a denser layout of 3.4–5.2 D. Seasonal variations further influenced wake dynamics, reinforcing the importance of considering temporal wind variability.

Overall, the research findings provided valuable insights for offshore wind development in Nova Scotia, emphasizing the need for accurate wind resource assessment and strategic turbine layout. Through integration of high-resolution wind datasets and accounting for seasonal and wake effects, wind farm design on the Scotian Shelf can be optimized for energy production and long-term efficiency.

*Data availability.*  The data generated in this study is available upon reasonable request.

## Appendix A: Appendix

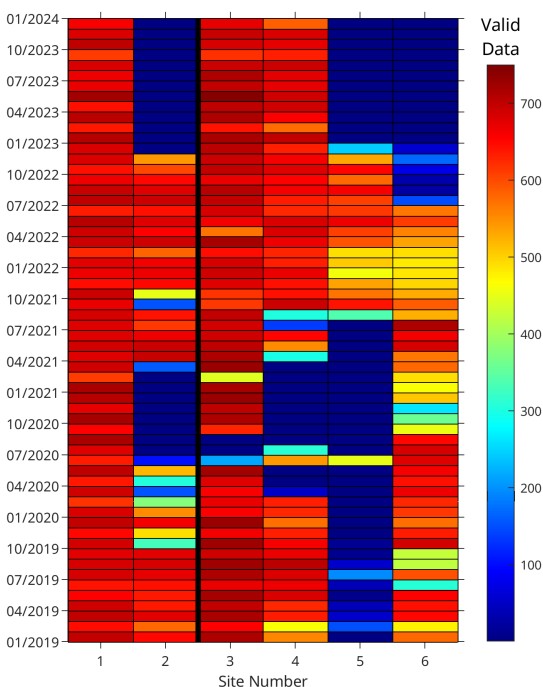

**Figure A1.** Pseudocolor plots showing the number of the valid observation data in each month from January 1, 2019 to December 31, 2023. Sites 1 and 2 are representative of the nearshore (left of bold black line) and Sites 3 - 6 are representative of the offshore (right of bold black line) on the Scotian Shelf. Refer to Table 1 and Figure 1 for information of the observation sites.

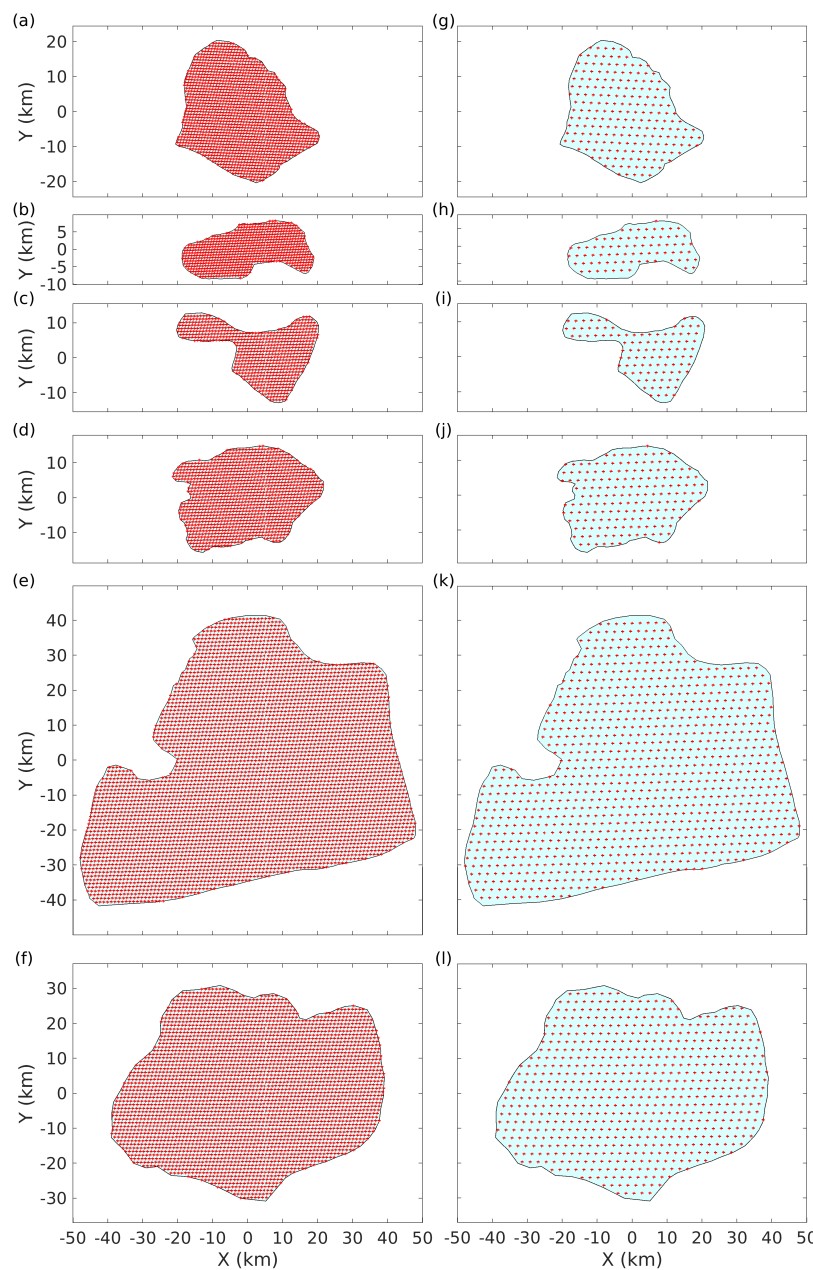

**Figure A2.** Diagrams of the layouts of wind turbines for the 6 potential future development areas (PFDAs): (a, g) Sydney Bight, (b, h) Canso Bank, (c, i) Eastern Shore, (d, j) Middle Bank, (e, k) Sable Island Bank, and (f, l) Emerald Bank. The normalized turbine spacings ($L/D$) for the left panels are 4.1, 3.7, 3.4, 4.2, 5.2 and 5.1, respectively, and 9.6 for all right panels. Refer to Figure 1 for the locations of PFDAs on the Scotian Shelf.

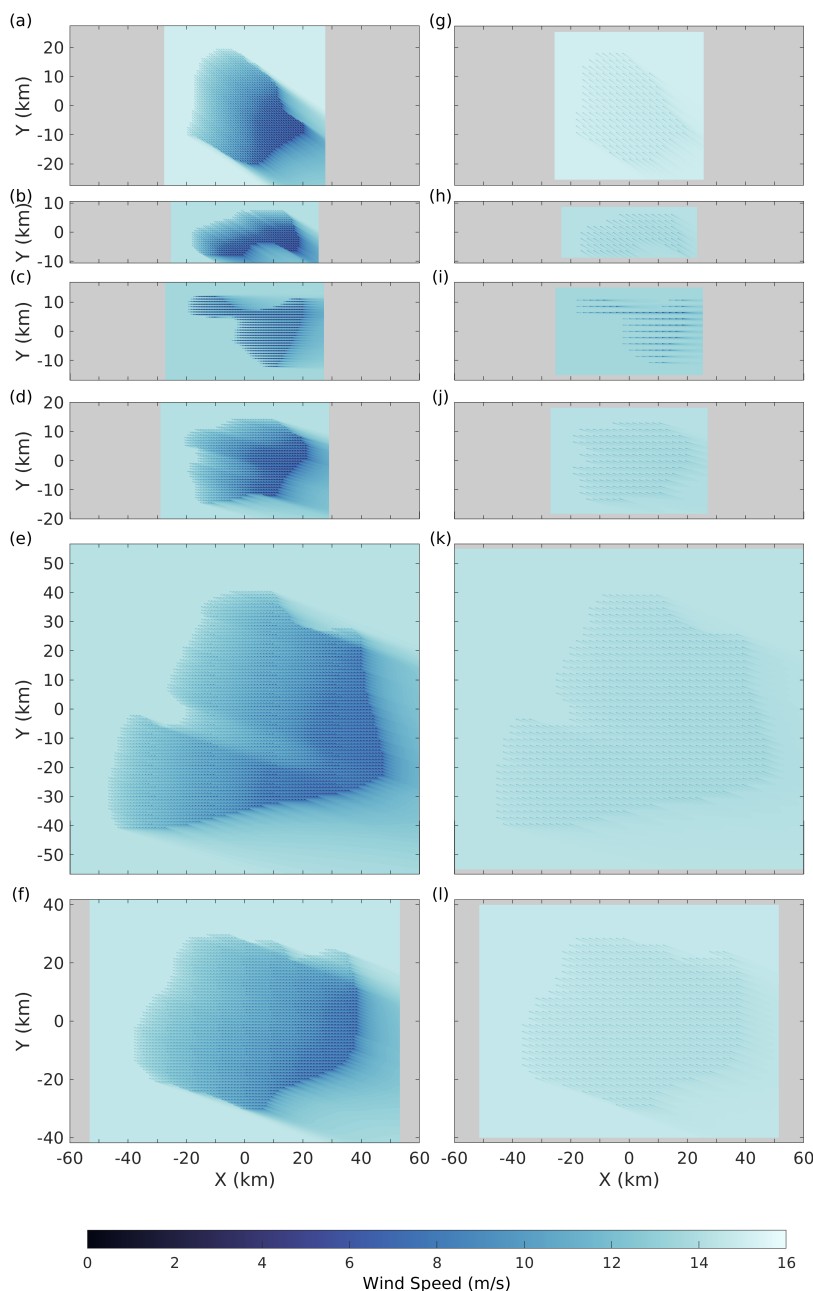

**Figure A3.** Flow maps of wind speed at the hub height of 150 m and wake effects simulated using PyWake for the 6 potential future development areas (PFDAs): (a, g) Sydney Bight, (b, h) Canso Bank, (c, i) Eastern Shore, (d, j) Middle Bank, (e, k) Sable Island Bank, and (f, l) Emerald Bank during winter, with the same layouts as in Figure A2. The wind dataset used was ERA5. Refer to Figure 1 for the locations of PFDAs on the Scotian Shelf.

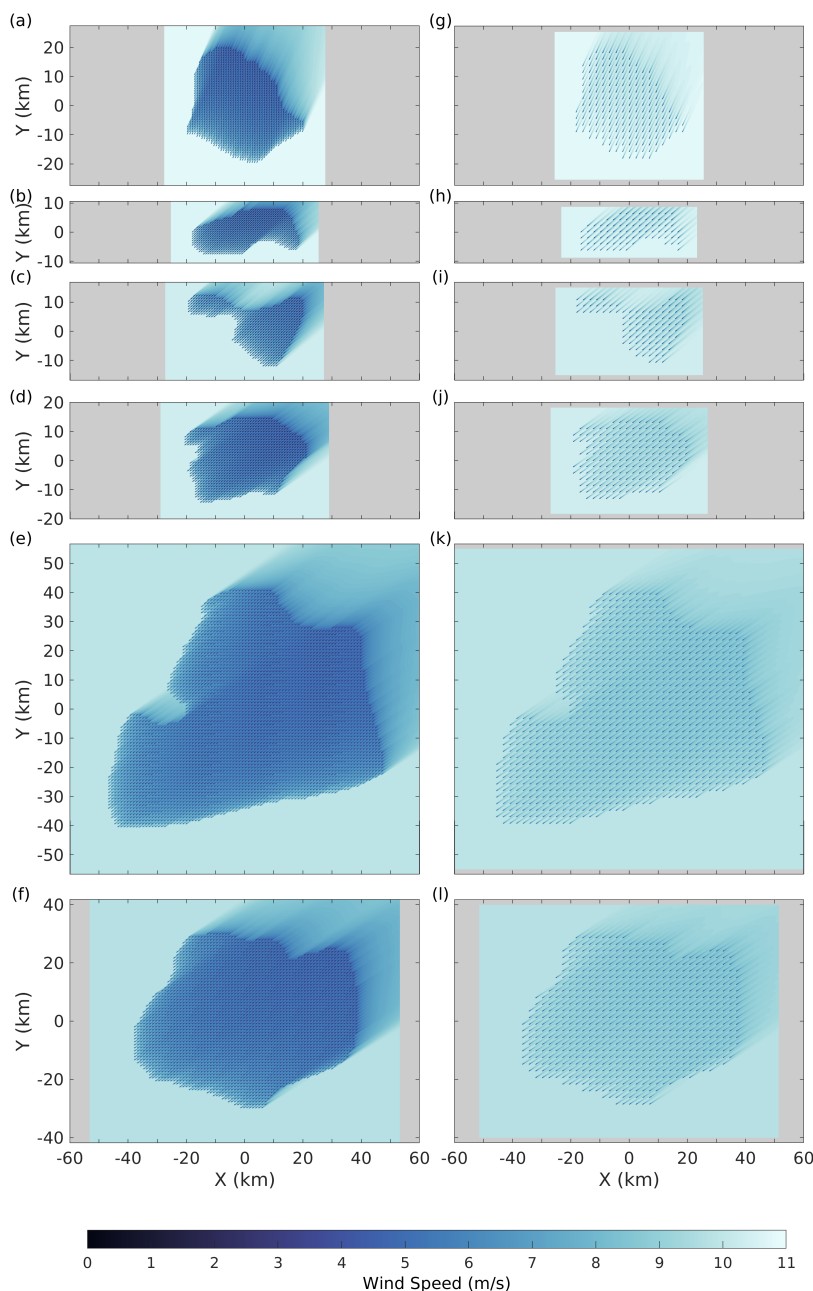

**Figure A4.** Flow maps of wind speed at the hub height of 150 m and wake effects simulated using PyWake for the 6 potential future development areas (PFDAs): (a, g) Sydney Bight, (b, h) Canso Bank, (c, i) Eastern Shore, (d, j) Middle Bank, (e, k) Sable Island Bank, and (f, l) Emerald Bank during summer, with the layouts same as in Figure A2. The wind dataset used was ERA5. Refer to Figure 1 for the locations of PFDAs on the Scotian Shelf.

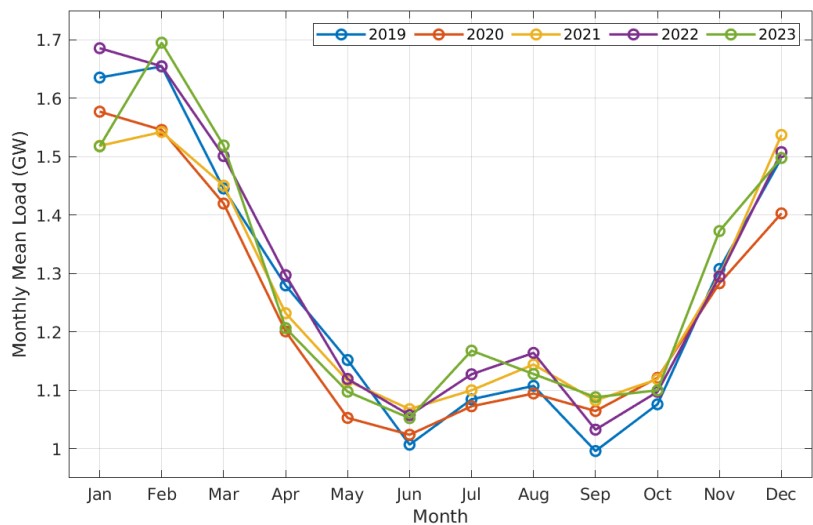

**Figure A5.** Monthly mean load for Nova Scotia, Canada, in 2019–2023. Data sourced from the website of Nova Scotia Power: https://www.nspower.ca/oasis/monthly-reports/hourly-total-net-nova-scotia-load.

*Author contributions.* Y.M.: Data Curation, Methodology, Software, Formal Analysis, Visualization, Writing —- Original Draft, Writing — Review & Editing. J.X.: Writing – Review & Editing. Y.W.: Conceptualization, Methodology, Funding Acquisition, Project Administration, Supervision, Writing – Review & Editing. M.Z. L.: Writing – Review & Editing. R.S.: Writing – Review & Editing. B.L.: Writing – Review & Editing. M.S.: Writing – Review & Editing. All authors have reviewed and approved the final manuscript.

*Competing interests.* The authors declare that they have no conflict of interest.

*Acknowledgements.* This research study was funded under the Fisheries and Oceans Canada (DFO) Competitive Science Research Fund (CSRF, fund number 2024-25-04-01) awarded to Y. Wu. The authors thank A. Cogswell for his invaluable support and advice throughout proposal preparation and submission. The authors also thank K. Curran and E. Nagel for their meaningful review and comments on an early version of the manuscript. Finally, the authors thank the two anonymous reviewers for their constructive feedback, which greatly improved the manuscript.

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
