# Peer review of "Wind dataset assessment and energy estimation for potential future offshore wind farm development areas on the Scotian Shelf"

_Wind Energy Science, 2025_

## Referee Comment (RC1)

**Review Comment**

May 27th, 2025

Title: Wind dataset assessment and energy estimation for potential future offshore wind farm development areas on the Scotian Shelf

Author(s): Yongxing Ma, Jinshan Xu, Yongsheng Wu, Michael Z. Li, Ryan Stanley, Brent Law, and Marc Skinner

MS No.: wes-2025-57

MS type: Research article

Iteration: Initial submission

**General comment**

This manuscript validated several reanalysis data against measurement data and calculated/discussed wake effect due to turbine spacing at Scotian Shelf offshore site in Canada. In introduction part, there are many paper reviews and well summarized. Although there are some uncertainties of measurement data remain, general trend of each reanalysis data is well presented in validation part. Then, authors calculated wake effect in wind farm for both high dense layout, which could be maximize total power production in wind farm, and low dense layout, which minimize wake loss, and considered and modelled optimal spacing to maximize total power production of wind farm. Also, seasonal differences of these two topics are compared and well presented. Although the result of this manuscript itself may site specific, the methodology will be good reference in future project and there are some scientific interests.

In conclusion, a reviewer consider that this manuscript can be accepted with MINOR REVISION.

| Clause/   | Line number | Comments                                                                                                                                                                                                     |
|-----------|-------------|--------------------------------------------------------------------------------------------------------------------------------------------------------------------------------------------------------------|
| Subclause |             |                                                                                                                                                                                                              |
| 1.2       | 63-64       | ERA 5 is explained as mean percentage for all stations, while
JRA-55 and MERRA-2 are explained as its ranges. Although
it is not a part of your research, it should be explained by
faire criteria. |
| 2.1       | Table 1     | It is suggested show station height above sea level and mounted height of measurement devices in the table.                                                                                                  |

Specific comment

| 2.1 | Overall | Authors have to explain how Quality level flag (if exist) is
handled. Also, explanations about measurement devices,
and its consistency in validation period, definition of wind
direction (i.e. true north or magnetic north) are missing. It
is also suggested to add a table about monthly data
availability. |
|-----|---------|---------------------------------------------------------------------------------------------------------------------------------------------------------------------------------------------------------------------------------------------------------------------------------------------------------------------------------|
| 2.4 | 174-175 | Although authors assume neutral stratification and alpha = 1/7, which is international standard, it is not clear if this assumption is correct in Scotian Shelf, and if not how big impact is given on validation result. Authors have to explain in this section or discussion section.                                        |
|     | 177     | Need explanations about symbols "U" and "z".                                                                                                                                                                                                                                                                                    |
| 2.4 | 178-179 | ERA5 has UV component 100m above ground and it reduces
vertical extrapolation uncertainty. Authors need to explain
the reasons why 10m height data is used, if there is.                                                                                                                                                  |
| 2.5 | 197     | Need explanation about symbol " $\bar{O}$ ".                                                                                                                                                                                                                                                                                    |
|     | 198     | "measured at 10m" should be "at 10m"                                                                                                                                                                                                                                                                                            |
| 3.1 | 248-249 | higher -> lower? The sentence is bit difficult to understand.                                                                                                                                                                                                                                                                   |
|     | 248-250 | CFSv2 is better than ERA5 at Site 2. It is better to add the explanation.                                                                                                                                                                                                                                                       |
|     | 258-260 | "RMSE values tended to increase during winter months",
This just may be because magnitude of wind speed is high.
Use of normalized RMSE may help further understanding.                                                                                                                                                   |
| 3.2 | 388     | The use of the word "overestimated" toward wind direction is weird. It is suggested to use "shifted (anti)clockwise".                                                                                                                                                                                                           |
|     | 390     | "April to December in 2022"
Need to explain that this explanation is about Site 5. It also                                                                                                                                                                                                                                   |

|     |                   | seems that July 2020 to July 2021 at Site 6 shows different
trend, compared to other years on the site. Need a
comment that how authors think this about this period.
Also, explanation that how authors handle these
measurement errors when calculate aggregate metrics
mentioned in text, Figure 8, Table 4 etc. in this sub section
is needed.                                        |
|-----|-------------------|-------------------------------------------------------------------------------------------------------------------------------------------------------------------------------------------------------------------------------------------------------------------------------------------------------------------------------------------------------------------------------------------------------------|
| 4.1 | Table 5           | It is better to add text that " $x_t$ "and " $x_m$ " will be explained later in section 4.2, like as Figure 9.                                                                                                                                                                                                                                                                                              |
|     | 438               | Explanation about data period used for "spatial and
seasonal mean" is needed. This may the same as line 478
but should be mentioned here.                                                                                                                                                                                                                                                             |
|     | 438-439, Figure 9 | "These values were determined as the spatial and seasonal
mean for each PFDA."
I understand wind speeds and wind directions shown in
Table 5 are used in this section to reduce computational
cost. Is my understanding correct? If so, why P unit in Figure
9 reaches 15MW? All wind speeds in Table 5 are lower than
10.6m/s, which is rated wind speed of IEA 15MW turbine. |
|     | 456               | "P unit revealed the average turbine efficiency"
I understand P unit is average of "all wind turbine" in a PFDA.
If so, the authors need to explain that (i.e. what consist of
average). Also, it is not "efficiency" but "power output",
isn't it?                                                                                                                       |
| 4.2 | Figure 11-(a)     | It is suggested to write brief explanation (e.g. Summer, 9.6D) in each figure.                                                                                                                                                                                                                                                                                                                              |
| 4.3 | 603               | "10 additional time series"
It is difficult to understand why 10? It is suggested to add
brief explanation (e.g. 5 yeas data and ±).                                                                                                                                                                                                                                                                  |

| 5.1 | Overall | Discussion regarding uncertainty of measurement data is needed |
|-----|---------|----------------------------------------------------------------|
|     |         | needed.                                                        |
| 5.2 | 663     | Wind frame -> wind farm?                                       |

---

## Author Comment (AC1)

**General comments:**

This manuscript validated several reanalysis data against measurement data and calculated/discussed wake effect due to turbine spacing at Scotian Shelf offshore site in Canada. In introduction part, there are many paper reviews and well summarized. Although there are some uncertainties of measurement data remain, general trend of each reanalysis data is well presented in validation part. Then, authors calculated wake effect in wind farm for both high dense layout, which could be maximize total power production in wind farm, and low dense layout, which minimize wake loss, and considered and modelled optimal spacing to maximize total power production of wind farm. Also, seasonal differences of these two topics are compared and well presented. Although the result of this manuscript itself may site specific, the methodology will be good reference in future project and there are some scientific interests.

In conclusion, a reviewer consider that this manuscript can be accepted with MINOR REVISION.

**Specific Comments and Responses**

**Comment:** ERA 5 is explained as mean percentage for all stations, while JRA-55 and MERRA-2 are explained as its ranges. Although it is not a part of your research, it should be explained by fair criteria.
**Response:** The authors intended to express that the mean percentage for other three datasets are in the range from -54.22% to 42%. This might be confusing. Now we have improved our expression as "*The authors found that ERA5 demonstrated the best overall performance among the five reanalysis wind dataset products, with ERA5 exhibiting a mean percent bias for all stations of -4.54 %, while the mean percent bias was -54.22% for JRA-55, -49.63% for CFSv2 and 42.03% for MERRA-2*".

**Comment:** It is suggested show station height above sea level and mounted height of measurement devices in the table.
**Response:** We modified the table as suggested.

**Comment:** Authors have to explain how Quality level flag (if exist) is handled. Also, explanations about measurement devices, and its consistency in validation period, definition of wind direction (i.e. true north or magnetic north) are missing. It is also suggested to add a table about monthly data availability.
**Response:** We added text about the quality level flags and the description of our quality control procedure. Information about wind measurement instruments was added. The definition of wind

direction is now stated in Section 2.1. The monthly data availability is shown in a new figure in the appendix.

**Comment:** Although authors assume neutral stratification and alpha = 1/7, which is international standard, it is not clear if this assumption is correct in Scotian Shelf, and if not how big impact is given on validation result.
**Response:** We added the reason for choosing the alpha value, following the IEC standard. The discussion about alpha's variability and dependence on atmospheric stability was also added.

**Comment:** Need explanations about symbols "U" and "z".
**Response:** We added the explanations for these symbols.

**Comment:** ERA5 has UV component 100m above ground and it reduces vertical extrapolation uncertainty. Authors need to explain the reasons why 10m height data is used, if there is.
**Response:** We now used both 10 m and 100 m wind speed data to calculate time-varying alpha values. These were then used for extrapolation to 150 m height.

**Comment:** Need explanation about symbol "$\overline{O}$".
**Response:** We explained it as: $\overline{O}$ denotes the average value of the observations.

**Comment:** "measured at 10m" should be "at 10m".
**Response:** We corrected this as suggested.

**Comment:** higher -> lower? The sentence is bit difficult to understand.
**Response:** We confirmed that ERA5 does show higher RMSE than HRDPS. We revised our sentence to make it clearer:
"Compared to HRDPS, ERA5 exhibited higher five-year averaged RMSE values of 1.76 ± 0.20 m/s and 2.08 ± 0.37 m/s at the two corresponding nearshore sites".

**Comment:** CFSv2 is better than ERA5 at Site 2. It is better to add the explanation.
**Response:** We revised the sentence to reflect that the performance of CFSv2 was mixed, and provided the relevant values.

**Comment:** "RMSE values tended to increase during winter months…", This just may be because magnitude of wind speed is high. Use of normalized RMSE may help further understanding.

**Response:** We thank the reviewer for this insightful comment. After conducting additional analysis using normalized RMSE (calculated as RMSE divided by the mean observed wind speed), we confirmed that the reviewer's thought is correct. The originally observed increase in RMSE during winter months was indeed influenced by the higher wind speeds in winter. When evaluated using the normalized RMSE, the results revealed the opposite seasonal pattern: normalized RMSE values were generally lower in winter and higher in summer.

We have added this clarification to the revised manuscript (Section 3.1), stating:
"*However, this pattern was primarily due to the higher wind speed magnitudes in winter compared to summer. When normalized RMSE (calculated as RMSE divided by the mean observed wind speed) was used, it was found that the normalized errors were actually smaller in winter and larger in summer.*"

**Comment:** The use of the word "overestimated" toward wind direction is weird. It is suggested to use "shifted (anti)clockwise".

**Response:** We rephrased the expression as suggested.

**Comment:** "April to December in 2022" Need to explain that this explanation is about Site 5. It also seems that July 2020 to July 2021 at Site 6 shows different trend, compared to other years on the site. Need a comment that how authors think this about this period. Also, explanation that how authors handle these measurement errors when calculate aggregate metrics mentioned in text, Figure 8, Table 4 etc. in this sub section is needed.

**Response:** We thank the reviewer for this careful observation. In the revised manuscript, we have clarified that the period from April to December 2022 refers specifically to Site 5 and have explicitly specified the corresponding period for Site 6. The revised text now reads: "*At the three offshore sites observed using buoys, notable biases were present during specific periods, i.e., from April to December 2022 at Site 5, and from June 2020 to July 2022 at Site 6, which were likely caused by systematic observational errors.*"

The selection criterion for wind direction data was the same as that used for wind speed: all data from months with at least 120 valid hourly records were included in the calculation of both monthly and aggregated metrics. Although the periods of large errors at Sites 5 and 6 are suspected to have resulted from systematic observational issues, the metrics presented in Figure 8 and Table 4 include these periods. We acknowledge that such anomalies may

influence the results; however, we chose not to exclude them in order to maintain a consistent and objective screening criterion across all sites. This decision is reflected in the box plots in Figure 8, where outliers indicate the presence of anomalous data during specific periods at Sites 5 and 6. Notably, offshore sites exhibit more outliers than nearshore sites, which suggests the influence of the suspected observational errors at Sites 5 and 6.

We have added a clarification in Section 3.2 stating:
"*It is noted that for the offshore group, periods with suspected systematic observational errors at Sites 5 and 6 were not excluded from the analysis. All data from months with at least 120 valid hourly records were retained to maintain a consistent screening criterion across all sites. As a result, the box plots in Figure 8 reflect the influence of these anomalies, as indicated by a greater number of outliers at offshore sites compared to nearshore sites.*"

**Comment:** It is better to add text that "x_t" and "x_m" will be explained later in section 4.2, like as Figure 9.
**Response:** We added such text in the table caption to remind readers that x_t and x_m are explained in Section 4.2.

**Comment:** Need to explain data period for 'spatial and seasonal mean'.
**Response:** We revised the sentence to specify that values were averaged across ERA5 grid points within each PFDA and over the 2019–2023 period.

**Comment:** "These values were determined as the spatial and seasonal mean for each PFDA." I understand wind speeds and wind directions shown in Table 5 are used in this section to reduce computational cost. Is my understanding correct? If so, why Punit in Figure 9 reaches 15MW? All wind speeds in Table 5 are lower than 10.6m/s, which is rated wind speed of IEA 15MW turbine.
**Response:** We clarified that the wind speeds in Table 5 are at 10 m, and when extrapolated to 150 m they exceed the rated speed of 10.6 m/s.

**Comment:** "Punit revealed the average turbine efficiency" I understand Punit is average of "all wind turbine" in a PFDA. If so, the authors need to explain that (i.e. what consist of average). Also, it is not "efficiency" but "power output", isn't it?
**Response:** We clarified that Punit represents the average power output per turbine, calculated by dividing Ptotal by the number of turbines.

**Comment:** It is suggested to write brief explanation (e.g. Summer, 9.6D) in each figure.
**Response:** We added such labels to each panel in Figure 11(a).

**Comment:** "10 additional time series"
It is difficult to understand why 10? It is suggested to add brief explanation (e.g. 5 yeas data and ±).
**Response:** We thank the reviewer for requesting clarification. We have now expanded the explanation in the revised manuscript to clarify why 10 additional time series were used. Unlike wind speed, where power production is a monotonic function, the effect of wind direction on power output is non-monotonic due to wake interactions. Within the ±RMSE bounds, we generated a range of perturbed wind direction time series at small directional intervals to capture this variability. Through sensitivity testing, we found that 10 evenly spaced perturbations were sufficient to capture the range of power variation while keeping computational demands reasonable. This clarification has been added to Section 4.3 of the revised manuscript.

The revised text reads:
"*To address this, multiple simulations were performed across a range of possible wind directions to capture the potential variation in power output. For each month, the range of possible wind directions was defined by adding and subtracting the monthly RMSE from the original wind direction time series. Within this range, additional wind direction time series were generated, with values evenly distributed between the upper and lower bounds. Based on sensitivity testing, it was found that using 10 perturbations provided sufficient resolution to capture variability in power production without incurring excessive computational cost.*"

**Comment:** Discussion regarding uncertainty of measurement data is needed.
**Response:** We thank the reviewer for this suggestion. In the revised manuscript, we have added a discussion in Section 5 acknowledging the uncertainties associated with the observational data used for validation. These include potential errors due to sensor calibration, platform motion (in the case of buoys), and long-term drift or bias in measurements. We also mention that certain persistent anomalies in wind direction (e.g., at Sites 5 and 6) may be attributed to observational errors.

**Comment:** Wind frame -> wind farm?
**Response:** This typo has been corrected.

---

## Author Comment (AC2)

**General comments:**

The manuscript "Wind data assessment and energy estimation for potential future offshore wind farm development areas in the Scotian Shelf" by the authors Yongying Ma, Jinshan Xu, Yongsheng Wu, Michael Z. Li, Ryan Stanley, Brent Law and Marc Skinner presents a study on the assessment of the wind potential in the Scotian shelf. Wind speed and wind direction data from four different reanalysis data sets (ERA5, CFSv2, NARR, HRDPS) are compared vs. observational data collected in that region in order to identify those model data sets that show the best agreement with observations (ERA5, HRDPS). Data from these data sets is then used to provide input data for wind farm simulations with the tool PyWake for several potential wind farm sites in the Scotian Shelf. The authors carry out a sensitivity study in which they investigate the dependency of the power output of the simulated wind farms from the distance between two wind turbines in those wind farms.

The manuscript does not supply the reader with new methodologies, but applies for the first-time standard tools and data sets to the assessment of wind resources in the geographic region of the Scotian shelf. While the language and overall structure of the manuscript is fine, in my opinion the readability of the paper could profit considerably from slightly restructuring its contents. My suggestion would be e.g. to organize the report on the performance of the different reanalysis data sets not by the metrics but strictly by the data sets. The reader might be most interested in an overall assessment of the data sets and less in a slightly too detailed presentation which data set provides the smallest bias, which the smallest RMSE and so on.

We thank the reviewer for this suggestion. We have adjusted the structure of Section 3 to present the evaluation results in a flow for each wind dataset of ERA5, HRDPS, CFSv2, and NARR.

I have a couple of specific comments which I ask the authors to consider when revising their manuscript.

**Specific comments:**

- **Lacking wind measurements at larger height:**
  According to table 1 the largest measurement height from which data was accessible to the authors was 16 m. Modern wind turbines operate in much larger heights. I'm wondering how transferable the results on the quality of the different reanalysis data sets assessed by the authors for heights close to the ground actually are to larger heights. In my opinion the authors should discuss this point when assessing their results.

We thank the reviewer for pointing out this limitation. Due to limitation of available observation data, there is potential uncertainty of assessing the performance of wind datasets at hub height. We added a discussion to Section 5.1 addressing this fact that the observational wind data used for validation were from relatively low heights (maximum 16 m), which is much lower than the hub height of modern offshore wind turbines (e.g., 150 m).

We acknowledge that while extrapolation was used, performance at higher altitudes may differ due to atmospheric conditions and vertical wind shear. We recognize this as a limitation of our study and suggest that future work include validation using higher-elevation measurements, such as lidar or tall towers, to improve assessment accuracy.

- **Lacking inclusion of atmospheric stability:**
  The vertical wind profile, the turbulence in the marine atmospheric boundary layer and the wake recovery are in practice dependent on the atmospheric stability. However, this parameter is not discussed at all in the manuscript. The authors should at least explain why they have excluded this parameter from their analysis and what the non-consideration of atmospheric stability means for the uncertainty of the results presented by the authors.

  Moreover, I'm lacking a discussion on the impact of atmospheric stability on the seasonal changes of the wind conditions in the Scotian Shelf area. Can seasonal changes in the error metrics be related to lacking consideration of atmospheric stability in the analysis?

We thank the reviewer for highlighting the importance of atmospheric stability in shaping wind profiles and wake recovery. As suggested, we have added a discussion of atmospheric stability to the revised manuscript.

In the section on wind speed extrapolation (Section 2.4), we now clarify that a constant wind shear exponent ($\alpha = 1/7$) was used to extrapolate observed wind speeds from 5 m to 10 m at buoy sites. The choice to exclude the effects of atmospheric stability in this step was due to the lack of available vertical wind profile or stability-related observation data at those locations.

However, for extrapolating wind speeds from 10 m to the turbine hub height (150 m) using wind datasets (i.e., ERA5), we implemented a more refined approach. Specifically, a spatially and temporally varying wind shear exponent was calculated using wind speeds at two heights (10 m and 100 m), allowing for a more realistic representation of vertical wind shear in the ambient atmosphere.

Additionally, in the Discussion section (Section 5.1), we have included comments on the uncertainty introduced by excluding atmospheric stability in the extrapolation of observed buoy data. We also added text in Section 5.1 discussing how seasonal patterns in wind

speed error metrics may, in part, reflect the unaccounted influence of atmospheric stability. For instance, stronger stratification in summer and more unstable conditions in winter could contribute to the observed seasonal variation in the metric of bias.

- **Interpolation of NARR data in time:**
  My suggestion would be to compare the different data sets for a temporal resolution of three hours with each other. The interpolation in time might introduce another uncertainty that is not in the original NARR data itself. It should be possible to quantify the impact of the interpolation in time by comparing error metrics for the original NARR data in the gap-filled NARR data with each other.

  We thank the reviewer for this thoughtful comment. Upon careful re-examination of our methodology, we realized that the original description in the manuscript was inaccurate. In fact, for all datasets, including NARR, we interpolated the wind speed and wind direction data to match the observation timestamps, not to match the hourly resolution of ERA5 dataset as initially stated. We have corrected the description in the revised manuscript to accurately reflect this processing step.

- **Filtering for wind speeds between 2 m/s and 17 m/s:**
  In my opinion it would be also an important criterion whether a reanalysis data set gives the right number of events with wind speeds above cut-out wind speed. I suggest to add such an analysis to the existing analysis. Or is the number of such events too low to have an impact on the calculation of the energy yield in the end?

  We thank the reviewer for this suggestion. As recommended, we have examined the percentage of wind speed events above the cut-out threshold (17 m/s) at 10 m height. At the end of Section 2.5, we have added the following text:
  "*The percentage of time with 10-m wind speeds exceeding 17 m/s was estimated using the ERA5 dataset. These strong wind events occurred approximately 0.3% of the time at both nearshore sites and between 1.8% and 2.5% at offshore sites.*"

- **PyWake:**
  In my opinion the current description of the wind farm model does not contain all the information that would be required by the user to repeat the calculations of the authors. Therefore, I ask the authors to extent the description of the setup of their PyWake runs. E.g., how has the background turbulence intensity been considered in these simulations? Is the model applicable also for calculations of wind turbines that operate in the near wake of other wind turbines? With the smallest turbine distances assumed in the sensitivity study of the authors they might already be in the near-wake range. This is an important comment e.g. for the accuracy of the results presented in figure 9.

  We thank the reviewer for pointing out the need for a more detailed description of the PyWake setup. In the revised manuscript, we have expanded Section 2.6 to include

additional information on the wake model configuration and input parameters used in our simulations.

Specifically, we used the Gaussian-profile wake deficit model developed by Bastankhah and Porté-Agel (2014), as implemented in PyWake. This model assumes self-similarity in the velocity deficit profile, which is supported by experimental findings from Medici and Alfredsson (2006). The model has been validated against both wind tunnel measurements and Large Eddy Simulations by Bastankhah and Porté-Agel (2014), and has shown good agreement for downstream distances greater than approximately 2–3 rotor diameters (D). In our sensitivity study, the minimum turbine spacing was set to 2 D, which lies at the lower bound of the model's validated range. We have clarified this point in the revised manuscript and now include a statement acknowledging that some near-wake effects may not be fully captured at this lower spacings of 2–3 D.

Additionally, we have added a description of the ambient turbulence intensity used in our simulations. A constant turbulence intensity of 0.1 was assumed for all scenarios, as now stated in Section 2.6. We also acknowledge in the Discussion section that the use of a constant turbulence intensity and the application of the Gaussian wake model at short turbine spacings may introduce uncertainty into the results.

- **Error metrics for the wind direction:**
  As averaging of wind directions is often not made correct I encourage the authors to sensitize the readers and present more details in how they handled the jump of the wind direction at 360°/0° in their analysis.

  We thank the reviewer for this comment. Wind direction, as a circular variable, indeed requires careful handling during interpolation and averaging to avoid spurious results. In the revised manuscript (Section 2.3), we have added a detailed explanation of how angular data were treated.

  Specifically, we noted that direct arithmetic operations on wind direction (e.g., averaging 10° and 350° yielding 180°) can lead to incorrect conclusions due to the discontinuity at 360°/0°. To properly address this, we adopted the method described by Berens (2009). Wind directions were first converted to unit vectors via their sine and cosine components. Interpolation and averaging were then applied separately to these components, after which the result was converted back to an angle using the four-quadrant inverse tangent function.

- Page 3, line 81: What is meant by characteristic wind speed in this context?

  We appreciate the reviewer pointing out this ambiguity. Our original use of the term "characteristic wind speed" was intended to refer to the typical time-mean wind speeds across the five stations, which were around 6 m/s. However, we agree that the term may be unclear or misleading in this context. We have revised the sentence to the following

for improved clarity: "*The all-time mean wind speeds at the five stations ranged from 5.35 m/s to 6.18 m/s, with biases between −0.64 m/s and 0.59 m/s, and correlation coefficients close to 0.8.*"

- Table 2: I'm wondering whether this table is actually required. E.g., the information on the time range and the spatial coverage is not of importance for this manuscript.

  We thank the reviewer for this suggestion. After reconsidering the content, we agree that Table 2 is not essential to the manuscript. The key information it contained, such as temporal and spatial resolution, has already been clearly described in the text. To avoid redundancy, we have removed Table 2 in the revised manuscript.

- Page 7, line 174: What is a "naturally-stable" atmospheric condition?

  We thank the reviewer for pointing this out. The term "naturally-stable" was a typographical error. It should have read "neutrally stable atmosphere." We have corrected this in the revised manuscript and also expanded the relevant discussion regarding atmospheric stability in the extrapolation of wind speeds.

- Page 8, line 175: Power law exponent 1/7. Don't ERA5 and HRDPS provide data on other heights as 10 m? If they provide such data, I suggest to determined the power law exponent from the reanalysis data sets. Or is there a special reason why the authors trust more in the 10 m wind speeds than in the wind speeds from other heights in these data sets?

  We appreciate the reviewer's helpful suggestion. In the revised manuscript, we now use ERA5 wind speeds at both 10 m and 100 m heights to calculate a time-varying wind shear exponent, $\alpha$, at each hourly time step. This $\alpha$ is then used to extrapolate wind speeds to the turbine hub height of 150 m. This approach provides a more realistic extrapolation than that using a fixed constant exponent, e.g., 1/7. The methodology has been described in Section 2.4.

- Section 4.1: I'm wondering whether the wind farm simulations for just the seasonal mean wind speed are sufficient here. What does this tell us concerning the energy yield to be expected when the power in the wind is actually depending on the cube of the wind speed?

  We thank the reviewer for raising this important point. The use of seasonal mean wind speed and direction in Section 4.1 was intended to isolate and better understand the impact of turbine spacing on total power production, while also reducing computational costs by avoiding year-long simulations. We acknowledge that wind energy yield is proportional to the cube of wind speed, and therefore, using mean wind speeds alone may not accurately capture the true seasonal energy output.

"*However, it is important to note that because wind turbine power output is proportional to the cube of wind speed, the simulation results using seasonal mean wind speed do not accurately represent seasonal mean energy yield. In Section 4.3, simulations for each PFDA were performed using time-varying wind speed and direction data to provide more realistic estimates of energy production.*"

- Table 5: The explanation of xm and xt should be presented before table 5 is presented. I had difficulties to interpret these parameters without having read the information on these parameters in the text.

  We appreciate the reviewer's comment. To improve clarity, we have revised the caption of Table 5 to explicitly note that the parameters $x_m$ and $x_t$ refer to the results of the piecewise function fitting described in Section 4.2.

**Technical corrections:**

- Abstract, line 1: "The Scotian Shelf is one of the top wind regimes in the world." Regimes should be replaced by regions.

  We thank the reviewer for catching this wording issue. We have replaced "wind regimes" with "wind regions" in the abstract.

---

## Author Response (AR1)

We appreciate the reviewers' thoughtful and constructive feedback. Below, we provide detailed responses to each comment and describe the changes made in the revised manuscript. All section and line references correspond to the revised version. Responses to Reviewer 1 are given on pages 1–6, and those to Reviewer 2 on pages 7–13.

**Response to Reviewer 1**

**General comments:**

This manuscript validated several reanalysis data against measurement data and calculated/discussed wake effect due to turbine spacing at Scotian Shelf offshore site in Canada. In introduction part, there are many paper reviews and well summarized. Although there are some uncertainties of measurement data remain, general trend of each reanalysis data is well presented in validation part. Then, authors calculated wake effect in wind farm for both high dense layout, which could be maximize total power production in wind farm, and low dense layout, which minimize wake loss, and considered and modelled optimal spacing to maximize total power production of wind farm. Also, seasonal differences of these two topics are compared and well presented. Although the result of this manuscript itself may site specific, the methodology will be good reference in future project and there are some scientific interests.

In conclusion, a reviewer consider that this manuscript can be accepted with MINOR REVISION.

**Specific Comments and Responses**

**Comment:** ERA 5 is explained as mean percentage for all stations, while JRA-55 and MERRA-2 are explained as its ranges. Although it is not a part of your research, it should be explained by fair criteria.
**Response:** The authors intended to express that the mean percentage for the other three datasets falls within the range of -54.22% to 42%. This presentation may be unclear. Now we have improved our expression in the revised manuscript.
**Changes:** Lines 61–64 now read**:**
"*The authors found that ERA5 demonstrated the best overall performance among the five reanalysis wind dataset products, with ERA5 exhibiting a mean percent bias for all stations of -4.54 %, while the mean percent bias was -54.22% for JRA-55, -49.63% for CFSv2 and 42.03% for MERRA-2*".

**Comment:** It is suggested show station height above sea level and mounted height of measurement devices in the table.
**Response:** We have updated Table 1 to include the suggested information.

**Changes:** A new column titled "Instrument Height (m)" was added to Table 1.
* * *
**Comment:** Authors have to explain how Quality level flag (if exist) is handled. Also, explanations about measurement devices, and its consistency in validation period, definition of wind direction (i.e. true north or magnetic north) are missing. It is also suggested to add a table about monthly data availability.

**Response:** We added text about the quality level flags and the description of our quality control procedure. Information about wind measurement instruments was added. The definition of wind direction is now stated in Section 2.1. The monthly data availability is shown in a new figure in the appendix.

**Changes:** Lines 147–151 now read:

"*Wind speed and direction data obtained from meteorological stations and marine buoys did not include quality flags (quality flags were only available for ocean wave variables in the buoy data). To ensure data reliability, a basic quality control procedure was applied: wind speed values equal to 0 m/s or greater than 50 m/s were considered erroneous and excluded from the analysis.*"
* * *
**Comment:** Although authors assume neutral stratification and alpha = 1/7, which is international standard, it is not clear if this assumption is correct in Scotian Shelf, and if not how big impact is given on validation result.

**Response:** We added the reason for choosing the alpha value, following the IEC standard. The discussion about the variability of $\alpha$ and dependence on atmospheric stability was also added. In the discussion section, we also added content about the magnitude of impacts of choosing a constant $\alpha$ on the estimated wind speed.

**Changes:** Lines 687–710 in the revised manuscript.
* * *
**Comment:** Need explanations about symbols "U" and "z".

**Response:** We have added definitions for these symbols in the Methods section.

**Changes:** Lines 193–194 in the revised manuscript.
* * *
**Comment:** ERA5 has UV component 100m above ground and it reduces vertical extrapolation uncertainty. Authors need to explain the reasons why 10m height data is used, if there is.

**Response:** We now use both 10 m and 100 m wind speed data to calculate time-varying $\alpha$ values, which are then used to extrapolate to 150 m height.

**Changes:** Lines 199–205 in the revised manuscript.
* * *
**Comment:** Need explanation about symbol "$\bar{O}$".

**Response:** We clarified that $\bar{O}$ denotes the average value of the observations.

**Changes:** Line 225 in the revised manuscript**.**
* * *
**Comment:** "measured at 10m" should be "at 10m".
**Response:** Corrected as suggested.
**Changes:** Line 226 in the revised manuscript**.**
* * *
**Comment:** higher -> lower? The sentence is bit difficult to understand.
**Response:** Confirmed that ERA5's RMSE is indeed higher than HRDPS's and revised the sentence for clarity.
**Changes:** Lines 288–289 now read:
"At nearshore Sites 1 and 2, ERA5's RMSE was higher than HRDPS, at 1.76 ± 0.20 m/s and 2.08 ± 0.37 m/s, respectively.".
* * *
**Comment:** CFSv2 is better than ERA5 at Site 2. It is better to add the explanation.
**Response:** Added a likely explanation: CFSv2's slightly finer resolution (0.2° vs. 0.25°) better captures local wind gradients in nearshore environments.
**Changes:** Lines 318–321 now read:
*"The better performance of CFSv2 at Site 2 than ERA5 in terms of smaller RMSE and bias closer to 0 was likely attributed to its slightly finer horizontal resolution (0.2° versus 0.25° for ERA5), which can better capture local wind gradients in the nearshore environment."*
* * *
**Comment:** "RMSE values tended to increase during winter months…", This just may be because magnitude of wind speed is high. Use of normalized RMSE may help further understanding.
**Response:** We thank the reviewer for this insightful comment. After recalculating normalized RMSE (RMSE divided by mean observed wind speed), we confirmed the reviewer's observation: the higher RMSE in winter was due to greater wind speeds. Normalized RMSE revealed the opposite seasonal pattern: lower in winter, higher in summer.
**Changes:** Lines 337–339 now read:
*"However, this pattern was primarily due to the higher wind speed magnitudes in winter compared to summer. When normalized RMSE (calculated as RMSE divided by the mean observed wind speed) was used, it was found that the normalized errors were actually smaller in winter and larger in summer."*
* * *
**Comment:** The use of the word "overestimated" toward wind direction is weird. It is suggested to use "shifted (anti)clockwise".
**Response:** We rephrased the wording as suggested.
**Changes:** Applied to Lines 425, 431, and 441 of the revised manuscript**.**

**Comment:** "April to December in 2022" Need to explain that this explanation is about Site 5. It also seems that July 2020 to July 2021 at Site 6 shows different trend, compared to other years on the site. Need a comment that how authors think this about this period. Also, explanation that how authors handle these measurement errors when calculate aggregate metrics mentioned in text, Figure 8, Table 4 etc. in this sub section is needed.

**Response:** We thank the reviewer for this careful observation. In the revised manuscript, we specified that April to December 2022 applies to Site 5, and also identified the anomalous period for Site 6.

The selection criterion for wind direction data was the same as that used for wind speed: all data from months with at least 120 valid hourly records were included in the calculation of both monthly and aggregated metrics. Although the periods of large errors at Sites 5 and 6 are suspected to have resulted from systematic observational issues, the metrics presented in Figure 8 and Table 4 include these periods. We acknowledge that such anomalies may influence the results; however, we chose not to exclude them in order to maintain a consistent and objective screening criterion across all sites. This decision is reflected in the box plots in Figure 8, where outliers indicate the presence of anomalous data during specific periods at Sites 5 and 6. Notably, offshore sites exhibit more outliers than nearshore sites, which suggests the influence of the suspected observational errors at Sites 5 and 6.

**Changes:**
- Lines 444–446 now read: "*At the buoy based offshore Sites 5 and 6, HRDPS (as well as the other datasets) exhibited notably large bias values during specific periods, i.e., from April to December 2022 at Site 5 and from June 2020 to July 2022 at Site 6, which was likely due to systematic observational errors in the recorded wind direction.*"

- Lines 452–454 now read:
  "*It is noted that for the offshore group, periods with suspected systematic observational errors at Sites 5 and 6 were not excluded from the analysis. All data from months with at least 120 valid hourly records were retained to maintain a consistent screening criterion across all sites. As a result, the box plots in Figure 8 reflect the influence of these anomalies, as indicated by a greater number of outliers at offshore sites compared to nearshore sites.*"

**Comment:** It is better to add text that "x_t" and "x_m" will be explained later in section 4.2, like as Figure 9.

**Response:** We added a note in the Table 4 caption to indicate that x_t and x_m are defined later in Section 4.2 of the manuscript.

**Changes:** Caption for Table 4 updated accordingly.

**Comment:** Need to explain data period for 'spatial and seasonal mean'.
**Response:** Revised to specify that values were averaged over ERA5 grid points within each PFDA for 2019–2023.
**Changes:** Added clarification to Lines 490–492.
* * *
**Comment:** "These values were determined as the spatial and seasonal mean for each PFDA." I understand wind speeds and wind directions shown in Table 5 are used in this section to reduce computational cost. Is my understanding correct? If so, why Punit in Figure 9 reaches 15MW? All wind speeds in Table 5 are lower than 10.6m/s, which is rated wind speed of IEA 15MW turbine.
**Response:** We clarified that the wind speeds in Table 4 (formerly Table 5) of the revised manuscript are given at 10 m height, whereas the power curve in Figure 2 corresponds to wind speeds at 150 m. In Line 240, we now note that a rated wind speed of 10.6 m s$^{-1}$ at 150 m corresponds to approximately 7.2 m s$^{-1}$ at 10 m, assuming a shear exponent of 0.14.
**Changes:** Added clarification to the caption of Table 4.
* * *
**Comment:** "Punit revealed the average turbine efficiency" I understand Punit is average of "all wind turbine" in a PFDA. If so, the authors need to explain that (i.e. what consist of average). Also, it is not "efficiency" but "power output", isn't it?
**Response:** We clarified that Punit represents the average power output per turbine, calculated as Ptotal divided by the number of turbines. We also replaced the term "turbine efficiency" with "wake efficiency" for accuracy.
**Changes:** Lines 501–507 updated accordingly.
* * *
**Comment:** It is suggested to write brief explanation (e.g. Summer, 9.6D) in each figure.
**Response:** Added labels such as "Summer, 9.6D" to each panel in Figure 11(a) to clarify layout.
**Changes:** Text labels added directly to Figure 11(a).
* * *
**Comment:** "10 additional time series"
It is difficult to understand why 10? It is suggested to add brief explanation (e.g. 5 yeas data and ±).
**Response:** We thank the reviewer for requesting clarification. We have now expanded the explanation in the revised manuscript to clarify why 10 additional time series were used. Unlike wind speed, where power production is a monotonic function, the effect of wind direction on power output is non-monotonic due to wake interactions. Within the ±RMSE bounds, we generated a range of perturbed wind direction time series at small directional intervals to capture this variability. Through sensitivity testing, we found that 10 evenly spaced perturbations were sufficient to capture the range of power variation while keeping computational demands reasonable. This clarification has been added to Section 4.3 of the revised manuscript.
**Changes:** Lines 640–645 now read:

"*To address this, multiple simulations were performed across a range of possible wind directions to capture the potential variation in power output. For each month, the range of possible wind directions was defined by adding and subtracting the monthly RMSE from the original wind direction time series. Within this range, additional wind direction time series were generated, with values evenly distributed between the upper and lower bounds. Based on sensitivity testing, it was found that using 10 perturbations provided sufficient resolution to capture variability in power production without incurring excessive computational cost.*"
* * *
**Comment:** Discussion regarding uncertainty of measurement data is needed.
**Response:** We thank the reviewer for this suggestion. In the revised manuscript, we have added a discussion in Section 5 acknowledging the uncertainties associated with the observational data used for validation. These include potential errors due to calibration process, limitations in instrument accuracy, degradation over time (e.g., corrosion or component aging), environmental influences (such as icing and salt spray), and effects of nearby terrain, structures that distort airflow.
**Changes:** Added new text at Lines 711–718.
* * *
**Comment:** Wind frame -> wind farm?
**Response:** This typo has been corrected.
**Changes:** Updated wording at Line 741.

**Response to Reviewer 2**

**General comments**

**Reviewer comment:**
The manuscript "Wind data assessment and energy estimation for potential future offshore wind farm development areas in the Scotian Shelf" by the authors Yongying Ma, Jinshan Xu, Yongsheng Wu, Michael Z. Li, Ryan Stanley, Brent Law and Marc Skinner presents a study on the assessment of the wind potential in the Scotian shelf. Wind speed and wind direction data from four different reanalysis data sets (ERA5, CFSv2, NARR, HRDPS) are compared vs. observational data collected in that region in order to identify those model data sets that show the best agreement with observations (ERA5, HRDPS). Data from these data sets is then used to provide input data for wind farm simulations with the tool PyWake for several potential wind farm sites in the Scotian Shelf. The authors carry out a sensitivity study in which they investigate the dependency of the power output of the simulated wind farms from the distance between two wind turbines in those wind farms.

The manuscript does not supply the reader with new methodologies, but applies for the first-time standard tools and data sets to the assessment of wind resources in the geographic region of the Scotian shelf. While the language and overall structure of the manuscript is fine, in my opinion the readability of the paper could profit considerably from slightly restructuring its contents. My suggestion would be e.g. to organize the report on the performance of the different reanalysis data sets not by the metrics but strictly by the data sets. The reader might be most interested in an overall assessment of the data sets and less in a slightly too detailed presentation which data set provides the smallest bias, which the smallest RMSE and so on.

**Response:** We agree that this restructuring improves readability. Section 3 has been revised so that evaluation results are presented sequentially for each wind dataset (ERA5, HRDPS, CFSv2, and NARR) rather than by metric. This format allows for a more cohesive assessment of each dataset's overall performance.

**Changes:** Updated text in Lines 285–334 and 418–441.

I have a couple of specific comments which I ask the authors to consider when revising their manuscript.

**Specific comments:**

**Comment:** Lacking wind measurements at larger height:
According to table 1 the largest measurement height from which data was accessible to the authors was 16 m. Modern wind turbines operate in much larger heights. I'm wondering how transferable the results on the quality of the different reanalysis data sets assessed by the authors for heights close to the ground actually are to larger heights. In my opinion the authors should discuss this point when assessing their results.

**Response:** We thank the reviewer for pointing out this limitation. Due to the lack of observational data at hub height, there is uncertainty in transferring performance assessments from 10 m to 150 m. We have added a discussion in Section 5.1 noting this limitation and citing studies that show the correspondence between low and high-altitude performance can be weak.

**Changes:** In revised manuscript (Lines 719–725), we added:
"*The wind datasets in this study were assessed at 10 m above the surface, well below the turbine hub height (150 m). The transferability of performance from low to high altitudes is uncertain. Ji et al. (2025) found that while $R^2$ at 10 m and 100 m were moderately correlated, bias and MAE showed weak correspondence. Similarly, Liu et al. (2023a) reported that using the power law relationship to estimate higher-level winds led to declining performance, with RMSE increasing from 1.50 m/s at 120 m to 2.42 m/s at 200 m. These findings suggest that strong performance at low heights does not guarantee similar accuracy at hub height. Validating wind datasets with observations at turbine operational heights, obtained using lidar or tall masts, can provide more accurate assessments for wind energy applications.*"
* * *
**Comment:** Lacking inclusion of atmospheric stability:
The vertical wind profile, the turbulence in the marine atmospheric boundary layer and the wake recovery are in practice dependent on the atmospheric stability. However, this parameter is not discussed at all in the manuscript. The authors should at least explain why they have excluded this parameter from their analysis and what the non-consideration of atmospheric stability means for the uncertainty of the results presented by the authors.

Moreover, I'm lacking a discussion on the impact of atmospheric stability on the seasonal changes of the wind conditions in the Scotian Shelf area. Can seasonal changes in the error metrics be related to lacking consideration of atmospheric stability in the analysis?

**Response:** We thank the reviewer for highlighting the importance of atmospheric stability in shaping wind profiles and wake recovery. As suggested, we have added a discussion of atmospheric stability to the revised manuscript.

In Section 2.4, we now clarify that a constant shear exponent (α = 1/7) was used to extrapolate buoy winds from 5 m to 10 m due to the lack of stability-related observations at those locations.

For ERA5 based extrapolation from 10 m to 150 m, we calculated a spatially and temporally varying shear exponent, α, using wind speed from the ERA5 dataset at 10 m and 100 m heights, which allowed for a more realistic representation of vertical wind shear in the ambient atmosphere. Additionally, in the Discussion section (Section 5.1), we have included comments on the uncertainty introduced by excluding atmospheric stability in the extrapolation of observed buoy data. We also added text in Section 5.1 discussing how seasonal patterns in wind speed error metrics may, in part, reflect the unaccounted influence of atmospheric stability. For instance, stronger stratification in summer and more unstable conditions in winter could contribute to the observed seasonal variation in the metric of bias.

**Changes:** Text added in the revised manuscript at Lines 687–710.
* * *
**Comment:** Interpolation of NARR data in time:
My suggestion would be to compare the different data sets for a temporal resolution of three hours with each other. The interpolation in time might introduce another uncertainty that is not in the original NARR data itself. It should be possible to quantify the impact of the interpolation in time by comparing error metrics for the original NARR data in the gap-filled NARR data with each other.

**Response:** We thank the reviewer for this thoughtful comment. Upon careful re-examination of our methodology, we realized that the original description in the manuscript was inaccurate. In fact, for all datasets, including NARR, we interpolated the wind speed and wind direction data to match the observation timestamps, not to match the hourly resolution of ERA5 dataset as initially stated. We have corrected the description in the revised manuscript to accurately reflect this processing step.

**Changes:** Corrected method description in Lines 177–179.
* * *
**Comment:** Filtering for wind speeds between 2 m/s and 17 m/s:
In my opinion it would be also an important criterion whether a reanalysis data set gives the right number of events with wind speeds above cut-out wind speed. I suggest to add such an analysis to the existing analysis. Or is the number of such events too low to have an impact on the calculation of the energy yield in the end?

**Response:** We thank the reviewer for this suggestion. As recommended, we have examined the percentage of wind speed events above the cut-out threshold (17 m/s) at 10 m height.

**Changes:** Text were added at Lines 230–232:
"*The percentage of time with 10 m wind speeds exceeding 17 m/s was estimated using the*

*ERA5 dataset. These strong wind events occurred approximately 0.3% of the time at both nearshore sites and between 1.8% and 2.5% at offshore sites."*
* * *
**Comment:** PyWake:
In my opinion the current description of the wind farm model does not contain all the information that would be required by the user to repeat the calculations of the authors. Therefore, I ask the authors to extent the description of the setup of their PyWake runs. E.g., how has the background turbulence intensity been considered in these simulations? Is the model applicable also for calculations of wind turbines that operate in the near wake of other wind turbines? With the smallest turbine distances assumed in the sensitivity study of the authors they might already be in the near-wake range. This is an important comment e.g. for the accuracy of the results presented in figure 9.

**Response:** We thank the reviewer for pointing out the need for a more detailed description of the PyWake setup. In the revised manuscript, we have expanded Section 2.6 to include additional information on the wake model configuration and input parameters used in our simulations.
Specifically, we used the Gaussian-profile wake deficit model developed by Bastankhah and Porté-Agel (2014), as implemented in PyWake. This model assumes self-similarity in the velocity deficit profile, which is supported by experimental findings from Medici and Alfredsson (2006). The model has been validated against both wind tunnel measurements and Large Eddy Simulations by Bastankhah and Porté-Agel (2014), and has shown good agreement for downstream distances greater than approximately 2–3 rotor diameters (D). In our sensitivity study, the minimum turbine spacing was set to 2 D, which lies at the lower bound of the model's validated range. We have clarified this point in the revised manuscript and now include a statement acknowledging that some near-wake effects may not be fully captured at this lower spacings of 2–3 D.
Additionally, we have added a description of the ambient turbulence intensity used in our simulations. A constant turbulence intensity of 0.1 was assumed for all scenarios, as now stated in Section 2.6.

**Changes:** In Lines 253–264 of the revised manuscript, we added model details, turbulence intensity, and near-wake limitations.
* * *
**Comment:** Error metrics for the wind direction:
As averaging of wind directions is often not made correct I encourage the authors to sensitize the readers and present more details in how they handled the jump of the wind direction at 360°/0° in their analysis.

**Response:** We thank the reviewer for this comment. Wind direction, as a circular variable, indeed requires careful handling during interpolation and averaging to avoid spurious results. In

the revised manuscript (Section 2.3), we have added a detailed explanation of how angular data were treated.

**Changes:** Lines 180–187 in the revised manuscript now read:
"*In contrast, handling wind direction required special attention due to the circular nature of angular data, e.g., 0° and 360° represent the same direction. Direct linear interpolation or averaging, as used in the calculation of metrics described in Section 2.5, can produce incorrect results. For instance, the arithmetic average of 10° and 350° is 180°, corresponding to a wind direction from south to north, even though both original directions are close to the direction from north to south. To address this issue, the method described by Berens (2009) was adopted. Angular values were first transformed into unit vectors using their sine and cosine components. Linear interpolation or averaging was then applied separately to each component, and the resulting vectors were converted back into angles using the four-quadrant inverse tangent function.*"
* * *
**Comment:** Page 3, line 81: What is meant by characteristic wind speed in this context?

**Response:** We appreciate the reviewer pointing out this ambiguity. Our original use of the term "characteristic wind speed" was intended to refer to the typical time-mean wind speeds across the five stations, which were around 6 m/s. However, we agree that the term may be unclear or misleading in this context. We have revised the sentence for improved clarity.

**Changes:** Lines 80–82 in the revised manuscript now read:
"*The all-time mean wind speeds at the five stations ranged from 5.35 m/s to 6.18 m/s, with biases between −0.64 m/s and 0.59 m/s, and correlation coefficients close to 0.8.*"
* * *
**Comment:** Table 2: I'm wondering whether this table is actually required. E.g., the information on the time range and the spatial coverage is not of importance for this manuscript.

**Response:** We thank the reviewer for this suggestion. After reconsidering the content, we agree that Table 2 is not essential to the manuscript. The key information it contained, such as temporal and spatial resolution, has already been clearly described in the text. To avoid redundancy, we have removed Table 2 in the revised manuscript.

**Changes:** Former Table 2 was removed.
* * *
**Comment:** Page 7, line 174: What is a "naturally-stable" atmospheric condition?

**Response:** We thank the reviewer for pointing this out. The term "naturally-stable" was a typographical error. It should have read "neutrally stable atmosphere." We have corrected this in the revised manuscript.

**Changes:** The phrase "naturally-stable" was removed from text.
* * *
**Comment:** Page 8, line 175: Power law exponent 1/7. Don't ERA5 and HRDPS provide data on other heights as 10 m? If they provide such data, I suggest to determined the power law exponent from the reanalysis data sets. Or is there a special reason why the authors trust more in the 10 m wind speeds than in the wind speeds from other heights in these data sets?

**Response:** We appreciate the reviewer's helpful suggestion. In the revised manuscript, we now use ERA5 wind speeds at both 10 m and 100 m heights to calculate a time-varying wind shear exponent, α, at each hourly time step. This α is then used to extrapolate wind speeds to the turbine hub height of 150 m. This approach provides a more realistic extrapolation than that using a fixed constant exponent, e.g., 0.14. The methodology has been described in Section 2.4.

**Changes:** Text was added at lines 199–206.
* * *
**Comment:** Section 4.1: I'm wondering whether the wind farm simulations for just the seasonal mean wind speed are sufficient here. What does this tell us concerning the energy yield to be expected when the power in the wind is actually depending on the cube of the wind speed?

**Response:** We thank the reviewer for raising this important point. The use of seasonal mean wind speed and direction in Section 4.1 was intended to isolate and better understand the impact of turbine spacing on total power production, while also reducing computational costs by avoiding year-long simulations. We acknowledge that wind energy yield is proportional to the cube of wind speed, and therefore, using mean wind speeds alone may not accurately capture the true seasonal energy output.

**Changes:** The second paragraph of Section 4.1 includes following text:
> "*However, it is important to note that because wind turbine power output is proportional to the cube of wind speed, the simulation results using seasonal mean wind speed do not accurately represent seasonal mean energy yield. In Section 4.3, simulations for each PFDA were performed using time-varying wind speed and direction data to provide more realistic estimates of energy production.*"
* * *
**Comment:** Table 5: The explanation of xm and xt should be presented before table 5 is presented. I had difficulties to interpret these parameters without having read the information on these parameters in the text.

**Response:** We appreciate the reviewer's comment. To improve clarity, we have revised the caption of Table 5 to explicitly note that the parameters $x_m$ and $x_t$ refer to the results of the piecewise function fitting described in Section 4.2.

**Changes:** Caption for Table 4 (formerly Table 5) updated accordingly.
* * *
**Technical corrections:**

**Comment:** Abstract, line 1: "The Scotian Shelf is one of the top wind regimes in the world." Regimes should be replaced by regions.

**Response:** We thank the reviewer for catching this wording issue. We have now corrected this in the abstract.

**Changes:** Text '"wind regimes" was replaced with "wind regions" in the abstract.

---

## Referee Report (RR1)

I would like to thank the authors for revising their manuscript. In my opinion the manuscript is now ready for being accepted for publication in Wind Energy Science. Finally, I would like to suggest some further (optional) changes to the manuscript. The line numbers provided in my suggestions refer to line numbers in the document with the tracked changes.

1. Line 76: "... ERA5 ranged from ..." should be changed to "... ERA5 ranging from ..."

2. Figure 1: An information on the meaning of the blue-shaded areas should be given in the caption of the figure.

3. Equation 6: Please add an information on the meaning of $R^2$, i.e. please mention that $R^2$ is the coefficient of determination.

4. Line 279: It might be helpful to add an information on the turbulence intensity that was assumed in the study of Bastankah and Porté-Agel (2014). The reason is that e.g. Sorensen et al. )2014) show that the length of the near-wake region is inversely proportional to the logarithm of the turbulence intensity.

Jens N Sørensen et al 2014, J. Phys.: Conf. Ser. 524 012155

5. Line 293: Please add an information on the turbulence intensity that was use

d in the calculations with PyWake.

6. Line 392: Please change "Sites" to "sites".

7. LIne 846: Please add a reason why you decided to use a value of 0.14 for the shear exponent.